# GLP-1R associates with VAPB and SPHKAP at ERMCSs to regulate β-cell mitochondrial remodelling and function

Glucagon-like peptide-1 receptor (GLP-1R) agonists (GLP-1RAs) ameliorate mitochondrial health by increasing mitochondrial turnover in metabolically relevant tissues. Mitochondrial adaptation to metabolic stress is crucial to maintain pancreatic β-cell function and prevent type 2 diabetes (T2D) progression. While the GLP-1R is well-known to stimulate cAMP production leading to Protein Kinase A (PKA) and Exchange Protein Activated by cyclic AMP 2 (Epac2) activation, there is a lack of understanding of the molecular mechanisms linking GLP-1R signalling with mitochondrial and β-cell functional adaptation. Here, we present a comprehensive study in β-cell lines and primary islets that demonstrates that, following GLP-1RA stimulation, GLP-1R-positive endosomes associate with the endoplasmic reticulum (ER) membrane contact site (MCS) tether VAPB at ER-mitochondria MCSs (ERMCSs), where active GLP-1R engages with SPHKAP, an A-kinase anchoring protein (AKAP) previously linked to T2D and adiposity risk in genome-wide association studies (GWAS). The inter-organelle complex formed by endosomal GLP-1R, ER VAPB and SPHKAP triggers a pool of ERMCS-localised cAMP/PKA signalling via the formation of a PKA-RIα biomolecular condensate which leads to changes in mitochondrial contact site and cristae organising system (MICOS) complex phosphorylation, mitochondrial remodelling, and β-cell functional adaptation, with important consequences for the regulation of β-cell insulin secretion and survival to stress.

Maintenance of mitochondrial function and adaptation to changes in nutritional state is vital for the preservation of pancreatic β-cell regulated secretion of insulin[1], the main hormone in charge of lowering blood glucose concentrations, with disruptions in this process underlying the development and/or progression of type 2 diabetes (T2D)[1,2]—an uncontrolled pandemic affecting ∼500 million people worldwide, and causing ∼1.5 million deaths per year[3]. The incretin glucagon-like peptide-1 (GLP-1), released from the gut during food intake, amplifies glucose-stimulated insulin release by binding to and activating its cognate G protein-coupled receptor (GPCR), the GLP-1 receptor (GLP-1R), triggering cAMP generation and downstream Protein Kinase A (PKA) and Exchange Protein Activated by cyclic AMP 2

(Epac2) signalling[4], a process that has been exploited to develop pharmacological GLP-1R agonists (GLP-1RAs) as T2D therapies. There is extensive data highlighting the beneficial role of GLP-1RAs in the maintenance and/or restoration of mitochondrial function in β-cells and beyond, with reports emphasising their capacity to induce mitochondrial remodelling/turnover and functional adaptation[5–8], two closely intertwined processes essential for the preservation of optimal metabolic activity[9]. While a range of GLP-1R-induced pathways have been suggested, there is no clear description of the molecular mechanism engaged by the receptor to modulate mitochondrial remodelling and improve mitochondrial function. Here, we present evidence in β-cells and primary mouse and human islets which

✉ e-mail: a.tomas-catala@imperial.ac.uk

demonstrates that, following agonist binding and internalisation, endosomal GLP-1R binds to endoplasmic reticulum (ER)-localised ER-mitochondria membrane contact site (ERMCS) organising factor VAPB to engage SPHKAP, a PKA-RIα-specific A-kinase anchoring protein (AKAP) whose gene coding variants have previously been linked to T2D and high BMI risk in genome-wide association studies (GWAS). We further show that SPHKAP itself localises to ERMCSs via its direct interaction with VAPB through a pFFAT motif present in SPHKAP. Establishment of a membrane contact site (MCS) between endosomal GLP-1R, ERMCS-localised VAPB and SPHKAP leads to the generation of a highly-localised cAMP/PKA signalling hub that triggers the PKA-dependent phosphorylation of the mitochondrial contact site and cristae organising system (MICOS) complex, mitochondrial remodelling, and improved mitochondrial function, leading to potentiation of insulin secretion and survival to ER stress downstream of GLP-1R action. This is, to our knowledge, the first description of a three-way contact between an endosomal GPCR, the ERMCS tether VAPB, and a PKA-RIα-recruiting AKAP, which uncovers the assembly of a GLP-1R-induced ERMCS-localised PKA biomolecular condensate to control mitochondrial turnover and function. Our data not only unveils a previously unknown pathway of ERMCS-localised GPCR signalling into mitochondria, potentially applicable to other GPCRs beyond the GLP-1R, but also opens the door to further roles of the GLP-1R in the regulation of mitochondrial lipid metabolism, a process tightly linked to changes in mitochondrial morphology governed by MICOS complex activity[10]. In summary, this study has uncovered the molecular mechanism employed by the GLP-1R to achieve mitochondrial regeneration leading to optimised β-cell function, with important repercussions for our understanding of the role of GLP-1RAs in disorders such as T2D, obesity, and neurodegeneration.

## Results

### Human GLP-1R β-cell interactome reveals binding of agonist-stimulated GLP-1R to signalling effectors, with strong prevalence of ERMCS-localised proteins

To gain a comprehensive understanding of the specific downstream partners and signalling mediators engaged by active GLP-1Rs in pancreatic β-cells, we performed an interactome analysis of proteins that bind to the receptor under vehicle *versus* 5-minute stimulation with the GLP-1RA exendin-4 in a subline of rat INS-1 832/3 β-cells stably expressing SNAP/FLAG-tagged human GLP-1R (SNAP/FLAG-hGLP-1R). Mass spectrometry (MS) analysis of GLP-1R co-immunoprecipitants revealed a list of proteins enriched in agonist-stimulated *versus* unstimulated conditions (Fig. 1a, b and Supplementary Fig. 1a). Gene ontology analysis of the identified factors indicated that these were involved in pathways related to endomembranes and the ER (Supplementary Fig. 1b). Amongst these factors, we found a high number of proteins localised to ERMCSs (Fig. 1c), which are points of close interaction, but not fusion, between the ER and mitochondria that fulfil specific functions including cross-organelle ion and lipid sensing and transfer, and signal transmission to coordinate mitochondrial remodelling[11]. While known GLP-1R interactors such as Gα$_S$, AP2, Rab5, and β-arrestin 2 were reassuringly identified in our interactomics list, the most prominently enriched factor interacting with active GLP-1R was VAMP Associated Protein B (VAPB), followed by its homologue VAPA, integral ER proteins whose main role is the establishment and maintenance of organelle-ER interactions, or MCSs[12,13], including those involving endosomes and/or mitochondria. We next validated the active GLP-1R binding to VAPB inferred from our MS results by co-immunoprecipitating both EGFP-VAPB (Fig. 1d) and endogenous VAPB (Fig. 1e and Supplementary Fig. 1c) with SNAP/FLAG-hGLP-1R in vehicle *versus* exendin-4-stimulated INS-1 832/3 SNAP/FLAG-hGLP-1R cells. Confocal microscopy analysis of vehicle *versus* exendin-4-stimulated SNAP/FLAG-hGLP-1R co-localisation with EGFP-VAPB in these cells revealed that, upon exendin-4 stimulation, the GLP-1R traffics from the

plasma membrane to endosomes intimately associated with VAPB-positive ER regions (Supplementary Fig. 1d,e), with the ER often surrounding a central core of GLP-1R-positive signal (Fig. 1f), suggesting that GLP-1R-containing endosomes directly contact VAPB-positive ER membranes, likely via MCSs. This hypothesis was validated by transmission electron microscopy (TEM) localisation of SNAP/FLAG-hGLP-1R (labelled at the cell surface with SNAP-biotin plus streptavidin-gold particles in living cells prior to exendin-4 stimulation) to endosome-ER MCSs (Fig. 1g). Further microscopy experiments performed to visualise GLP-1R engagement with EGFP-VAPB in response to exendin-4 stimulation include confocal imaging of serial optical sections covering the entire cell thickness after SNAP/FLAG-hGLP-1R surface labelling with a membrane impermeable SNAP-tag probe followed by exendin-4 stimulation, represented as a maximum intensity projection (MIP) in Fig. 1h, and time-lapse confocal microscopy analysis of exendin-4-induced SNAP/FLAG-hGLP-1R trafficking from the plasma membrane to endosomes in close proximity to EGFP-VAPB-positive ER regions (Supplementary Fig. 1f and Supplementary Movie 1). Two-colour nanometre-scale resolution imaging by Minimal Photon Fluxes (MINFLUX), an ultra-high resolution microscopy technique that combines aspects of single-molecule localisation microscopy and stimulated emission depletion (STED) microscopy[14], was employed to determine the average distance between endosomal SNAP/FLAG-hGLP-1R and VAPB, estimated at around 28 nm (Fig. 1i, j), compatible with the expected distance of an ER-endosome MCS[15].

The interaction of active GLP-1R with VAPB was deemed functionally relevant, as it was no longer present when VAPB P56S, a self-aggregating loss-of-function VAPB mutant associated with amyotrophic lateral sclerosis (ALS)[16,17], was expressed instead of its wildtype (WT) counterpart (Supplementary Fig. 1g–i), and as both VAPB knockdown by RNAi (Fig. 1k and Supplementary Fig. 1j, k), and overexpression of VAPB P56S, but not of WT VAPB (Supplementary Fig. 1l), significantly reduced the capacity of GLP-1R to potentiate insulin secretion from INS-1 832/3 cells.

### GLP-1R binding to VAPB requires GLP-1R internalisation and is differentially modulated by GLP-1RAs with varying GLP-1R internalisation propensities

To elucidate whether active GLP-1R requires its prior internalisation to engage VAPB, we co-transfected INS-1 832/3 SNAP/FLAG-hGLP-1R cells with dominant negative (K44A) mutants of dynamin 1 and 2 to block GLP-1R endocytosis triggered by exendin-4 (Supplementary Fig. 2a). Under these conditions, increased binding of the receptor to VAPB in stimulated *versus* vehicle conditions was abrogated (Fig. 2a), demonstrating that active GLP-1R interacts with VAPB following its agonist-induced internalisation. Supporting this notion, MS interactomic analyses of GLP-1R co-immunoprecipitates in INS-1 832/3 SNAP/FLAG-hGLP-1R cells after a 5-minute stimulation with the endogenous agonist GLP-1, as well as with the exendin-4-based biased GLP-1RAs exendin-F1 and exendin-D3, previously shown to trigger differing degrees of GLP-1R internalisation[18,19], resulted in marked differences in the propensity of the receptor to associate with VAPB (Supplementary Fig. 2b), with agonists that trigger robust receptor internalisation (GLP-1, exendin-4, exendin-D3) showing a stronger degree of GLP-1R-VAPB interaction compared to slow GLP-1R-internalising agonist exendin-F1. Marked kinetic differences in GLP-1R-VAPB interaction following exendin-4 *versus* exendin-F1 stimulation were also demonstrated in INS-1 832/3 cells by a novel NanoBRET assay, with Nanoluciferase-tagged GLP-1R co-expressed with Venus-tagged VAPB (Fig. 2b and Supplementary Fig. 2c). We extended our analysis to a panel of seven GLP-1RAs, including, as well as exendin-4, exendin-F1 and exendin-D3, fatty acid-modified semaglutide[20] and tirzepatide[21], as well as small molecules orforglipron[22] and danuglipron[23]. We first tested these GLP-1RAs for their capacity to induce GLP-1R internalisation in purified mouse islets transduced with SNAP/FLAG-hGLP-1R-expresssing adenoviruses

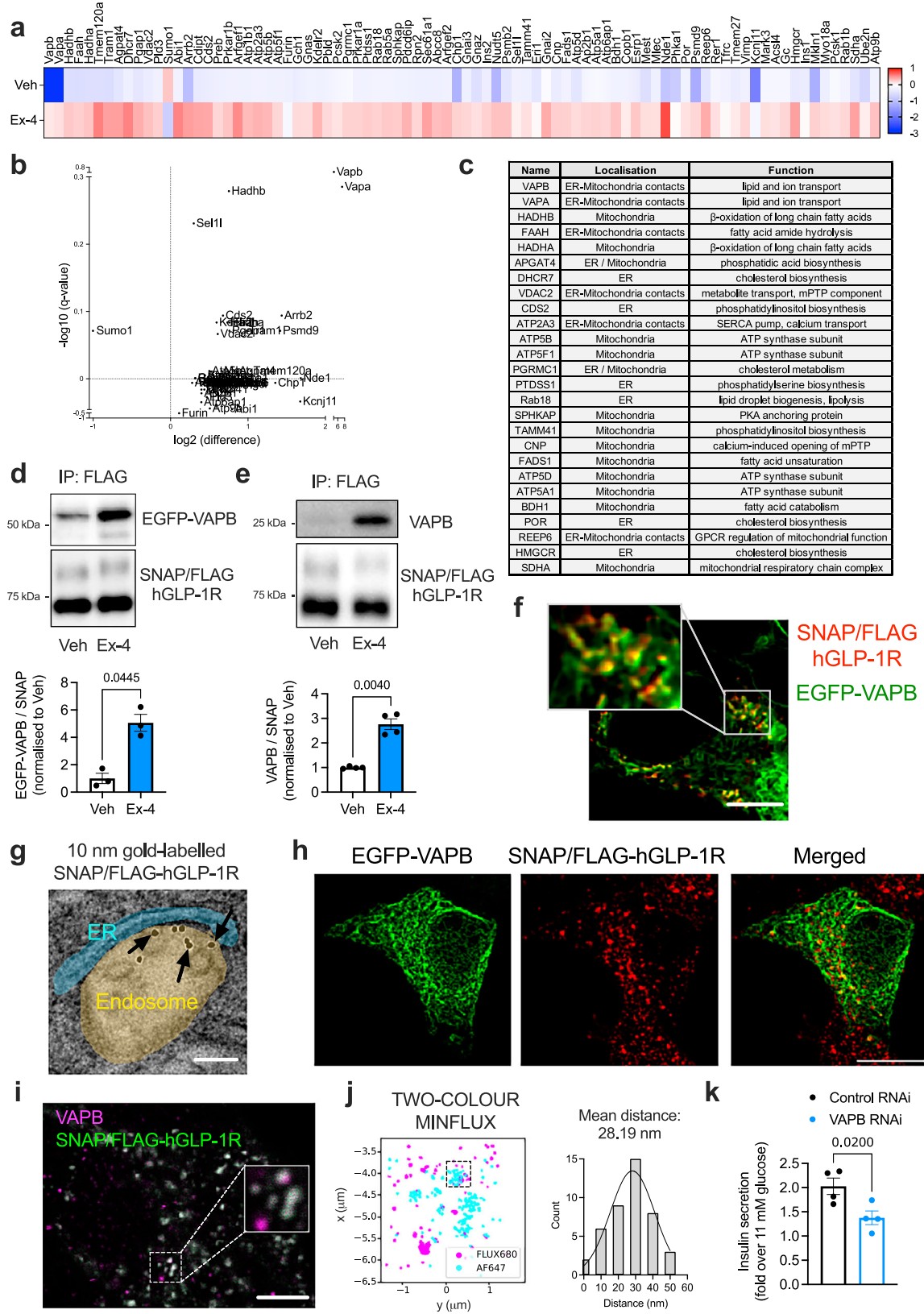

(Fig. 2c). As previously shown, acute (5-minute) exposure to exendin-4 and exendin-D3 triggered noticeable GLP-1R internalisation, while this was much reduced with the biased agonist exendin-F1. Additionally, and as previously published[24], semaglutide triggered a much more robust GLP-1R internalisation than tirzepatide, an agonist that shows a similar degree of GLP-1R Gα$_S$ bias as exendin-F1. We also detected a clear difference in GLP-1R internalisation triggered by orforglipron versus danuglipron, with the latter showing a similar level of receptor internalisation to that of exendin-4, while the former resulted in reduced receptor internalisation 5 minutes post-agonist exposure. VAPB co-immunoprecipitation using the panel of GLP-1RAs described above unveiled a correlation between the level of GLP-1R−VAPB

**Fig. 1 | Human GLP-1R β-cell interactome and validation of VAPB-hGLP-1R interaction. a** Heatmap of GLP-1R interactor enrichment from anti-FLAG co-immunoprecipitates of vehicle (Veh) *versus* 5-minute exendin-4 (Ex-4)-stimulated INS-1 832/3 SNAP/FLAG-hGLP-1R cells; blue, decreased; red, increased binding; LC-MS/MS data analysed in LFQ-Analyst and normalised to hGLP-1R levels for each experimental repeat; colour scale centred around overall median value of the experiment; *n* = 4 biologically independent experiments. **b** Volcano plot of data from (**a**) depicting log$_2$ fold change to vehicle (*x*-axis) *versus* -log10$_q$value (y-axis). **c** List of ER, mitochondria or ERMCS-localised hGLP-1R interactors enriched in exendin-4-stimulated *versus* vehicle conditions. **d** Co-immunoprecipitation (co-IP) of EGFP-VAPB with SNAP/FLAG-hGLP-1R in Veh *versus* Ex-4-stimulated INS-1 832/3 SNAP/FLAG-hGLP-1R cells. Representative blots and quantification of EGFP-VAPB over SNAP levels shown; *n* = 3 biologically independent experiments, *p* value as indicated by two-tailed paired *t*-test. **e** As for (**d**) for endogenous VAPB; *n* = 4 biologically independent experiments, *p* value as indicated by two-tailed paired *t*-test. **f** Confocal microscopy analysis of SNAP/FLAG-hGLP-1R (red) co-localisation with EGFP-VAPB (green) in Ex-4-stimulated INS-1 832/3 SNAP/FLAG-hGLP-1R cells; box, magnification inset; size bar, 5 μm; image representative of *n* = 3 biologically independent experiments. **g** Electron microscopy micrograph depicting gold-labelled SNAP/FLAG-hGLP-1R (arrows) localised to an endosome–ER MCS in an Ex-4-stimulated INS-1 832/3 SNAP/FLAG-hGLP-1R cell; endosome highlighted in gold and ER in blue; size bar, 100 nm; image representative of *n* = 3 biologically independent experiments. **h** Maximum intensity projection (MIP) of a confocal z-stack of EGFP-VAPB (green) and SNAP/FLAG-hGLP-1R (red) from an Ex-4-stimulated INS-1 832/3 SNAP/FLAG-hGLP-1R cell; size bar, 10 μm; image representative of *n* = 3 biologically independent experiments. **i** Confocal detection of FLUX 680-labelled VAPB (magenta) and SNAP-Surface Alexa Fluor 647-labelled SNAP/FLAG-hGLP-1R (green), in Ex-4-stimulated INS-1 832/3 SNAP/FLAG-hGLP-1R cells, showing crosstalk across detector channels (Cy5 near: 650–685 nm, Cy5 far: 685–720 nm); inset, region selected for MINFLUX imaging; size bar, 5 μm; image representative of *n* = 2 biologically independent experiments. **j** Two-colour MINFLUX analysis of selected region from (**i**) after spectral separation, including histogram of nearest neighbour distance between FLUX 680 and Alexa Fluor 647 clusters within the region indicated by dashed box. **k** Ex-4-induced insulin secretion (fold over 11 mM glucose) in non-targeting Control *versus* VAPB RNAi-treated INS-1 832/3 cells; *n* = 4 biologically independent experiments, *p* value as indicated by two-tailed paired *t*-test. Data are mean ± SEM.

binding and the degree of receptor internalisation elicited by each of the tested GLP-1RAs (Fig. 2d and Supplementary Fig. 2d).

## Exendin-4-stimulated GLP-1R triggers VAPB-localised cAMP generation and PKA activity at the outer mitochondrial membrane

As stated above, besides VAPB, our interactome analysis highlighted several factors enriched for their association with active GLP-1R localised at the ER, mitochondria and/or ERMCSs, a location where VAPB plays a prominent organising role[17], suggesting that ERMCSs could be a potential hub for endosomal GLP-1R signalling. To determine if GLP-1R induces cAMP generation specifically at VAPB locations, we modified a fluorescence resonance energy transfer (FRET)-based system of biosensors targeted to endogenous proteins (FluoSTEPs)[25], so that a FRET cAMP sensor would fully reconstitute only in the presence of the GFP$_{11}$ fragment fused to VAPB, resulting in functional biosensor assembly specifically at VAPB loci (Supplementary Fig. 3a). This modified Fluo-STEP system was used to measure VAPB-localised cAMP generation in response to either exendin-4 or exendin-F1 stimulation (Fig. 2e). Results in WT INS-1 832/3 cells (expressing endogenous levels of GLP-1R) show VAPB-localised cAMP responses with both agonists, which are less pronounced for the slow internalising exendin-F1 compared to exendin-4, demonstrating that GLP-1R actively signals at VAPB loci. Next, we evaluated whether GLP-1R signalling would also trigger a pool of PKA activity at the surface of mitochondria, a process known to be important for the control of mitochondrial function[26]. To this end, we employed either a global (whole cell) or a mitochondrially-restricted PKA biosensor expressed at the outer mitochondrial membrane (OMM)[27] (Supplementary Fig. 3b). Exendin-4 stimulation of WT INS-1 832/3 cells triggered, as expected, a robust global PKA response, which was also evident with the OMM-localised PKA biosensor, demonstrating that GLP-1R stimulation triggers PKA activity at the mitochondrial surface (Supplementary Fig. 3c). Furthermore, utilising the same biosensors as above, we detected reduced OMM-localised, but no change in global, PKA activity with slow internalising exendin-F1 *versus* exendin-4 (Fig. 2f), leading to increased mitochondrial over global PKA signalling propensity with exendin-4 at this acute (5-minute) stimulation time-point (Fig. 2g). This, together with the VAPB-localised cAMP FluoSTEP results from above, indicates that GLP-1R internalisation is important for GLP-1R-induced cAMP generation at VAPB loci and PKA signalling at OMMs, at least acutely.

## Exendin-4-stimulated GLP-1R associates with ERMCSs

To examine the potential docking of GLP-1R-positive endosomes to ERMCSs, we employed a split-GFP-based contact site sensor (SPLICS)[28]

to specifically visualise ERMCSs in conjunction with SNAP/FLAG-hGLP-1R in vehicle *versus* exendin-4-stimulated INS-1 832/3 SNAP/FLAG-hGLP-1R cells (Fig. 3a–c and Supplementary Movie 2). This approach allowed us to observe an intimate association between GLP-1R-positive endosomes and ERMCSs over time, validating our hypothesis. Additional cell labelling with fluorescent MitoTracker prior to exendin-4 exposure showed SNAP/FLAG-hGLP-1R extensively colocalised with EGFP-VAPB in proximity to mitochondria, an organelle with which VAPB is known to interact (Fig. 3d).

Further high-resolution 3D imaging to confirm endosomal GLP-1R localisation to ERMCSs included in situ correlative light and electron microscopy (CLEM) at room temperature to correlate ultrastructural features with positive fluorescent signals for SNAP/FLAG-hGLP-1R, EGFP-VAPB and mitochondria. To this end, INS-1 832/3 SNAP/FLAG-hGLP-1R cells were labelled at the cell surface with a membrane impermeable fluorescent SNAP-tag probe prior to exendin-4 stimulation, with correlation of the confocal slice to the corresponding TEM tomogram (Fig. 3e) revealing two mitochondrial volumes in close contact with EGFP-VAPB-positive ER, itself docked to a vesicle positive for SNAP/FLAG-hGLP-1R, indicative of a GLP-1R-positive endosome (Fig. 3f and Supplementary Movie 3). Similar results were obtained using a cryo-CLEM approach, with a SNAP/FLAG-hGLP-1R endosome docked to the EGFP-VAPB-positive ER network, itself in contact with a nearby mitochondrion, following correlation of cryo-confocal and cryo-FIB-SEM data (Supplementary Fig. 4a–c).

These results were further replicated using an immuno-EM approach in primary mouse islets transduced with a SNAP/FLAG-hGLP-1R-expressing adenovirus by labelling cell surface SNAP/FLAG-hGLP-1R in living islets with membrane-impermeable SNAP-biotin followed by 5 nm gold-conjugated streptavidin prior to exendin-4 stimulation and TEM processing. TEM imaging of ultrathin (70 nm) sections from the resin-embedded islets from above revealed 5 nm gold-positive endosomes (indicative of the presence of SNAP/FLAG-hGLP-1R) forming contacts with the ER, itself contacting nearby mitochondria (Fig. 3g).

## Active GLP-1R interacts with the PKA-RI-recruiting AKAP SPHKAP at ERMCSs via VAPB

Amongst the interactors enriched for active GLP-1R binding in our MS interactome analysis, we identified both PKA-RIα/β and the A-kinase anchoring protein (AKAP) SPHKAP, a paralog of AKAP11 (also known as AKAP220) that binds to two RIα subunits of the PKA holoenzyme[29]. Both SPHKAP and AKAP11 harbour a FFAT (two phenylalanines in an acidic tract) motif[30,31], a specific short protein sequence that binds to the Major Sperm Protein (MSP) domain of VAPs (Supplementary

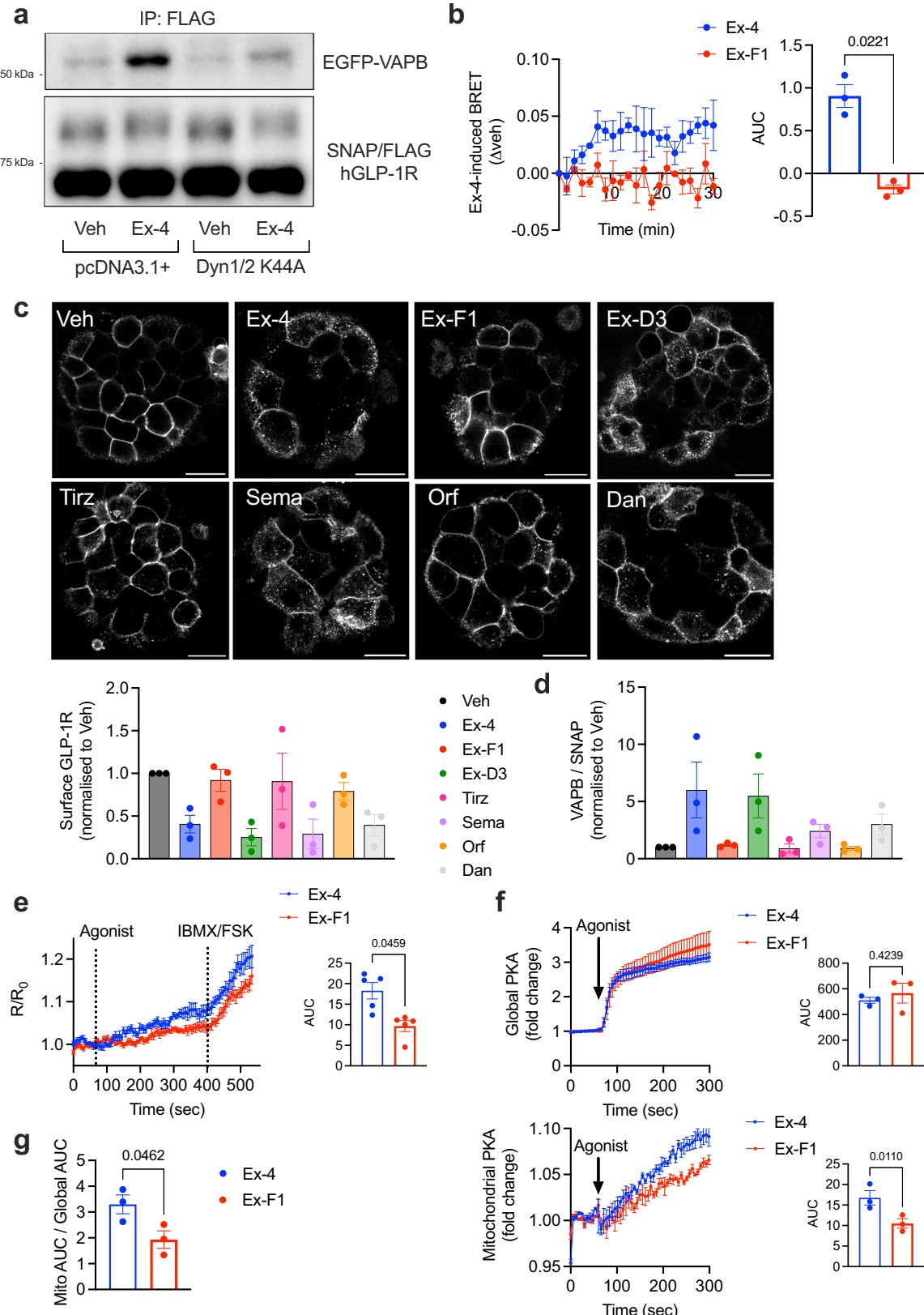

Fig. 5a), which, in the case of SPHKAP, can be regulated by phosphorylation (phosphoFFAT, or pFFAT)[31]. Agreeably, SPHKAP has previously been shown to interact with VAP proteins in neurons and assemble functionally relevant PKA-RI biomolecular condensates at VAPA/B-positive ER locations[32,33]. SPHKAP is expressed in metabolically relevant tissues, including pancreatic islets and intestinal K and L cells,

and has been previously shown to play important roles in both incretin secretion from the gut and incretin-dependent glucoregulation[34,35]. Additionally, we found SPHKAP mRNA expression to be decreased in a published RNA-Seq database of islets from *db/db versus* WT mice (Supplementary Fig. 5b), and meta-analyses from four separate genome-wide association studies (GWAS) in humans uncovered

**Fig. 2 | Internalisation-dependent hGLP-1R-VAPB binding and VAPB-localised GLP-1R signalling. a** EGFP-VAPB co-IP with SNAP/FLAG-hGLP-1R in vehicle (Veh) *versus* exendin-4 (Ex-4)-stimulated INS-1 832/3 SNAP/FLAG-hGLP-1R cells co-expressing control pcDNA3.1+ *versus* dominant negative dynamin (Dyn)1/2 K44A; *n* = 1. **b** Veh-subtracted Ex-4- *versus* exendin-F1 (Ex-F1)-induced BRET kinetic traces (left) and corresponding AUCs (right) of SNAP/FLAG-hGLP-1R-NLuc – Venus-VAPB interactions in INS-1 832/3 cells; *n* = 3 biologically independent experiments, p value as indicated by two-tailed paired *t*-test. **c** (top), Confocal microscopy analysis of SNAP/FLAG-hGLP-1R localisation in mouse primary islets transduced with SNAP/FLAG-hGLP-1R adenoviruses under Veh conditions or following 5-minute stimulation with the indicated agonists: Ex-4, Ex-F1, exendin-D3 (Ex-D3), tirzepatide (Tirz) and semaglutide (Sema) used at 100 nM; orforglipron (Orf) and danuglipron (Dan) used at 5 µM; size bars, 20 µm; **c** (bottom), Quantification of residual surface SNAP/FLAG-hGLP-1R levels in islets from above; *n* = 3 biologically independent experiments. **d** Quantification of VAPB co-IP with SNAP/FLAG-hGLP-1R in INS-1 832/3 SNAP/FLAG-hGLP-1R cells stimulated for 5 minutes with the indicated agonists; *n* = 3 biologically independent experiments. **e** VAPB-localised cAMP generation

(measured by FluoSTEP) in response to 100 nM Ex-4 *versus* Ex-F1 stimulation in INS-1 832/3 cells expressing pcDNA3-GFP11(x7)-VAPB + pcDNA3.1(+)-FluoSTEP-ICUE; maximal responses [to forskolin (FSK) + 3-Isobutyl-1-methylxanthine (IBMX)] shown; kinetic traces (left) and corresponding AUCs for the agonist-stimulated period (right) shown; *n* = 5 biologically independent experiments, *p* value as indicated by two-tailed paired *t*-test. **f** (top), Ex-4 *versus* Ex-F1-induced global PKA activity, measured with the global PKA biosensor pcDNA3.1(+)- ExRai-AKAR2 in INS-1 832/3 cells; kinetic traces (left) and corresponding AUCs (right) shown; *n* = 3 biologically independent experiments; *p* value as indicated by two-tailed paired *t*-test. **f** (bottom), Ex-4 *versus* Ex-F1-induced mitochondrial PKA activity, measured with the mitochondrial PKA biosensor pcDNA3-mitoExRai-AKAR2 in INS-1 832/3 cells; kinetic traces (left) and corresponding AUCs (right) shown; *n* = 3 biologically independent experiments, *p* value as indicated by two-tailed paired *t*-test. **g,** Mitochondrial over global PKA activity, calculated as AUC ratio from data in (**f**); *n* = 3 biologically independent experiments, *p* value as indicated by two-tailed paired *t*-test. Data are mean ± SEM.

genome-wide significant associations between *SPHKAP* gene variants, T2D risk, and high BMI[36–39] (Supplementary Fig. 5c, d). We found the effect of *SPHKAP* gene variants in T2D *versus* high BMI to be tightly correlated (Supplementary Fig. 5e), suggesting a shared mechanism of action for SPHKAP in both disorders where GLP-1RAs are also known to be effective. Our own genetic analysis also demonstrated a significant association between *SPHKAP* gene coding variants and random blood glucose levels in whole exome sequencing (WES) data from UK Biobank (UKBB) individuals without diabetes (Supplementary Fig. 5f), strongly supporting a role for SPHKAP in the regulation of β-cell function.

We therefore next investigated the potential involvement of SPHKAP/PKA-RI in GLP-1R signal transduction from ERMCSs. We first confirmed active GLP−1R−SPHKAP protein-protein interaction, inferred from our MS interactome analysis, by measuring increased co-immunoprecipitation of SPHKAP-EGFP (Fig. 4a, b, Control RNAi conditions; Supplementary Fig. 6a), as well as of endogenous SPHKAP (Supplementary Fig. 6b), with SNAP/FLAG-hGLP-1R in exendin-4-stimulated *versus* vehicle-treated INS-1 832/3 SNAP/FLAG-hGLP-1R cells. Interestingly, we also detected a similar association between the other incretin receptor, glucose-dependent insulinotropic polypeptide receptor (GIPR), and endogenous SPHKAP following GIP stimulation of INS-1 832/3 cells stably expressing SNAP/FLAG-hGIPR (Supplementary Fig. 6c), suggesting that this localised PKA signalling module is shared between both incretin receptors.

We next determined that the exendin-4-induced GLP-1R−SPHKAP interaction requires VAPB, as it is abrogated by VAPB knockdown (Fig. 4a, b, VAPB RNAi conditions; Supplementary Fig. 6a). Confocal analysis of SPHKAP-EGFP localisation in WT INS-1 832/3 cells revealed that, in contrast with a previous report suggesting mitochondrial localisation[40], SPHKAP localises to an intracellular network occasionally in contact with, but not inside, mitochondria (Fig. 4c, Control RNAi panels; Supplementary Fig. 6d). Indeed, labelling of ER with the fluorescent dye ER-Tracker revealed the punctate localisation of SPHKAP-EGFP along ER membranes (Fig. 4d, e), in agreement with recent reports which also indicate SPHKAP localisation and PKA-RI targeting to the ER[32,33]. Also in agreement with these reports, the localisation of SPHKAP-EGFP to ER loci required VAPB, as it was lost in VAPB RNAi-treated cells (Fig. 4c, VAPB RNAi panels). We next generated a mutant version of SPHKAP-EGFP harbouring a point mutation in its pFFAT motif (SPHKAP-EGFP A215E, DFLT**A**SE to DFLT**E**SE). This strategy was based on previous structural data showing that position 5 of the FFAT core motif is located within a hydrophobic pocket in the MSP domain of VAP[41], with previous studies indicating that introducing a negative charge in this position causes a steric hindrance which effectively disrupts FFAT motif−VAP interaction[42]. Agreeably, our SPHKAP-EGFP pFFAT mutant lost both its binding to exendin-4-

stimulated SNAP/FLAG-hGLP-1R (Supplementary Fig. 6e) and its localisation to the ER network (Supplementary Fig. 6f). We additionally demonstrated the co-existence of SNAP/FLAG-hGLP-1R, VAPB and SPHKAP in mitochondria-associated membranes (MAMs) purified from exendin-4-stimulated INS-1 832/3 SNAP/FLAG-hGLP-1R cells (Supplementary Fig. 6g), and a protein-protein interaction between SPHKAP and EGFP-VAPB in the same cells (Supplementary Fig. 6h). Taken as a whole, these results demonstrate the association of SPHKAP with active GLP-1R at ERMCSs in a VAPB-dependent manner via the pFFAT motif of SPHKAP.

## ERMCS-localised GLP-1R-dependent PKA signalling requires SPHKAP

We next tested the effect of SPHKAP downregulation on GLP-1R-dependent downstream signalling. SPHKAP knockdown efficiency in INS-1 832/3 cells was validated against both endogenous SPHKAP (Supplementary Fig. 6i) and SPHKAP-EGFP (Supplementary Fig. 6j). Loss of SPHKAP did not affect the interaction between the receptor and VAPB (Supplementary Fig. 7a-c), indicating that GLP-1R is recruited to VAPB-positive ERMCSs independently of SPHKAP. Loss of SPHKAP also did not affect the capacity for the receptor to generate cAMP following exendin-4 stimulation of WT INS-1 832/3 cells (Fig. 4f). In contrast, while global PKA responses to exendin-4 were not affected (Fig. 4g), exendin-4-induced OMM-localised PKA activity was blunted by SPHKAP downregulation (Fig. 4h). Finally, we employed a similar modified FluoSTEP system to the one shown in Fig. 2, but this time adapted to assess VAPB-localised PKA activity in response to exendin-4 stimulation in Control *versus* SPHKAP RNAi-treated WT INS-1 832/3 cells (see Supplementary Fig. 7d for a schematic of this method). Results showed that exendin-4 stimulation triggers VAPB-localised PKA activity in WT INS-1 832/3 cells, and that this activity requires SPHKAP (Fig. 4i), presumably via SPHKAP binding to PKA-RI and assembly of a PKA signalling hub at ERMCSs. Consequently, we could observe the appearance of PKA-RIα-EGFP-positive puncta, indicative of liquid-liquid phase separation (LLPS) and biomolecular condensate assembly[25], in exendin-4-stimulated WT INS-1 832/3 cells (Supplementary Fig. 7e).

## Control of mitochondrial and β-cell function by SPHKAP-dependent GLP-1R signalling

We next assessed the effect of ERMCS-localised GLP-1R-induced PKA signalling in the control of mitochondrial function in INS-1 832/3 cells. Exendin-4 stimulation triggered a small increase in mitochondrial membrane potential under 6 mM glucose conditions, with this response further enhanced under increased (20 mM) glucose concentrations. Both membrane potential rises were significantly reduced in SPHKAP RNAi-treated *versus* Control WT INS-1 832/3 cells (Fig. 5a).

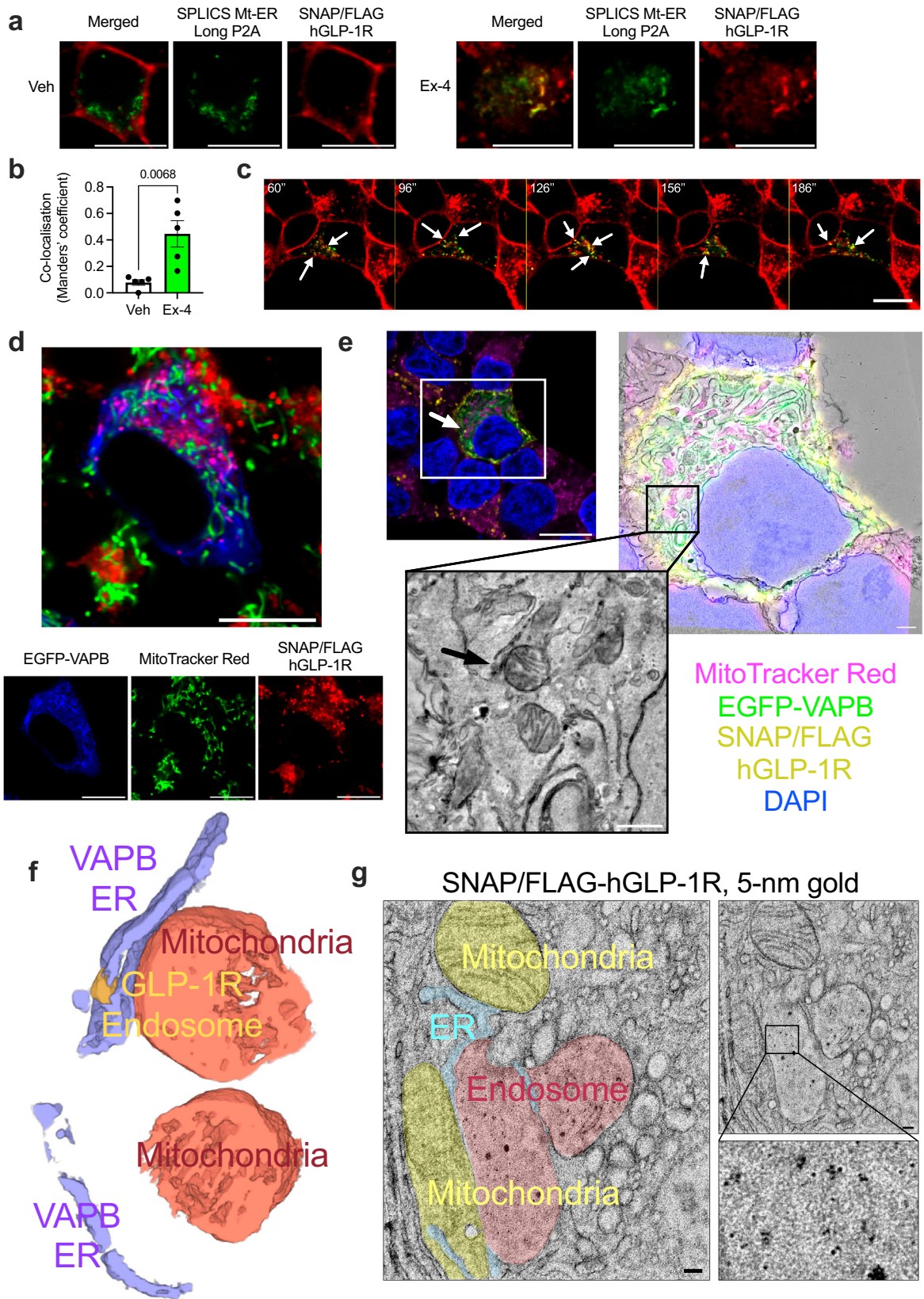

Accordingly, we observed a loss of ATP tone in exendin-4-stimulated INS-1 832/3 cells following SPHKAP knockdown (Fig. 5b and Supplementary Fig. 7f), suggesting a defect in mitochondrial respiration. We further determined the impact of ERMCS-localised GLP-1R signalling on β-cell function by analysing acute insulin secretion responses and β-cell survival to ER stress downstream of GLP-1R stimulation in SPHKAP

RNAi-treated *versus* Control WT INS-1 832/3 cells. Exendin-4-stimulated insulin secretion was significantly reduced in SPHKAP RNAi *versus* control conditions (Fig. 5c and Supplementary Fig. 7g), without any effect of SPHKAP over-expression in this parameter (Supplementary Fig. 7h). Additionally, the anti-apoptotic effect of exendin-4 in cells exposed to thapsigargin (to trigger ER stress) was blunted following

**Fig. 3 | Human GLP-1R localisation to ERMCSs. a** Confocal microscopy analysis of SNAP/FLAG-hGLP-1R (red) co-localisation with ERMCSs (SPLICS Mt-ER Long P2A, green) in vehicle (Veh, left) *versus* exendin-4 (Ex-4, right)-stimulated INS-1 832/3 SNAP/FLAG-hGLP-1R cells; size bars, 10 μm; images representative of *n* = 3 biologically independent experiments. **b** SNAP/FLAG-hGLP-1R - SPLICS Mt-ER Long P2A co-localisation (Mander's coefficient) in Veh *versus* Ex-4-stimulated conditions from data in (**a**); *n* = 5 cells from 3 independent experiments, p value as indicated by two-tailed unpaired *t*-test. **c** Time-lapse confocal microscopy analysis of SNAP/FLAG-hGLP-1R–ERMCS association, imaged as in (**a**); arrows indicate co-localised spots; time post-Ex-4 addition indicated; size bar, 10 μm; images representative of *n* = 3 biologically independent experiments. **d** Confocal microscopy analysis of EGFP-VAPB (blue), mitochondria (MitoTracker Red, green) and SNAP/FLAG-hGLP-1R (SNAP-Surface Alexa Fluor 647, red) in an Ex-4-stimulated INS-1 832/3 SNAP/FLAG-hGLP-1R cell; size bar, 10 μm; images representative of *n* = 3 biologically independent experiments. **e** Single confocal slice from an Ex-4-stimulated INS-1 832/3 SNAP/FLAG-hGLP-1R cell, with signals for EGFP-VAPB (green), SNAP/FLAG-hGLP-1R (yellow), mitochondria (magenta), and DAPI (blue) shown; white arrow points to a SNAP/FLAG-hGLP-1R-positive endosome used for CLEM analysis; size bar, 10 μm; also included CLEM overlay of aligned confocal slice with corresponding single slice of an EM tomogram (size bar, 10 μm) and magnification of EM tomogram slices (size bar, 1 μm) with indicated SNAP/FLAG-hGLP-1R-positive endosome (black arrow); images representative of *n* = 2 cell regions from the same biological experiment. **f** 3D rendering of segmentation of data from (**e**), with VAPB-positive ER (purple), mitochondria (orange) and SNAP/FLAG-hGLP-1R-positive endosome (yellow) shown; Created with ORS Dragonfly. **g** TEM micrograph of a 70 nm ultra-thin section from resin embedded mouse islets transduced with SNAP/FLAG-hGLP-1R adenovirus, with SNAP/FLAG-hGLP-1R labelled with SNAP-Surface biotin + Alexa Fluor 488 Streptavidin, 5 nm colloidal gold prior to Ex-4 stimulation and fixation; ER (blue), mitochondria (yellow) and SNAP/FLAG-hGLP-1R-positive endosome (pink); inset shows magnified endosome area containing 5 nm gold particles; size bars, 100 nm; images representative of *n* = 3 biologically independent experiments. Data are mean ± SEM.

SPHKAP knockdown (Fig. 5d). Finally, we measured exendin-4-induced mitochondrial and ER Ca²⁺ responses in Control *versus* SPHKAP RNAi-treated WT INS-1 832/3 cells. Increases in both ER and mitochondrial Ca²⁺ in response to exendin-4 stimulation were apparent in these cells. However, while ER responses were not affected by SPHKAP knockdown, we detected a delay in the mitochondrial Ca²⁺ response peak time in SPHKAP RNAi *versus* Control RNAi-treated WT INS-1 832/3 cells (Supplementary Fig. 7i, j).

As SPHKAP has previously been shown to bind to the mitochondrial cristae junction regulator and MICOS complex component MIC19[40], a prominent PKA substrate involved in both cristae assembly during mitochondrial remodelling[43] and in regulation of ERMCSs, we next tested whether exendin-4 stimulation would trigger MIC19 phosphorylation in WT INS-1 832/3 cells using a MIC19 immunoprecipitation strategy coupled with western blotting with an antibody against PKA phosphosites (RRxpS/T)[44]. Exendin-4 exposure resulted in a significant increase in PKA-specific MIC19 phosphorylation, an effect that was potentiated by the adenylate cyclase activator forskolin (Fig. 5e). Dual knockdown of SPHKAP and VAPB in INS-1 832/3 cells led to widespread changes in PKA phosphorylation of several MICOS complex components downstream of GLP-1R activation, suggesting that ERMCS-localised GLP-1R signalling modulates the activity of the MICOS complex (Supplementary Fig. 7k).

## Control of mitochondrial morphology by SPHKAP-dependent ERMCS-localised GLP-1R signalling

As the main role of the MICOS complex is to orchestrate mitochondrial cristae assembly[45], and MIC19 plays an important role in the transfer of lipid precursors across ERMCSs during mitochondrial remodelling[10], we next assessed whether GLP-1R stimulation would play a role in mitochondrial remodelling processes in INS-1 832/3 cells, and whether this would be affected by SPHKAP knockdown. We first investigated if acute exendin-4 stimulation triggers changes in mitochondrial turnover. Despite no detectable changes in mitochondrial DNA copy number after 30 minutes of exendin-4 exposure in Control *versus* SPHKAP RNAi-treated, or in GLP-1R KO INS-1 832/3 cells (Supplementary Fig. 7l), acute (10-minute) stimulation of WT INS-1 832/3 cells with exendin-4 triggered increased mitophagy, assessed using an mKeima-based assay, an effect not present following SPHKAP downregulation (Fig. 5f). Further evidence of exendin-4-induced mitophagy was obtained by CLEM analysis of INS-1 832/3 SNAP/FLAG-hGLP-1R cells, where we could find instances of SNAP/FLAG-hGLP-1R-positive endolysosomes surrounded by mitochondria (Supplementary Fig. 8a,b). We also detected increased recruitment of the mitochondrial fission protein mCherry-Drp1 to mito-BFP-expressing mitochondria in exendin-4-stimulated *versus* vehicle-exposed WT INS-1 832/3 cells (Fig. 5g, Supplementary Fig. 8c, Supplementary Movies 4 and 5). Drp1 phosphorylation at Serine 616, which, unlike Serine 637[46], is important for

regulating mitochondrial fission and is not directly PKA-dependent[47,48], was, as expected, not modified by acute exendin-4 exposure in Control RNAi-treated cells, but was progressively lost 15 minutes post-exendin-4 stimulation in SPHKAP RNAi-treated WT INS-1 832/3 cells (Fig. 5h).

We next directly assessed changes in mitochondrial morphology by confocal microscopy imaging of MitoTracker-labelled mitochondria in vehicle *versus* exendin-4-stimulated WT INS-1 832/3 cells. Here, we employed a similar approach to[49] to manually score cells by mitochondrial phenotype into three categories: long tubular (referred to as "hyperfused"), intermediate/short tubular (referred to as "tubular") and fragmented. Results showed that 10 minutes of exendin-4 exposure completely remodels the mitochondrial network from a predominantly hyperfused to a fragmented phenotype (Fig. 6a). As expected, and serving as validation of our classification approach, this remodelling did not take place in INS-1 832/3 cells lacking GLP-1R (GLP-1R KO) (Fig. 6b). We further confirmed our results by re-analysing the data from above using the automated Mitochondria Analyzer plugin in Fiji to determine changes in mitochondrial form factor, which provides information on mitochondrial shape, as well as mitochondrial mean area (Supplementary Fig. 8d). Agreeably, we found a significant reduction on mitochondrial form factor and mean area, indicative of remodelling, upon exendin-4 stimulation of WT INS-1 832/3, but not of INS-1 832/3 GLP-1R KO cells.

Next, we performed mitochondrial morphology analysis of WT INS-1 832/3 cells treated with Control *versus* SPHKAP RNAi prior to vehicle or exendin-4 exposure (Fig. 6c). While exendin-4 triggered a similar degree of mitochondrial network remodelling as above in Control RNAi-treated cells, SPHKAP knockdown resulted in disrupted mitochondrial morphology at vehicle conditions, which somewhat resembled that of exendin-4-stimulated Control RNAi-treated cells in terms of tubularity but had further visual abnormalities. Moreover, exendin-4 stimulation had no further effect on the mitochondrial morphology of SPHKAP RNAi-treated cells. These results were validated with the Mitochondria Analyzer plugin in Fiji, which showed reduced mitochondrial form factor and mean area following exendin-4 exposure of Control RNAi-treated cells, with SPHKAP RNAi-treated cells under vehicle conditions showing similar values to those of exendin-4-stimulated control cells (Supplementary Fig. 8e). However, with this analysis method, we also detected an apparent increase in form factor and mitochondrial mean area in exendin-4-stimulated *versus* vehicle-treated SPHKAP RNAi-treated cells, an effect opposite to that triggered by exendin-4 in control conditions.

To better visualise the changes in mitochondrial morphology under these different conditions, we performed z-series analysis of confocal sections across whole cells, represented as MIPs in Fig. 6d, as well as STED microscopy analysis of mitochondrial morphologies using the fixable STED-compatible mitochondrial dye PKmito ORANGE FX (Fig. 6e), with Control *versus* SPHKAP RNAi-treated cells showing

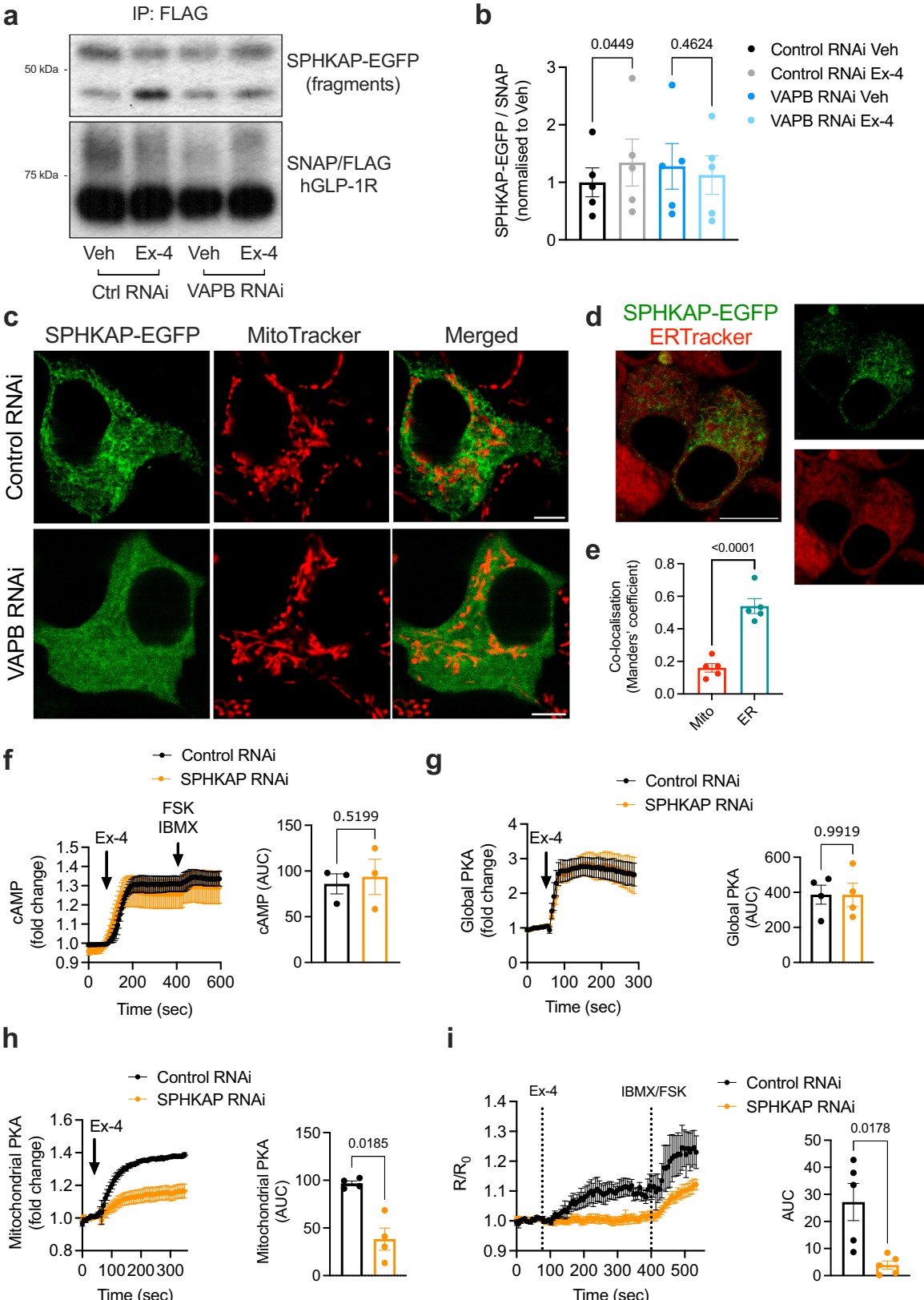

similar changes in mitochondrial morphology as those described above.

Finally, we performed high-resolution TEM imaging for precise quantification of mitochondrial area and number of cristae per mitochondria in ultrathin (70 nm) sections on resin-embedded Control *versus* SPHKAP RNAi-treated WT INS-1 832/3 cells, as well as INS-1 832/3

GLP-1R KO cells, both under vehicle conditions and following exendin-4 stimulation. In agreement with our previous results, we measured an exendin-4-induced reduction in both parameters in Control RNAi-treated cells, but no change in response to exendin-4 in either SPHKAP RNAi-treated or in GLP-1R KO cells (Fig. 7a, b). Interestingly, the mitochondrial morphology changes triggered by exendin-4 in Control

**Fig. 4 | Human GLP-1R–SPHKAP association via shared VAPB binding and SPHKAP-dependent GLP-1R-induced VAPB-localised PKA activity. a** SPHKAP-EGFP co-IP with SNAP/FLAG-hGLP-1R in Control *versus* VAPB RNAi-treated INS-1 832/3 SNAP/FLAG-hGLP-1R cells under vehicle (Veh) *versus* exendin-4 (Ex-4)-stimulated conditions; blots representative of *n* = 5 biologically independent experiments; GFP-positive fragments detected due to intrinsic instability of co-IPed full-length fusion protein. **b** Quantification of SPHKAP-EGFP over SNAP/FLAG-hGLP-1R levels from (**a**); *n* = 5 biologically independent experiments, p values as indicated by one-way ANOVA with Šidák post-hoc test. **c** Confocal microscopy analysis of SPHKAP-EGFP *versus* mitochondria (imaged with MitoTracker Red) localisation in INS-1 832/3 cells treated with Control *versus* VAPB RNAi; size bars, 5 μm; images representative of *n* = 3 biologically independent experiments. **d** Confocal microscopy analysis of SPHKAP-EGFP *versus* ER (imaged with ER-Tracker Red) localisation in INS-1 832/3 cells; size bar, 10 μm; images representative of *n* = 3 biologically independent experiments. **e** SPHKAP-EGFP co-localisation (Mander's coefficient) with mitochondria (Mito) *versus* ER; *n* = 5 cells from 3 independent experiments,

p value as indicated by two-tailed unpaired *t*-test. **f** Ex-4-stimulated cAMP responses in Control *versus* SPHKAP RNAi-treated INS-1 832/3 cells, measured with cADDis cAMP biosensor; left, cAMP traces; right, AUC quantification; *n* = 3 biologically independent experiments; p value as indicated by two-tailed paired *t*-test. **g** Ex-4-stimulated global PKA responses in Control *versus* SPHKAP RNAi-treated INS-1 832/3 cells; left, PKA traces; right, AUC quantification; *n* = 4 biologically independent experiments; *p* value as indicated by two-tailed paired *t*-test. **h** Ex-4-stimulated mitochondrial PKA responses in Control *versus* SPHKAP RNAi-treated INS-1 832/3 cells; left, PKA traces; right, AUC quantification; *n* = 4 biologically independent experiments; p value as indicated by two-tailed paired *t*-test. **i**, VAPB-localised PKA activity (measured by FluoSTEP) in response to 100 nM Ex-4 stimulation in Control *versus* SPHKAP RNAi-treated INS-1 832/3 cells expressing pcDNA3-GFP11(x7)-VAPB + pcDNA3.1(+)-FluoSTEP-AKAR; maximal responses (to FSK + IBMX) shown; kinetic traces (left) and corresponding AUCs for the agonist-stimulated period (right) shown; *n* = 5 biologically independent experiments; *p* value as indicated by two-tailed paired *t*-test. Data are mean ± SEM.

RNAi-treated cells were accompanied by a significant reduction in ERMCS length, which appeared already reduced under vehicle conditions without further effect of exendin-4 following SPHKAP down-regulation or in GLP-1R KO cells, with no apparent effect of SPHKAP-EGFP overexpression in INS-1 832/3 cells in this parameter (Fig. 7c). This ERMCS length reduction was accompanied by a reduced number of ERMCSs per mitochondrial area in exendin-4-stimulated *versus* vehicle-exposed WT INS-1 832/3 cells, assessed using a splitFAST approach as in ref. 50 (Fig. 7d, e).

### GLP-1R localisation and GLP-1R-induced PKA signalling at ERMCSs in primary mouse and human islets

To validate the interaction of GLP-1R with VAPB and SPHKAP at ERMCSs in a physiologically relevant setting with endogenous levels of receptor expression, proximity ligation assays (PLAs) were performed in purified mouse primary islets using one of the few existing validated mouse GLP-1R antibodies from the Developmental Studies Hybridoma Bank[51–53], with GLP-1R–VAPB and GLP-1R–SPHKAP interactions readily observed in exendin-4-stimulated islets (Fig. 8a, Supplementary Fig. 9a). To verify that these interactions are present within pancreatic β-cells, we counterstained both vehicle and exendin-4-stimulated islets for insulin (Supplementary Fig. 9b). Next, to analyse the functional impact of SPHKAP downregulation on islet function, we employed a lentiviral shRNA approach to knock down SPHKAP on intact islets, as most functional outputs from primary islets cannot be optimally recapitulated with dispersed islet cells due to the loss of important intra-islet cell-cell interactions. Lentiviral transduction of cells within intact islets is restricted to the outermost islet cell layers so that only a small fraction of cells (which we estimate at ~25% of the total) is effectively transduced. However, those represent a higher functional cell pool due to necrosis of the islet core in non-vascularised isolated islets[54], so that it is still possible to assess potential functional differences from a relatively small percentage of transduced cells. Accordingly, we could only detect a tendency for reduced *SPHKAP* mRNA levels in whole islets transduced with SPHKAP *versus* Control shRNA lentiviruses (Supplementary Fig. 9c), which nevertheless suggests a good level of downregulation from the pool of transduced cells. We next analysed some key signalling outcomes from these islets. As for INS-1 832/3 cells, ATP levels in exendin-4-stimulated *versus* vehicle-treated islets were reduced with SPHKAP shRNA (Fig. 8b). Exendin-4 exposure also led to a significant reduction in the level of reactive oxygen species (ROS) in Control shRNA islets exposed to thapsigargin, but this effect was no longer present in SPHKAP shRNA-transduced islets, which presented with basally higher ROS levels than the corresponding control islets (Fig. 8c). Finally, while the effect of SPHKAP knockdown in the acute potentiation of insulin secretion triggered by exendin-4 in these islets was mild, we found a small but significant increase in insulin secretion at 11 mM glucose alone in SPHKAP *versus*

Control shRNA-transduced islets (Fig. 8d, left), which led to a near-significant reduction in exendin-4-induced insulin secretion fold (Fig. 8d, right), an effect which agrees with previously published results in a SPHKAP KO mouse model, with increased insulin secretion under glucose alone conditions but dampened incretin responses in islets from these mice[34].

We next assessed changes in mitochondrial morphology in intact mouse islets by confocal imaging of mitochondria from the outermost layer of islet cells, to maximise our chances of recording phenotypes from lentivirally transduced cells. We used a similar approach as in INS-1 832/3 cells to segment and classify islet cells as predominantly presenting a hyperfused, tubular or fragmented mitochondrial phenotype (see Supplementary Fig. 9d for an example of our method of islet segmentation and cell classification). Cells in WT mouse islets presented a similar distribution of mitochondrial morphologies as in INS-1 832/3 cells, with mitochondria primarily hyperfused or tubular under vehicle conditions but remodelling towards a more fragmented phenotype following exendin-4 exposure (Fig. 9a). Islets extracted from GLP-1R KO mice presented a higher level of hyperfused mitochondria that did not undergo remodelling with exendin-4 (Fig. 9b). As in INS-1 832/3 cells, transduction of WT islets with SPHKAP shRNA-expressing lentiviruses triggered a change in mitochondrial morphology under vehicle conditions, with a higher degree of fragmentation *versus* control islets and no effect of exendin-4 exposure following SPHKAP downregulation (Fig. 9c).

We also performed mitochondrial morphology assessments in dispersed islet cells from mice transfected with Control *versus* SPHKAP siRNA, which responded similarly to intact islets in terms of increased mitochondrial fragmentation following exendin-4 exposure in control conditions, but showed disturbed mitochondrial morphology under vehicle conditions and no further effect of exendin-4 exposure in SPHKAP siRNA-treated cells (Fig. 9d). These phenotypic changes were still present when assessed as changes in mitochondrial form factor and mean area using the Mitochondria Analyser Fiji plugin (Supplementary Fig. 9e).

Lastly, we repeated some of our assessments with human donor islets. PLA assays again demonstrated interactions between endogenous GLP-1R, VAPB and SPHKAP in exendin-4-stimulated human islets (Fig. 10a, b). As in mouse islets, assessment of mitochondrial morphology by confocal microscopy analysis in intact human islets revealed predominantly hyperfused mitochondria under vehicle conditions, which was rapidly remodelled by acute exendin-4 exposure to a more fragmented morphology (Fig. 10c, d).

### Discussion

Mitochondrial malfunction is a prominent feature of diabetes, encompassing defects in mitochondrial structure and dynamics, ROS overproduction and apoptosis, breakdown of nutrient sensing and

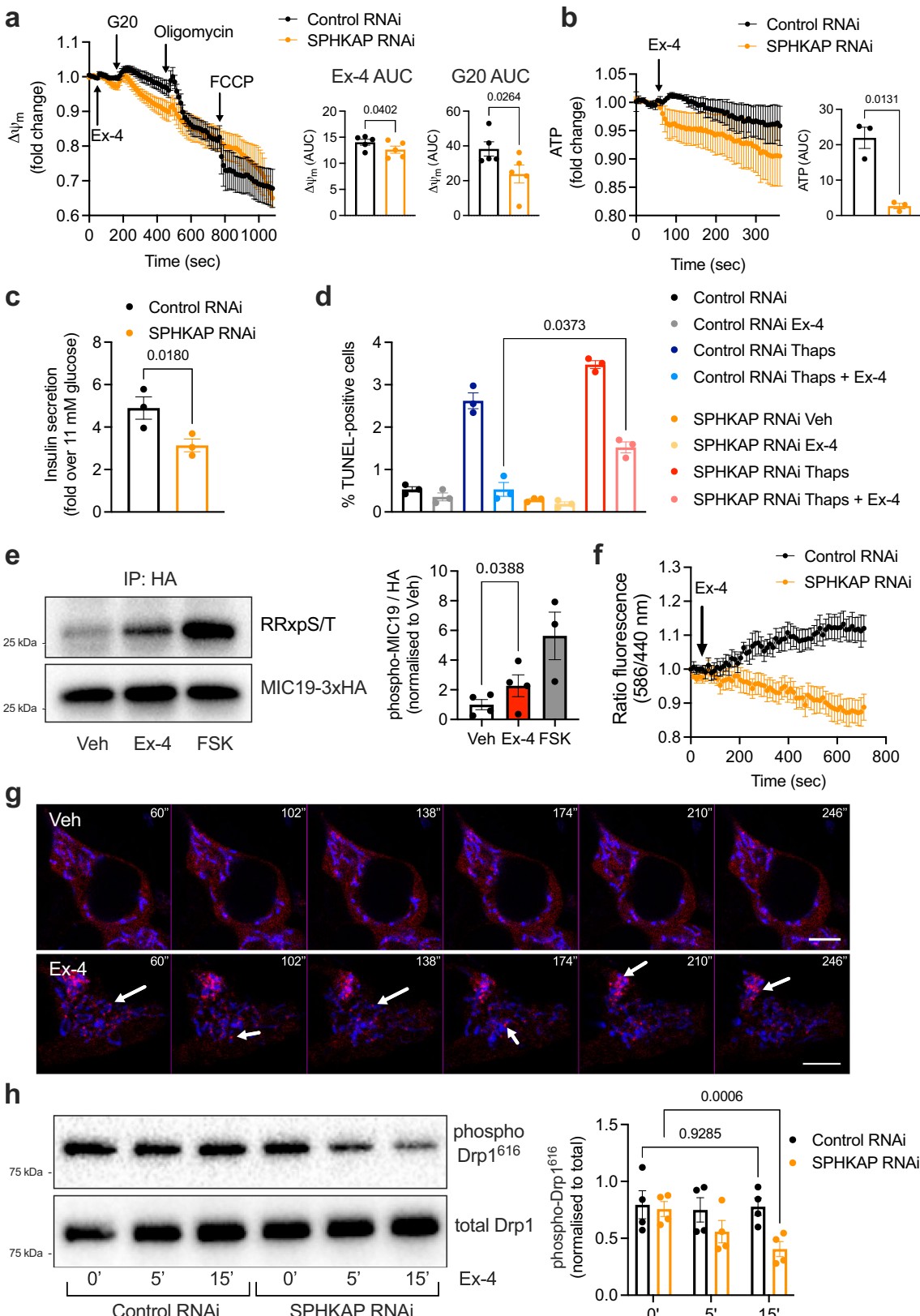

metabolic uncoupling[55,56], with strong backing evidence from human genetic studies[57–59]. Mitochondrial quality control mechanisms have been recently found to control cellular identity and maturity in metabolically relevant tissues[60], including in pancreatic β-cells, highlighting the importance that mitochondria have in the control of β-cell function. Strategies to correct the detrimental effect of T2D in affected

tissues are therefore likely to involve amelioration of mitochondrial function; GLP-1RAs are no exception, with numerous reports highlighting their capacity to potentiate mitochondrial quality control mechanisms and improve mitochondrial health[5–8], with mitochondrial regulation potentially key to explain their beneficial effects in metabolic tissues.

**Fig. 5 | Regulation of GLP-1R function by SPHKAP. a** Exendin-4 (Ex-4)-stimulated changes in mitochondrial membrane potential in Control *versus* SPHKAP RNAi-treated INS-1 832/3 cells, measured with TMRE; left, membrane potential traces; G20, 20 mM glucose; right, quantification of AUCs during the Ex-4 and G20 periods; $n = 5$ biologically independent experiments, p values as indicated by two-tailed paired *t*-test. **b** Ex-4-stimulated ATP level in Control *versus* SPHKAP RNAi-treated INS-1 832/3 cells, measured with the cyto-Ruby3-iATPSnFR1.0 biosensor; left, ATP traces; right, AUC quantification; $n = 3$ biologically independent experiments, p value as indicated by two-tailed paired *t*-test. **c** Ex-4-induced insulin secretion (fold over 11 mM glucose) in Control *versus* SPHKAP RNAi-treated INS-1 832/3 cells; $n = 3$ biologically independent experiments, p value as indicated by two-tailed paired *t*-test. **d** Percentage of apoptosis ± thapsigargin exposure in Control *versus* SPHKAP RNAi-treated INS-1 832/3 cells under vehicle (Veh) *versus* Ex-4-stimulated conditions; $n = 3$ biologically independent experiments; *p* value as indicated by one-way ANOVA with Šidák post-hoc test. **e** PKA-dependent phosphorylation of MIC19-3xHA in INS-1 832/3 cells under Veh *versus* Ex-4-stimulated conditions, determined by western blotting of anti-HA immunoprecipitates with an anti-RRx-pS/T antibody; FSK, positive control for PKA activation; left, representative blots; right, quantification of phospho-MIC19 levels; $n = 3$ to 4 biologically independent experiments, p value as indicated by one-way ANOVA with Šidák post-hoc test. **f** Ex-4-induced mitophagy, measured as changes in ratio fluorescence intensities (580/440 nm) over time with the dual-excitation ratiometric biosensor mKeima-Red-Mito-7 in Control *versus* SPHKAP RNAi-treated INS-1 832/3 cells; $n = 6$ biologically independent experiments. **g** Time-lapse confocal microscopy imaging of Drp1 localisation to mitochondria in Veh *versus* Ex-4-stimulated INS-1 832/3 cells; mitochondria (mito-BFP, blue) and Drp1 (mCherry-Drp1, red); arrows indicate Drp1-positive spots on mitochondria; time post-Veh or Ex-4 exposure indicated; size bars, 5 μm; images representative of $n = 3$ biologically independent experiments. **h**, Changes in Drp1 phosphorylation at Ser[616] following Ex-4 stimulation of Control *versus* SPHKAP RNAi-treated INS-1 832/3 cells; left, representative blots; right, quantification; $n = 4$ biologically independent experiments; *p* values as indicated by two-way ANOVA with Tukey post-hoc test. Data are mean ± SEM.

Despite the accumulating evidence around pathways engaged by the GLP-1R and other GPCRs in target tissues, there is still substantial uncertainty about the precise molecular mechanisms involved, particularly with regards to the spatiotemporal organisation of their signalling and their presence in spatially restricted subcellular (or even suborganelle) signalling hubs or nanodomains[61]. In this context, the study of MCSs has become an area of considerable importance to understand the spatial architecture of signalling, with recent reports recognising their prominent role in pancreatic β-cells for key processes ranging from insulin granule biogenesis to calcium homoeostasis[62–64]. In this study, we took an unbiased approach to unveil molecular partners working in concert with the GLP-1R to control β-cell function, finding not only that the receptor engages with ER-endosome MCSs to organise its signalling, but that the main constituents of these inter-organelle contacts, namely the VAPs, are by a considerable margin the most prominent intracellular binding partners of active GLP-1R. This suggests that engagement of VAPs and subsequent generation of ER-localised GLP-1R signalling is not just one of many pathways employed by the receptor, but the main site of action of endosomal GLP-1R following its internalisation from the plasma membrane.

This also suggests that GLP-1RAs with distinct rates of receptor internalisation and lysosomal targeting might differentially regulate the kinetics of this engagement, so that slow GLP-1R internalising compounds are expected to trigger initially reduced VAPB interactions, which might however paradoxically result in prolonged VAPB engagement over time, and therefore potentially lead to sustained ER signalling, as these biased compounds are also associated with reduced lysosomal degradation from the pool of internalised receptors[18,24]. Of note, we have found here a similar protein-protein interaction between SPHKAP and GIP-stimulated GIPR, suggesting that compounds such as tirzepatide, which target the GLP-1R as a biased, but the GIPR as a balanced agonist, might present with an optimal signalling profile by means of maximising prolonged plasma membrane signalling from the GLP-1R while at the same time engaging in GIPR-dependent acute ER signalling.

In the present study, we have not only unveiled the interaction of endosomal GLP-1R with the ER, but also its localisation to VAPB-containing ERMCSs, from where the receptor assembles a newly described PKA signalling nanodomain, enabling cAMP-dependent PKA activation at the interface between the ER and mitochondria. This ERMCS-localised GLP-1R signalling hub requires the engagement of the AKAP SPHKAP, known to assemble PKA-RIα aggregates at the ER[32,33], with VAPB playing a central role as the tethering factor responsible for the separate recruitments of GLP-1R and SPHKAP to ERMCSs, the latter via the direct interaction of VAPB with the pFFAT motif of SPHKAP. It is important to highlight that SPHKAP is one of a few AKAPs known to recruit PKA-RI instead of RII subunits[40]. This is highly relevant as PKA-RIα aggregates undergo LLPS to form biomolecular condensates[65] which spatially restrict PKA activity, leading to signal compartmentalisation[25], explaining why the direct presence of endosomal GLP-1R at ERMCSs is required to assemble this signalling hub, and uncovering a new modality of GLP-1R signal transduction which involves the local assembly of a biomolecular condensate containing not only signalling mediators such as SPHKAP/PKA, and also presumably adenylate cyclase, but also downstream targets for PKA phosphorylation such as the MICOS complex and other ERMCS components, some likely found amongst the active GLP-1R interactors identified in our interactome analysis. An additional level of regulation could be provided by kinases and/or phosphatases controlling VAPB/SPHKAP/PKA-RIα assembly/disassembly, such as DYRK1A, recently shown to be recruited to VAPB/SPHKAP/PKA-RIα condensates in neurons[33] and previously identified in a kinome study to be regulated by GLP-1R activity[66], as well as by the autophagic degradation of the VAPB/SPHKAP/PKA-RIα complex via VAPB/SPHKAP interaction with its paralog AKAP11 and GSK3α/β, a pathway recently shown in neurons to control neurotransmission, and to be defective in AKAP11-associated psychiatric disorders[33].

The abovementioned signalling hub appears to be equally important for the control of β-cell function, a notion supported by human genetic data of genome-wide significance unveiling an association between *SPHKAP* gene variants, random blood glucose levels/ T2D, and high BMI. As we observe a correlation between *SPHKAP* variant effects in T2D and high BMI (as a proxy for obesity), we propose here that SPHKAP might play a pleiotropic role in both disorders by the engagement of common signalling pathways in different tissues. These could involve the GLP-1R, known to signal in anorectic neurons to regulate food intake as well as in β-cells to potentiate insulin secretion and β-cell survival[67], as well as other metabolically relevant GPCRs, via VAPB engagement in tissues where this AKAP is expressed, including the pancreas, brain and heart[29,32]. For example, this mechanism might potentially regulate liver mitochondrial function, a possibility supported by the previously published interaction between the closely related glucagon receptor and VAPB in hepatocytes[68]. The ERMCS-localised signalling hub identified here is therefore likely to mediate the action of multiple GPCRs, for example, GIPR, as also shown here. Agreeably, beyond the effect observed in exendin-4-induced responses, we find a significant impact of SPHKAP downregulation in mitochondrial morphology and other parameters already present under vehicle conditions.

Another outcome of the present study is the identification of the MICOS complex component MIC19 as a downstream target of ERMCS-localised GLP-1R-induced PKA phosphorylation. MIC19 is not only a key member of mitochondrial cristae junctions, responsible for the organisation of mitochondrial inner and outer membrane contacts and architecture of respiratory complexes and the proton pump, but also plays a prominent role in the transfer of lipid precursors across ERMCSs for the biosynthesis of the key mitochondrial lipid

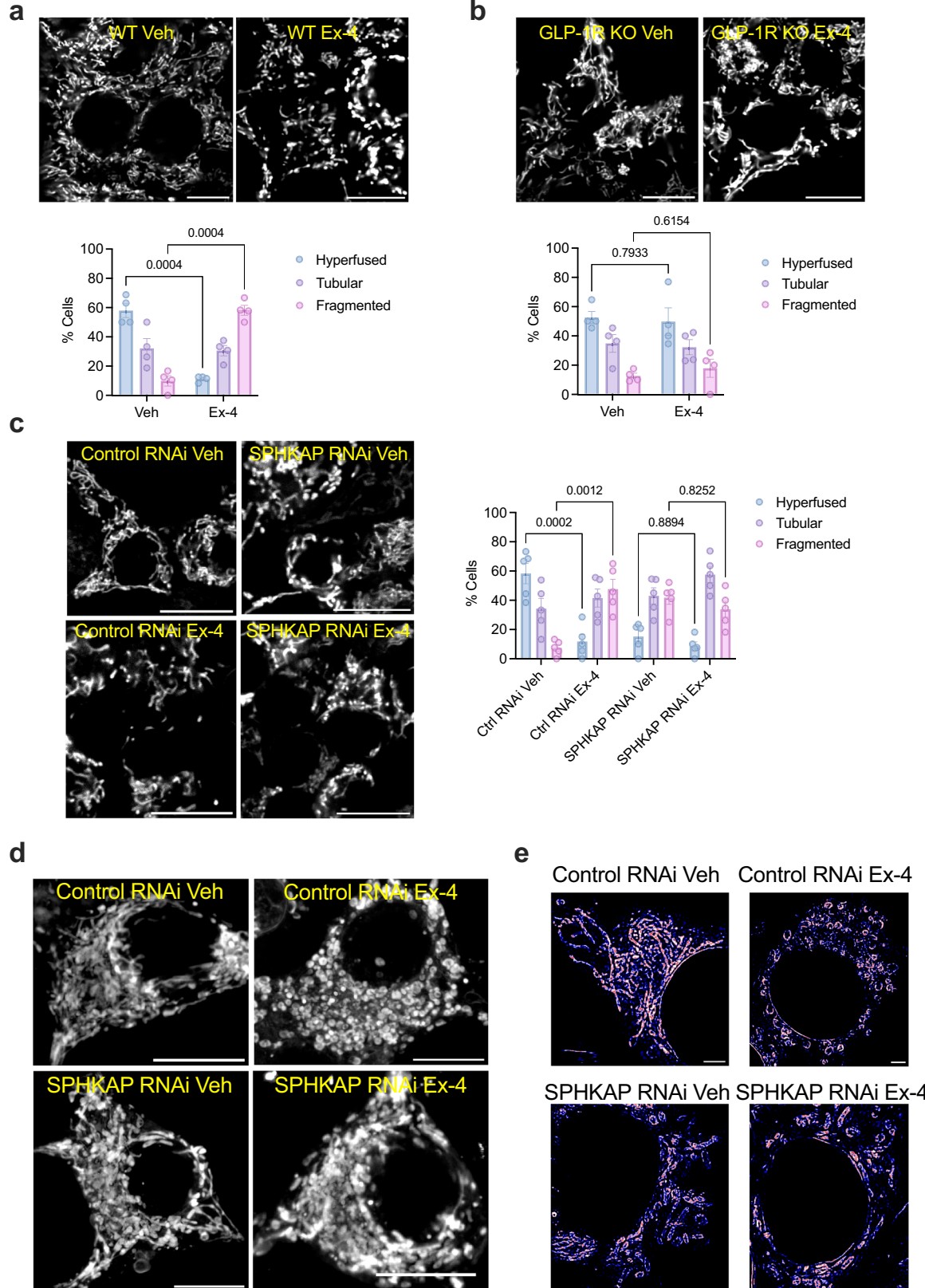

cardiolipin[69], therefore connecting mitochondrial cristae organisation with the generation of membrane curvature and control of mitochondrial function afforded by this lipid[10]. We therefore hypothesise that the GLP-1R-dependent phosphorylation of MIC19 at ERMCSs might not only highlight a direct role for the receptor in the control of mitochondrial dynamics, but also regulate lipid precursor transfer across ERMCSs to sustain mitochondrial turnover. This is particularly relevant as alterations in cardiolipin profiles have been reported in several diseases where GLP-1RAs play beneficial roles, including Alzheimer's, Parkinson's, obesity, and T2D[70]. Interestingly, our MS interactomic analysis also identifies TAMM41, an essential enzyme for the biosynthesis of cardiolipin[71], as an active GLP-1R interactor.

**Fig. 6 | GLP-1R/SPHKAP-dependent mitochondrial remodelling. a** Confocal microscopy analysis of mitochondrial morphology in vehicle (Veh) *versus* exendin-4 (Ex-4)-stimulated WT INS-1 832/3 cells; top, representative images with mitochondria labelled with MitoTracker Red; size bars, 10 μm; bottom, percentage of cells classified as presenting hyperfused, tubular or fragmented mitochondria per condition; n = 4 biologically independent experiments, p values as indicated by two-way ANOVA with Tukey post-hoc test. **b** As in (**a**) for INS-1 832/3 GLP-1R KO cells; n = 4 biologically independent experiments; p values as indicated by two-way ANOVA with Tukey post-hoc test. **c** As in (**a**) for Control *versus* SPHKAP RNAi-treated INS-1 832/3 cells under Veh *versus* Ex-4-stimulated conditions; n = 5 biologically independent experiments, p values as indicated by two-way ANOVA with Tukey post-hoc test. **d** Maximum intensity projections (MIP) from confocal z-stacks of mitochondria in vehicle (Veh) *versus* Ex-4-stimulated Control and SPHKAP RNAi-treated INS-1 832/3 cells; size bars, 10 μm; images representative of n = 3 biologically independent experiments. **e** Mitochondrial morphology in Control *versus* SPHKAP RNAi-treated INS-1 832/3 cells under Veh *versus* Ex-4-stimulated conditions, imaged by super-resolved STED microscopy; size bars, 2 μm. Images representative of n = 3 biologically independent experiments. Data are mean ± SEM.

Amongst the mitochondrial processes modulated by SPHKAP-dependent ERMCS-localised GLP-1R cAMP/PKA signalling, we have found acute increases in mitochondrial fission and mitophagy, two key quality control mechanisms which enable the clearance of damaged, unhealthy, or old mitochondria, and hold a prominent role in the maintenance of β-cell function[56] and identity[60]. Loss of these functions can lead to impaired insulin release and glucose intolerance[72], with the present study indicating that controlling these processes is an important mechanism engaged by the GLP-1R to exert its beneficial action on β-cell function and survival under oxidative stress conditions.

Amongst the major metabolic triggers of β-cell dysfunction is glucose and lipid-induced cytotoxicity. In the context of obesity, elevated glucose and fatty acids increase insulin demand and contribute to β-cell oxidative stress. Additionally, excess free fatty acids accumulated in β-cells induce further ROS generation, contributing to ER stress and β-cell dysfunction. β-cell mitochondria therefore need to appropriately adapt to glucolipotoxic conditions to avoid progression towards overt T2D[73]. Changes in mitochondrial morphology have been previously associated with modified nutrient utilisation, with a strong correlation between mitochondrial fragmentation and increased lipid utilisation via fatty acid oxidation (FAO)[74], a process inhibited in β-cells under high glucose conditions to favour metabolic coupling towards insulin secretion[73,75] that could potentially be modulated by the GLP-1R under excess lipids to enable lipid clearance and detoxification[73]. In support of this hypothesis, our MS interactome indicates the association of active GLP-1R with a range of factors involved in lipid metabolism, including lipases and regulators of lipase transfer to lipid droplets such as Mest[76] and Copb1[77], lipid metabolising enzymes including TMEM120A[78], Acsl4[79], Fads1[80], Chp1[81], Cds2[82], and Apgat4[83], and mitochondrial FAO and ketone body enzymes (HADHA/B[84], Bdh1[85]), as well as the ER/lipid droplet small GTPase Rab18[86], several subunits of the mitochondrial ATP synthase, the respiratory chain subunit SDHA, and the FAO-fuelled plasma membrane Na⁺/K⁺ pump[87]. Interestingly, a recent study has identified ERMCSs as required for lipid radical transfer from mitochondria to the ER to reduce mitochondrial oxidative stress[88], which could represent an alternative mechanism for ERMCS-localised GLP-1R signalling to enhance β-cell survival under glucolipotoxic conditions. Overall, the control of lipid metabolism is a likely effect of GLP-1R signalling at ERMCSs worthy of further investigation, with the SPHKAP paralog AKAP11 already shown to play an important role in lipid metabolism in astrocytes[89].

Beyond interaction with lipid enzymes, our MS interactome analysis indicates that the GLP-1R binds to ERMCS-localised Sarcoendoplasmic Reticulum Calcium ATPase (SERCA) pump in the ER and voltage-dependent anion-selective channel protein 2 (VDAC2) in the OMM, both involved in calcium/ion transfer across ERMCSs[90], suggesting a prominent role for the receptor in the control of intra-organelle ion homoeostasis, with important implications for the regulation of mitochondrial function, ER stress responses and insulin secretion[91]. Overall, these MS-identified interactions suggest that the effect of GLP-1R PKA signalling at ERMCS might extend well beyond the MICOS complex into a more comprehensive programme of ER, mitochondria and lipid metabolic pathway regulation.

In summary, this investigation has unveiled the existence of a newly characterised inter-organelle signalling hub in pancreatic β-cells which involves active endosomal GLP-1R localisation to ERMCSs via VAPB binding and engagement of the AKAP SPHKAP to enable ERMCS-restricted cAMP/PKA activity via the assembly of PKA-RIα biomolecular condensates, leading to PKA phosphorylation of the MICOS complex, enhanced mitochondrial remodelling and improved mitochondrial function, with important implications for the modulation of β-cell function and survival and the potential to serve as a mechanism for GLP-1R-induced mitochondrial adaptation during metabolic stress and excess nutrient exposure. Further studies to investigate this later possibility, including examination of the effect of GLP-1R in β-cell lipid metabolism and mitochondrial nutrient utilisation under normal and glucolipotoxic stress conditions are warranted, as well as the investigation of the in vivo role of SPHKAP in GLP-1R-dependent glucoregulation.

## Methods

### Cell culture

INS-1 832/3 rat insulinoma cells were maintained in RPMI-1640 at 11 mM D-glucose, supplemented with 10% foetal bovine serum (FBS), 10 mM HEPES, 2 mM L-glutamine, 1 mM pyruvate, 50 μM β-mercaptoethanol and 1% penicillin/streptomycin. INS-1 832/3 cells stably expressing SNAP/FLAG-tagged human GLP-1R (hGLP-1R) or human GIPR (hGIPR) were generated by transfection of pSNAP-GLP-1R or pSNAP-GIPR (Cisbio) into INS-1 832/3 cells lacking endogenous GLP-1R or GIPR expression (INS-1 832/3 GLP-1R or GIPR KO, a gift from Dr Jacqueline Naylor, AstraZeneca[92]), followed by G418 (1 mg/mL) selection. Mycoplasma testing was performed biannually. Cell culture reagents were obtained from Sigma or Thermo Fisher Scientific.

### Peptides and fluorescent probes

Exendin-4, GLP-1 and GIP were purchased from Bachem, semaglutide was from Imperial College London Healthcare NHS Trust pharmacy, and tirzepatide was a gift from Sun Pharmaceuticals. Custom analogues (exendin-F1 and exendin-D3) were from WuXi AppTec, China, and danuglipron and orforglipron were purchased from MedChemExpress. SNAP-Surface fluorescent probes were from New England Biolabs and MitoTracker™ Red CMXRos and ER-Tracker Red were from Thermo Fisher Scientific.

### GLP-1R β-cell interactome

INS-1 832/3 SNAP/FLAG-hGLP-1R cells were treated in 6-cm dishes with vehicle or 100 nM GLP-1RA (GLP-1, exendin-4, exendin-F1 or exendin-D3) for 5 minutes. Following stimulation, cells were washed in ice cold PBS and osmotically lysed in lysis buffer (50 mM Tris-HCl, pH 7.4, 150 mM NaCl, 1% Triton X-100, 1 mM EDTA) with 1% protease inhibitor (Roche) and 1% phosphatase inhibitor cocktail (Sigma–Aldrich) at 4 °C. Cell lysates were centrifuged at 17,000 × g and 4 °C for 10 minutes, and supernatants added to 40 μL anti-FLAG M2 agarose beads (A2220, Sigma-Aldrich) for overnight rotating incubation at 4 °C to increase binding efficiency. Samples were centrifuged at 5000 × g for 30 seconds, and bead pellets washed 3× in 500 μL wash buffer (50 mM Tris-HCl, 150 mM NaCl, pH 7.4) before storage at −80 °C until further

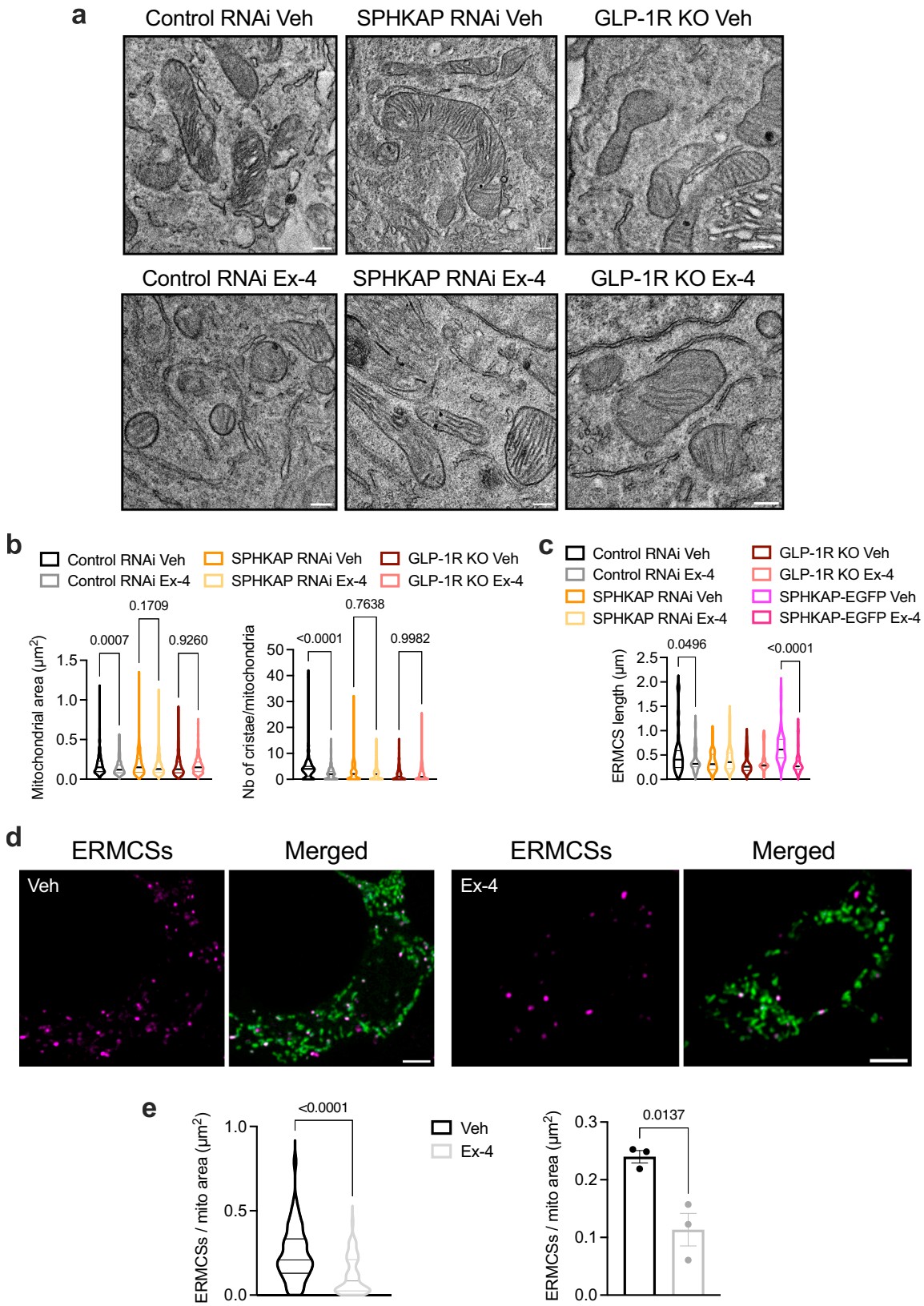

analysis. The number of biological replicates per condition was 4 for all conditions apart from GLP-1-stimulated samples, which included 3 replicates. Negative controls included INS-1 832/3 SNAP/FLAG-hGLP-1R cell lysates incubated with empty Sepharose beads (Sigma–Aldrich) and INS-1 832/3 cell lysates (not expressing the tagged receptor) incubated with anti-FLAG M2 beads.

Beads were resuspended in 150 μL 1 M urea in 20 mM HEPES pH 8.0 with 1 mM DTT, followed by addition of 1.5 μg Trypsin gold (Promega) with overnight digestion at 37 °C. Supernatants were subsequently acidified with 0.5% trifluoroacetic acid (TFA), followed by desalting using reversed-phase spin tips (Glygen Corp) and drying using vacuum centrifugation. Dried peptides were resuspended in 0.1%

**Fig. 7 | Exendin-4-induced changes in mitochondrial morphology and number of ERMCSs per mitochondria. a** TEM micrographs of resin-embedded ultrathin sections depicting mitochondrial morphology in Control *versus* SPHKAP RNAi-treated INS-1 832/3 cells, and INS-1 832/3 GLP-1R KO cells, under vehicle (Veh) *versus* exendin-4 (Ex-4)-stimulated conditions; size bars, 100 nm; images representative of *n* = 3 biologically independent experiments. **b** Quantification of mitochondrial area and number of cristae per mitochondria from ~20 TEM micrographs per condition per experimental repeat from (**a**); *n* = 3 biologically independent experiments, *p* values as indicated by one-way ANOVA with Šidák post-hoc test. **c** Quantification of ERMCS length in Control *versus* SPHKAP RNAi-treated INS-1 832/3 cells and INS-1 832/3 GLP-1R KO cells from (**a**), as well as in Veh *versus* Ex-4-treated SPHKAP-EGFP-

expressing WT INS-1 832/3 cells; *p* values as indicated by one-way ANOVA with Šidák post-hoc test. **d** Confocal images of WT INS-1 832/3 cells co-expressing the mitochondrial marker 4xMTS-Halo (green) and splitFAST reporters ER-RspA-NFAST and OMM-short-RspA-CFAST, labelled with a Lime splitFAST ligand to detect ERMCSs (magenta) under Veh *versus* Ex-4-stimulated conditions; size bars, 5 μm; images representative of *n* = 3 biologically independent experiments. **e** Quantification of the number of ERMCSs per mitochondrial area from data in (**d**); results per cell (left) and per independent repeat (right) shown; 50–57 cells imaged from *n* = 3 biologically independent experiments, p values as indicated by two-tailed unpaired *t*-tests. Data are violin plots or mean ± SEM.

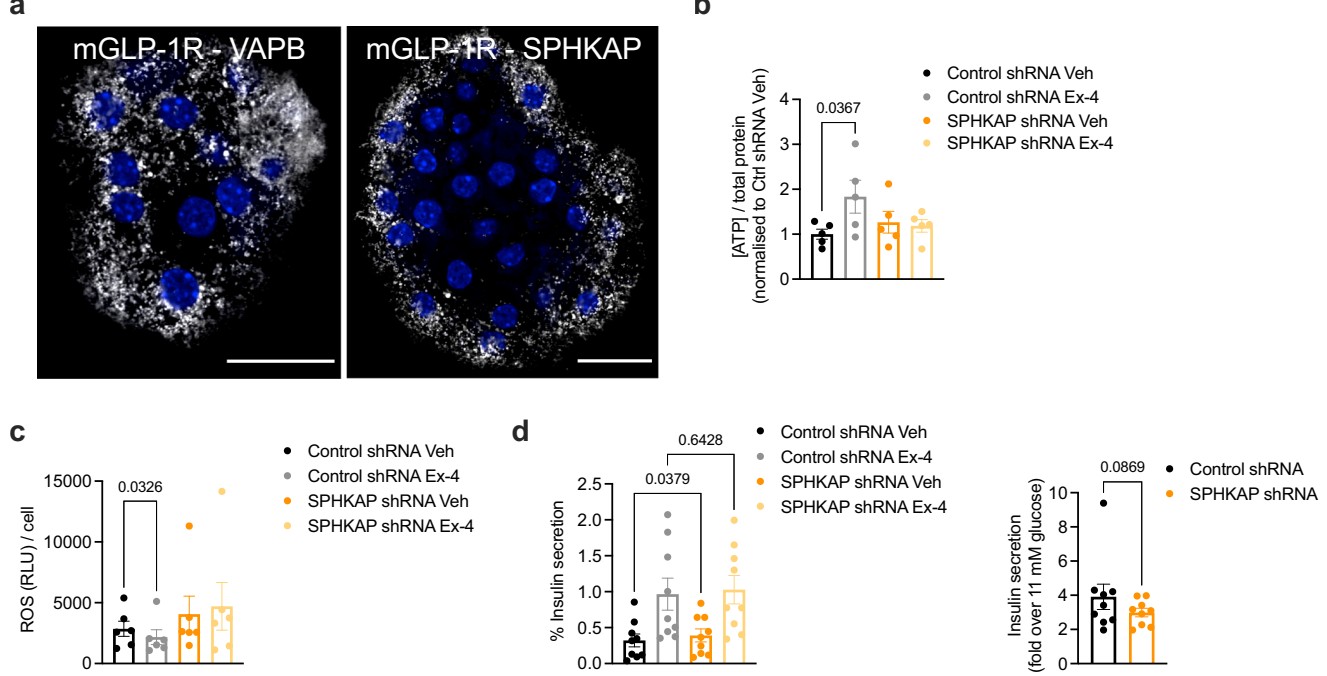

**Fig. 8 | SPHKAP control of GLP-1R function in primary mouse islets. a** PLA showing interactions between endogenous GLP-1R and VAPB (left) or SPHKAP (right) in exendin-4 (Ex-4)-stimulated primary mouse islets; size bars, 20 μm; images representative of *n* = 3 biologically independent experiments. **b** Ex-4-induced changes in ATP levels per total protein in lysates from Control *versus* SPHKAP shRNA-transduced primary mouse islets; *n* = 5 biologically independent experiments, p value as indicated by one-way ANOVA with Šidák post-hoc test. **c** Ex-4-induced changes in thapsigargin-triggered ROS accumulation per cell in Control

*versus* SPHKAP shRNA-transduced primary mouse islets; *n* = 6 biologically independent experiments, *p* value as indicated by one-way ANOVA with Šidák post-hoc test. **d** Ex-4-induced potentiation of insulin secretion in Control *versus* SPHKAP shRNA-transduced primary mouse islets; *n* = 9 biologically independent experiments, *p* values as indicated by one-way ANOVA with Šidák post-hoc test; fold over 11 mM glucose also shown, *p* value as indicated by two-tailed ratio paired *t*-test. Data are mean ± SEM.

TFA and Liquid Chromatography Tandem Mass Spectrometry (LC-MS/MS) performed in single replicate injections as previously described[93] using an Ultimate 3000 nano-HPLC coupled to a Q-Exactive mass spectrometer (Thermo Scientific) via an EASY-Spray source. Peptides were loaded onto a trap column (Acclaim PepMap 100 C18, 100 μm × 2 cm) at 8 μL/minute in 2% acetonitrile and 0.1% TFA. Peptides were eluted on-line to an analytical column (Acclaim Pepmap RSLC C18, 75 μm × 75 cm). A 90-minute stepped gradient separation was used with 4–25% acetonitrile 80%, formic acid 0.1% for 60 minutes, followed by 25–45% for 30 minutes. Eluted peptides were analysed by Q-Exactive operating in positive polarity and data-dependent acquisition mode. Ions for fragmentation were determined from an initial MS1 survey scan at 70,000 resolution (at m/z 200), followed by higher-energy collisional dissociation of the top 12 most abundant ions at a resolution of 17,500. MS1 and MS2 scan AGC targets were set to 3e6 and 5e4 for a maximum injection time of 50 and 75 mseconds, respectively. A survey scan range of 350–1800 m/z was used, with a normalised collision energy set to 27%, minimum AGC of 1e3, charge

state exclusion enabled for unassigned and +1 ions and a dynamic exclusion of 45 seconds.

Data was processed using the MaxQuant software platform (v1.6.10.43), with database searches carried out by the in-built Andromeda search engine against the Swiss-Prot Rattus Norvegicus database (Downloaded—21st May 2022; entries: 8132) concatenated with the human GLP-1R protein sequence. A reverse decoy database approach was used at a 1% false discovery rate (FDR) for peptide spectrum matches and protein identifications. Search parameters included: maximum missed cleavages set to 2, variable modifications of methionine oxidation, protein N-terminal acetylation, asparagine deamidation and cyclisation of glutamine to pyroglutamate.

Label-free quantification (LFQ) was enabled with a minimum ratio count of 1. Proteins were filtered for those containing at least one unique peptide and identified in all biological replicates included per condition. For normalisation, proteins were ranked by fold change increase in abundance compared with control (vehicle-treated) immunoprecipitates. The resulting data was pre-processed by

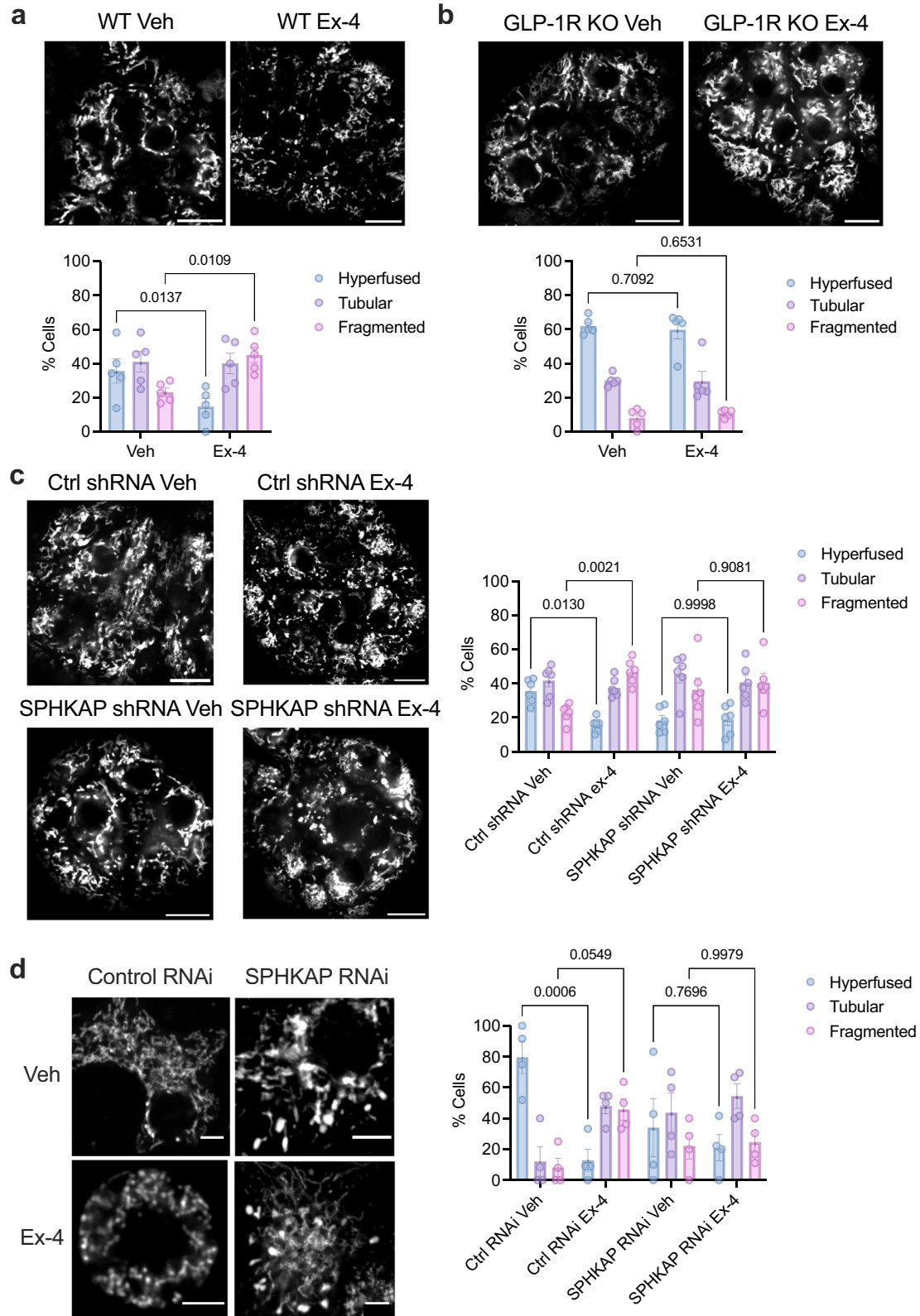

elimination of proteins that are non-specifically bound and residual background proteins, such as common contaminants and reverse database proteins. To further facilitate the selection of GLP-1R interactors, the following criteria were established; (i) identified at least twice from four runs; (ii) not nuclear proteins; and (iii) previously identified in pancreatic β-cells. Untargeted LFQ intensity values,

normalised to hGLP-1R levels in each immunoprecipitated sample, were analysed by LFQ Analyst[94] to identify differences in levels of individual protein-GLP-1R interactions between vehicle and GLP-1RA-treated conditions. Paired *t*-tests adjusted for multiple comparisons using Benjamin–Hochberg FDR correction and Perseus type imputations were used for statistical analysis. To generate novel information

**Fig. 9 | SPHKAP control of GLP-1R-dependent mitochondrial remodelling in primary mouse islets. a** Confocal microscopy analysis of mitochondrial morphology in vehicle (Veh) *versus* exendin-4 (Ex-4)-stimulated WT primary mouse islets; top, representative images with MitoTracker Red-labelled mitochondria; size bars, 20 μm; bottom, percentage of islet cells classified as presenting hyperfused, tubular or fragmented mitochondria per condition; *n* = 5 biologically independent experiments (islet preps from separate mice), *p* values as indicated by two-way ANOVA with Tukey post-hoc test. **b** As in (**a**) for islets from GLP-1R KO mice; *n* = 5 biologically independent experiments; *p* values as indicated by two-way ANOVA with Tukey post-hoc test. **c** As in (**a**) for Control *versus* SPHKAP shRNA-transduced mouse islets under Veh *versus* Ex-4-stimulated conditions; *n* = 6 biologically independent experiments, p values as indicated by two-way ANOVA with Tukey post-hoc test. **d** As in (**a**) for Control *versus* SPHKAP RNAi-treated dispersed primary mouse islet cells under Veh *versus* Ex-4-stimulated conditions; size bars, 5 μm; *n* = 4 biologically independent experiments, *p* values as indicated by two-way ANOVA with Tukey post-hoc test. Data are mean ± SEM.

about the functional pathways represented by these potential GLP-1R interactors, we performed pathway enrichment analysis with the GLP-1R binding partners exhibiting changes between vehicle and exendin-4-stimulated conditions using gProfiler.

## cDNA and siRNA transfection

INS-1 832/3 cells were transfected with relevant amounts of plasmid cDNA outlined in Supplementary Table 1, or siRNA outlined in Supplementary Table 2, using Lipofectamine 2000 (Life Technologies) in media without antibiotics according to the manufacturer's instructions. Experiments were carried out 24–48 hours after plasmid cDNA, or 72 hours after siRNA transfection.

## Isolation and culture of pancreatic islets

WT or GLP-1R KO mice (generated *in house*[95]) were humanely euthanised before inflating the pancreas through the common bile duct with RPMI-1640 medium containing 1 mg/mL collagenase (Nordmark Biochemicals). The pancreas was then dissected, incubated at 37 °C for 10 minutes, and digestion halted with cold RPMI-1640 supplemented with 10% FBS (F7524, Sigma–Aldrich) and 1% (v/v) penicillin/streptomycin solution (15070-063, Invitrogen). Mouse islets were subsequently washed, purified using a Histopaque gradient (Histopaque-1119 and Histopaque-1083, Sigma–Aldrich) and rested overnight at 37 °C in a 95% $O_2$/5% $CO_2$ incubator before performing experiments in mouse islet culture medium (RPMI-1640 with 11 mM D-glucose, supplemented with 10% FBS and 1% penicillin/streptomycin).

Human islets were received from the isolation unit at the Centre Européen d'Etude du Diabète (CEED), Strasbourg, or from isolation centres ascribed to the Integrated Islet Distribution Program (IIDP), National Institute of Diabetes and Kidney Diseases (NIDDK), National Institutes of Health (NIH), USA, with all ethical approvals in place. Human islets were cultured on arrival in RPMI-1640 medium supplemented with L-glutamine, 10% FBS, 1% penicillin/streptomycin solution, 0.25 μg/mL amphotericin B, and 5.5 mM glucose at 37 °C in a 95% $O_2$/5% $CO_2$ incubator. Donor characteristics are indicated in Supplementary Table 3.

## Islet shRNA transduction

Mouse islets were transduced with Control or SPHKAP shRNA-expressing lentiviral particles (MISSION® shRNA, Target Clone ID TRCN0000088765 mouse *SPHKAP*, Sigma–Aldrich) at an MOI of 1 and cultured for 72 hours before experiments.

## Co-immunoprecipitation

INS-1 832/3 cells previously transfected with the relevant plasmids, INS-1 832/3 SNAP/FLAG-hGLP-1R or SNAP/FLAG-hGIPR cells were seeded in 6-well plates at $1.2 \times 10^6$ cells per well and incubated overnight. Cells were treated in normal media at 37 °C with vehicle, 100 nM exendin-4 for 5 minutes or 100 nM GIP for the indicated times, washed once with ice cold PBS and lysed with 1 mL lysis buffer supplemented with 1% phosphatase inhibitor cocktail (Sigma-Aldrich) and 1% complete EDTA-free protease inhibitor cocktail (Sigma–Aldrich) on ice. A cross-linking step [2 mM 3,3′-Dithiodipropionic acid di(N-hydroxysuccinimide ester) (DSP, Sigma-Aldrich) for 30 minutes at room temperature followed by quenching with 20 mM Tris HCl for 15 minutes and 3× PBS

washes] was performed prior to cell lysis for the SPHKAP-EGFP co-immunoprecipitation experiments to preserve protein-protein interactions despite proteolytic cleavage of the full-length fusion protein during the IP process. Cell lysates were centrifuged at $17,000 \times g$ for 10 minutes, and supernatants added to 40 μL anti-FLAG M2 Magnetic beads (Sigma–Aldrich), anti-HA Magnetic beads (Thermo Fisher Scientific), or ChromoTek GFP-Trap® Magnetic Particles (Proteintech), and incubated at 4 °C for 1 hour. Beads were then washed 3x in 500 μL wash buffer and proteins eluted with urea buffer (200 mM Tris-HCl, pH 6.8; 5% w/v SDS, 8 M urea, 100 mM dithiothreitol, 0.02% w/v bromophenol blue) at 37 °C for 10 minutes before SDS-PAGE and western blot analysis.

## SDS-PAGE and western blotting

Samples were loaded onto 10% SDS-polyacrylamide gels then transferred to 0.45 μm PVDF membranes and blocked with 5% skimmed milk [or 5% bovine serum albumin (BSA) for anti-phospho antibodies] in TBS buffer supplemented with 0.1% Tween 20 (TBS-T). Membranes were incubated with primary antibody overnight diluted in TBS-T/milk (or TBS-T/BSA for anti-phospho antibodies) at 4 °C, washed in TBS-T and incubated for 1 hour at RT with the corresponding HRP-conjugated secondary antibody in TBS-T/milk. Following a further 3× washes in TBS-T, membranes were developed with Clarity Western ECL substrate (BioRad) and imaged on a Biorad ChemiDoc or in a Xograph Compact X5 processor with Fuji medical X-ray films. Specific band densities were quantified in Fiji. The following antibodies were used (source, catalogue number, dilution): rabbit anti-VAPB (Antibodies.com, A14703, 1:5000); rabbit anti-SNAP tag (New England Biolabs, P93105, 1:5000); mouse anti-SPHKAP[32] (kind gift from Dr Nicholas Vierra, University of Utah, L131/17; 1:1000); rabbit anti-SPHKAP[32] (kind gift from Dr Nicholas Vierra, University of Utah, 36690, 1:1000); mouse anti-Tubulin (Sigma-Aldrich, T-5168, 1:2000); rabbit anti-HA tag (Abcam, Ab9110, 1:1000); rabbit anti-GFP (Proteintech, 50430-2-AP, 1:2000); rabbit anti-phospho-PKA Substrate (RRXS*/T*) (Cell Signaling, 9624, 1:1000); rabbit anti-Drp1 (D6C7) (Cell Signaling, 8570S, 1:1000); rabbit anti-phospho-Drp1 Ser616 (D9A1) (Cell Signaling, 4494S, 1:1000); mouse anti-human Tom20 (F-10) (Santa Cruz Biotechnology, sc-17764, 1:1000); goat anti-mouse IgG-HRP (Abcam, ab205719, 1:5000); goat anti-rabbit IgG HRP (Abcam, ab205718, 1:2000).

## SNAP/FLAG-hGLP-1R localisation by confocal microscopy

INS-1 832/3 SNAP/FLAG-hGLP-1R cells seeded onto 35-mm glass-bottom dishes (MatTek) were transfected with the indicated constructs. SNAP-FLAG-hGLP-1R was labelled for 15 minutes at 37 °C with SNAP-Surface Alexa Fluor 546 or 647 in full media prior to stimulation with 100 nM exendin-4 and imaging in a Leica Stellaris 8 inverted confocal microscope with a 63×/1.40 NA oil immersion objective with Lightning confocal super resolution modality from the facility for imaging by light microscopy (FILM) at Imperial College London. Mouse islets were transduced 24 hours prior to imaging with an adenoviral SNAP/FLAG-hGLP-1R vector (custom made by VectorBuilder) at a MOI of 1 and subsequently labelled for 10 minutes at 37 °C with SNAP-Surface Alexa Fluor 647 before 5-minute stimulation with indicated agonists and imaging of whole islets in the same microscope as above. Images were analysed in Fiji.

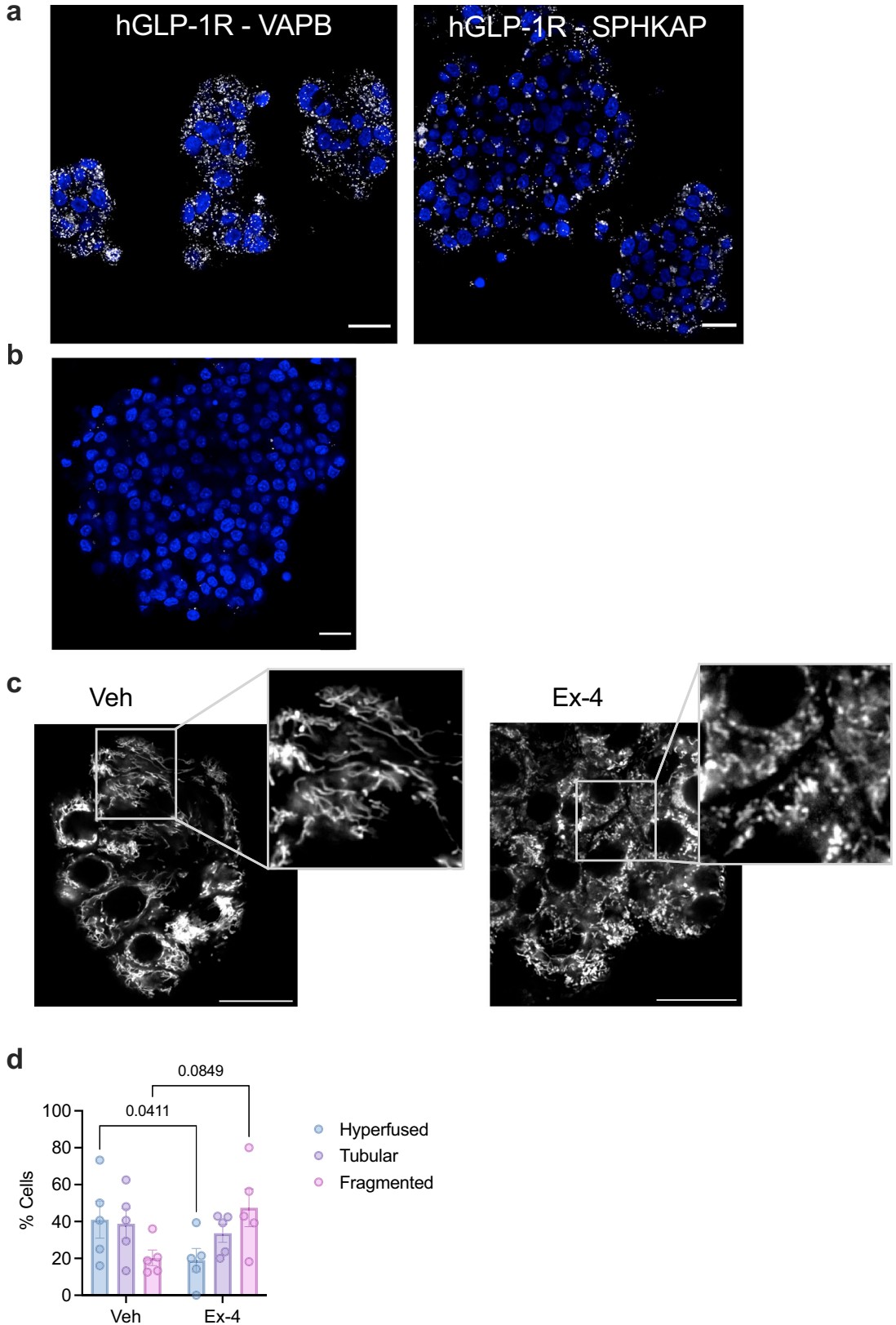

## MINFLUX microscopy

INS-1 832/3 SNAP/FLAG-hGLP-1R cells were seeded onto 18-mm coverslips, labelled with SNAP-Surface Alexa Fluor 647 and stimulated with 100 nM exendin-4 as above. Cells were then fixed in 4% paraformaldehyde and processed for immunofluorescence with primary rabbit anti-VAPB antibody (Sigma-Aldrich, HPA013144, 1:150) followed by secondary anti-rabbit FLUX 680 antibody (custom-made at Abberior, 1:200). Directly before MINFLUX imaging, samples were incubated with 150 nm gold fiducials (BBI solutions) for 5 minutes. After three brief washes with PBS, the samples were incubated with 10 mM $MgCl_2$ in PBS for a further 5 minutes and then mounted on a well slide in freshly prepared GLOX buffer (100 mM Tris-HCl pH 8, 10% w/v glucose,

**Fig. 10 | GLP-1R–VAPB/SPHKAP interactions and GLP-1R-dependent mito-chondrial remodelling in primary human islets. a** PLA showing interactions between endogenous GLP-1R and VAPB (left) or SPHKAP (right) in exendin-4 (Ex-4)-stimulated primary human islets; size bars, 20 µm. Images representative of $n = 3$ biologically independent experiments (islets from independent donors). **b** Negative control (no primary antibody) for PLA data shown in (**a**); nuclei (DAPI), blue; size bar, 20 µm. **c** Confocal microscopy analysis of mitochondrial morphology in vehicle (Veh) *versus* Ex-4-stimulated primary human islets; mitochondria labelled with MitoTracker Red; size bars, 20 µm; inset, magnification areas; images representative of $n = 5$ biologically independent experiments (islets from independent donors). **d** Percentage of human islet cells classified as presenting hyperfused, tubular or fragmented mitochondria per condition; $n = 5$ biologically independent experiments, $p$ values as indicated by two-way ANOVA with Tukey post-hoc test. Data are mean ± SEM.

0.4 mg/mL glucose oxidase, 64 µg/mL catalase) supplemented with 10 mM MEA and sealed with two part silicon (eco-sil speed, Picodent).

MINFLUX imaging was performed on an Abberior instruments 3D MINFLUX microscope[14]. Briefly, the microscope is built on a motorised inverted microscope body (IX83, Olympus) with a CoolLED pE-4000 for epifluorescence illumination. The system is equipped with 405 nm, 485 nm, 561 nm, and 642 nm laser lines, as well as a 980 nm IR laser and xyz piezo (Piezoconcept) for active sample stabilisation. Images were acquired with a 100×/1.45 NA UPLXAPO oil objective (Olympus). During acquisition, the pinhole was set to 0.83 AU. Detection was performed with two APDs in spectral windows Cy5 near (650–685 nm) and Cy5 far (685–740 nm). MINFLUX imaging was performed with the default 2D imaging sequence supplied with the system. In brief, the sequence defines parameters used in the iterative localising process of MINFLUX, culminating in a final iteration with a targeted coordinate pattern (TCP) diameter of 40 nm and minimum photon threshold of 150 photons.

Two-colour spectral separation was performed based on detection channel ratio (DCR) values accounting for background inhomogeneity. DCR values from the HeadStart iterations were weighted averaged by the corresponding effective photon counts at the MINFLUX pattern offset. A two-component Gaussian Mixture Model was then fitted to the resulting values and the assigned probability of belonging to distribution 1 or 2 was retrieved. A localisation was then assigned to one colour if its probability was greater than 95% or otherwise set to a 'not defined' category. All localisations associated with the same molecule were subsequently assigned to the colour category that occurred most frequently within that molecule's trace. Molecules for which most localisations were not defined were not accounted for in the two-colour data. After two-colour separation, localisations were filtered by efo (<80 kHz) and number of localisations per trace (>3). The centroid of each remaining trace was calculated and DBSCAN cluster analysis used to identify centroid clusters (minimum points = 10, epsilon = 150 nm). Nearest neighbour analysis was performed between FLUX 680 and Alexa Fluor 647 clusters using the K-D Tree neighbour search algorithm.

### Transmission electron microscopy (TEM)
Cells and islets were prepared for TEM and imaged as in ref. 18. Briefly, for SNAP/FLAG-hGLP-1R gold labelling, INS-1 832/3 SNAP/FLAG-hGLP-1R cells grown on Thermanox coverslips (Agar Scientific), or mouse islets transduced with SNAP/FLAG-hGLP-1R adenoviruses, were labelled with 2 µM cleavable SNAP-Surface biotin probe (kind gift of Dr Ivan Corrêa Jr, New England Biolabs) in full media, followed by incubation with 5 µg/mL NaN₃-free Alexa Fluor 488 Streptavidin, 5 or 10 nm colloidal gold conjugate (Thermo Fisher Scientific) to label SNAP/FLAG-hGLP-1R prior to stimulation with 100 nM exendin-4, fixation and processing for conventional TEM. Cells or islets were fixed at room temperature for 20 minutes in 2% PFA + 2% glutaraldehyde in 0.1 M sodium cacodylate, washed 2× with 0.1 M sodium cacodylate and 1× with ddH₂O, and then post-stained on ice with 2% osmium tetroxide (OsO₄) for 1 hour followed by 1% tannic acid at room temperature for 45 minutes, and washed 2× in 0.05 M sodium cacodylate before stepwise dehydration in 70%, 90%, and 100% EtOH and Epon resin embedding. Cells mounted on Epon stubs and resin-embedded islets in moulds were baked at 65 °C overnight. Samples were trimmed

and cut *en face* with a diamond knife (DiATOME) in a Leica Ultracut UCT 6 ultramicrotome before examination on an FEI Tecnai G2 Spirit TEM. Images were acquired in a charge-coupled device camera (Eagle) and analysed in Fiji. Mitochondrial area, number of cristae per mito-chondria and ERMCS length was determined as in ref. 96 by manually drawing mitochondria and ERMCSs visible from the TEM micrographs.

### GLP-1R–VAPB interaction assay by NanoBRET
INS-1 832/3 cells were co-transfected with 1 µg Venus-VAPB (generated *in house* from EGFP-VAPB) and 0.5 µg GLP-1R-NanoLuc[97]. NanoBRET assays were carried out as previously described[97]. Briefly, 24 hours after transfection, cells were washed with PBS, detached, and resuspended in NanoGlo Live Cell Reagent (Promega) with furimazine (1:40 dilution in HBSS) and seeded into white 96-well half-area plates. A baseline reading was taken for 5 minutes at 37 °C at 460 nm for NanoLuc emission and at 535 nm for Venus emission in a Flexstation 3 microplate reader. Cells were then stimulated with or without 1 µM of the indicated agonist and luminescent signal read for 30 minutes. Ratio (535/460 nm) BRET readings were normalised to baseline and to average vehicle readings for agonist-specific responses. Area under the curve (AUC) was calculated per agonist response using GraphPad Prism 10.2.1.

### Global and mitochondrial PKA assays
INS-1 832/3 cells were seeded onto 35-mm glass-bottom dishes (Mat-Tek) and transfected with enhanced excitation-ratiometric PKA bio-sensor pcDNA3.1(+)-ExRai-AKAR2 (for global PKA) or pcDNA3-mitoExRai-AKAR2 (for mitochondrial PKA)[27]. Cells were washed with PBS 24 hours after transfection and imaged in Krebs-Ringer bicarbonate-Hepes (KRBH) buffer (140 mM NaCl, 3.6 mM KCl, 1.5 mM CaCl₂, 0.5 mM MgSO₄, 0.5 mM NaH₂PO₄, 2 mM NaHCO₃, 10 mM Hepes, 0.1% BSA, saturated with 95% O₂/5% CO₂, pH 7.4) supplemented with 11 mM glucose and imaged ratiometrically at excitation wavelengths of 405 and 488 nm and emission of 530 nm using a Zeiss LSM-780 inverted confocal laser-scanning microscope with a 63× oil immersion objective, with images taken every 6 seconds, in a 37 °C heated chamber. A baseline reading was taken for 1 minute followed by stimulation with 1 µM exendin-4 and imaging for 5 minutes before addition of 10 µM forskolin (FSK) + 100 µM isobutyl methylxanthine (IBMX) for the last 2 minutes of acquisition to record maximal responses. AUCs were calculated using GraphPad Prism 10.2.1.

### cAMP assay
INS-1 832/3 cells were seeded onto 35-mm glass-bottom dishes (Mat-Tek) and incubated overnight with the Green Up cADDis biosensor in a BacMam vector (Montana Molecular) according to the manufacturer's instructions. Cells were washed with PBS and imaged in 11 mM glucose KRBH buffer using a Nikon Eclipse Ti microscope with an ORCA-Flash 4.0 camera (Hamamatsu) and Metamorph software (Molecular Devices) with a 20× air objective and 488 nm laser, with images taken every 6 seconds, on a 37 °C stage. A baseline reading was taken for 1 minute followed by stimulation with 1 µM exendin-4 for 5 minutes, and 10 µM FSK + 100 µM IBMX addition for the last 2 minutes of the acquisition for maximal responses. Raw fluorescent intensities were extracted in Fiji to plot an average intensity trace and AUCs calculated in GraphPad Prism 10.2.1.

## Mitochondrial morphology assay

INS-1 832/3 cells were seeded at 50,000 cells per well in 96-well glass-bottom imaging plates (Sigma–Aldrich) coated with poly-L-lysine and laminin (1:50). Primary mouse islets were dispersed by trituration with trypsin-EDTA 0.25% for 3 minutes and seeded as above, or imaged whole without dispersion. Both cells and islets were stimulated with 100 nM exendin-4 or vehicle for 10 minutes and subsequently labelled with MitoTracker Red CMXRos (Thermo Fisher Scientific), washed 1× with PBS and imaged in phenol red free RPMI-1640 in a Leica Stellaris 8 inverted confocal microscope with 63×/1.40 NA (islets) or 100×/1.40 NA (cells) oil immersion objectives. Five images were collected per condition and analysed using the Mitochondria Analyzer plugin for Fiji to determine mitochondrial mean area and form factor. Cells were also visually classified by predominant mitochondrial phenotype as in refs. 49,98 into (a) hyperfused (highly interconnected and elongated), (b) tubular (mixed) or (c) fragmented mitochondrial types. For INS-1 832/3 and dispersed islet cell experiments, 30–130 cells were analysed from $n = 4$, 5 independent repeats per experiment; for intact islet experiments, all visible cells in a total of 15–30 islets from $n = 5$, 6 independent repeats were analysed per experiment.

## Imaging of mitochondria by STED microscopy

INS-1 832-3 cells were seeded in glass-bottom dishes (MatTek), labelled with 500 nM PKmito Orange FX (Spirochrome) for 30 minutes, washed 3× in PBS and incubated in complete media for 30 minutes to recover. Cells were then treated with vehicle or 100 nM exendin-4 in complete media for 10 minutes prior to fixation in 4% PFA and imaging in PBS using a Leica Stellaris 8 microscope in TauSTED mode with a 100×/1.40 NA oil immersion objective. Cells were excited at 591 nm and pulsed depletion performed at 775 nm with 750 ps delay at 50% intensity.

## Mitochondrial membrane potential assay

INS-1 832/3 cells were seeded in 35-mm glass-bottom dishes (MatTek), incubated in 3 mM glucose KRBH for 1 hour, followed by incubation with 50 μM tetramethylrhodamine (TMRE) in 3 mM glucose KRBH for 15 minutes to label active mitochondria[99]. Cells were then washed with KRBH and imaged in 3 mM glucose KRBH with images taken every 12 seconds in a 60×/1.4 NA oil immersion objective with a 561 nm laser in a Nikon Eclipse Ti microscope with an ORCA-Flash 4.0 camera (Hamamatsu) and Metamorph software (Molecular Devices). A baseline reading was recorded for 1 minute followed by stimulation with 1 μM exendin-4 in 6 mM glucose KRBH for 2 minutes. Glucose levels were then increased to 20 mM and cells imaged for a further 5 minutes, followed by mitochondrial hyperpolarisation using 20 μg/mL oligomycin for 5 minutes and depolarisation using 20 μM carbonyl cyanide 4-trifluoromethoxyphenylhydrazone (FCCP) for 5 minutes. Raw fluorescent intensities were extracted in Fiji to obtain average intensity traces and AUCs calculated using GraphPad Prism 10.2.1.

## ATP assays

**By time-lapse microscopy.** INS-1 832/3 cells were seeded in 35-mm glass-bottom dishes (MatTek) and transfected with the ATP biosensor cyto-Ruby3-iATPSnFR1.0[100]. Cells were imaged 24 hours after DNA transfection in 11 mM glucose KRBH buffer in a Zeiss LSM-780 inverted confocal laser-scanning microscope using a 63×/1.4 NA oil immersion objective with a 488 nm laser, with images taken every 6 seconds. A baseline reading was recorded for 1 minute followed by stimulation with 1 μM exendin-4 for 5 minutes. Raw fluorescent intensities were extracted using Fiji to obtain average intensity traces and AUCs calculated using GraphPad Prism 10.2.1.

**Biochemical assay.** INS-1 832/3 cells seeded in 12-well plates, or mouse islets, were pre-incubated with KRBH buffer supplemented with 3 mM glucose for 30 minutes followed by stimulation with KRBH supplemented with 11 mM glucose ± 100 nM exendin-4 for 15 minutes. Cells

were lysed in KRBH + 0.1% Triton X-100 and sonicated, while islets were lysed in acidic ethanol (75% ethanol, 15 mM HCl). Samples were centrifuged at $17,000 \times g$ and stored at −20 °C until further analysis. ATP concentrations from islets and cells were determined using the ATP Determination Kit (A22066, Thermo Fisher Scientific) and results normalised to total protein levels per sample estimated by BCA assay.

## Insulin secretion assay

INS-1 832/3 cells were seeded in 6-well plates at $1.2 \times 10^6$ cells per well. For VAPB WT *versus* P56S experiments, cells were co-transfected with 40 ng proinsulin-NLuc and 200 ng of either pEGFP-VAPB, pEGFP-VAPB-P56S [generated *in house* by site directed mutagenesis from pEGFP-VAPB with Fwd 5′-CGATGATTCCGCTGTTGCTCCTCACACAGTACCTAC-3′ and Rev 5′- GTAGGTACTGTGTGAGGAGCAACAGCGGAATCA TCG-3′ mutagenesis primers and the Quick-Change Lightning kit (Agilent)], or pcDNA3.1+ as a control. For SPHKAP-EGFP experiments, cells were co-transfected as above with proinsulin-NLuc and either empty EGFP or SPHKAP-EGFP. For RNAi experiments, cells were transfected with Control *versus* Target (VAPB or SPHKAP) siRNA. Cells were incubated overnight in INS-1 832/3 media at 3 mM glucose 24 hours after cDNA and 48 hours after siRNA transfections and then pre-incubated for 30 minutes in KRBH buffer supplemented with 3 mM glucose before stimulation in KRBH buffer supplemented with 11 mM glucose ± 100 nM exendin-4 for 60 minutes. Cell supernatants were collected to assess secreted insulin and cells lysed in KRBH + 0.1% Triton X-100 with sonication for total insulin extraction. For cDNA-transfected cells, NanoGlo Live Cell Reagent with furimazine (1:40 dilution in HBSS) was added and relative luciferase units (RLU) recorded at 460 nm using a Flexstation 3 microplate reader.

Mouse islets were incubated in KRBH buffer supplemented with 3 mM glucose for 1 hour followed by stimulation in KRBH buffer supplemented with 11 mM glucose ± 100 nM exendin-4 for 30 minutes. Supernatants were collected and centrifuged at $150 \times g$ for 3 minutes. Islets were lysed in acidic ethanol (75% ethanol, 15 mM HCl), sonicated $3 \times 10$ seconds in a water bath and centrifuged at $17,000 \times g$ for 10 minutes to determine intracellular insulin concentrations.

For RNAi-transfected cells and shRNA-transduced islets, sample insulin concentration was determined using the Insulin Ultra-Sensitive HTRF Assay kit (Cisbio) according to the manufacturer's instructions.

## TUNEL assay

INS-1 832/3 SNAP/FLAG-hGLP-1R cells seeded onto coverslips were incubated with or without 5 μM thapsigargin in vehicle or 100 nM exendin-4-supplemented media for 8 hours before fixation and terminal deoxynucleotidyl transferase dUTP nick-end labelling (TUNEL) assay performed using the DeadEnd™ Fluorometric TUNEL system (Promega) following the manufacturer's instructions. Coverslips were mounted with ProLong™ Diamond Antifade Mountant with DAPI (Thermo Fisher Scientific) and imaged for fluorescein and DAPI (to assess total cell number) in a Zeiss LSM-780 inverted confocal laser-scanning microscope with a 20×/0/8 NA air objective. The percentage of TUNEL-positive cells was calculated in Fiji from 5 regions per coverslip.

## Proximity ligation assay (PLA)

Mouse and human islets were stimulated with vehicle or 100 nM exendin-4 in full media for 10 minutes at 37 °C. Islets were washed with PBS and fixed for a minimum of 1 hour. Islets were washed again with PBS and processed by PLA with the NaveniFlex Tissue MR Red kit (Navinci) following the manufacturer's instructions, with an extra permeabilization step. Antibody pairs for mouse islets were mouse anti-mouse GLP-1R Ab Mab 7F38 (DSHB, 5 μg/mL) and either rabbit anti-VAPB (A14703, Antibodies.com, 1:100), or rabbit anti-mouse SPHKAP (36690, kind gift from Dr Nicholas Vierra, University of Utah[32], 1:100); antibody pairs for human islets were mouse anti-human

GLP-1R Mab 3F52 (DSHB, 5 µg/mL) and either rabbit anti VAPB (A14703, Antibodies.com, 1:100), or rabbit anti-human SPHKAP (HPA042499, Atlas Antibodies, 1:500). For insulin co-staining, islets were further incubated with anti-insulin antibody (IR002, Dako Agilent Technologies, undiluted) overnight followed by secondary anti-guinea pig Alexa Fluor 647 (A21450, Life Technologies, 1:200) for 1 hour. Islets were imaged using the confocal settings on a Leica Stellaris STED FALCON microscope and a 40×/0.8 air objective.

### Room temperature correlative light and electron microscopy (CLEM)

The CLEM protocol was adapted from ref. 101. Briefly, INS-1 832/3 SNAP/FLAG-hGLP-1R cells were seeded on gridded coverslips (MatTek) and transfected with EGFP-VAPB before labelling with MitoTracker Red CMXRos and SNAP-Surface Alexa Fluor 647 and stimulation with 100 nM exendin-4 for 5 minutes. Cells were initially fixed for confocal microscopy with 4% PFA + 0.1% glutaraldehyde, and confocal stacks acquired with a C2+ confocal microscope (Nikon) and a 60x oil immersion objective. After fluorescence imaging, cells were additionally fixed for TEM with 4% PFA + 2.5% glutaraldehyde in sodium phosphate buffer. Samples were further processed according to a protocol modified from ref. 102 using $OsO_4$ and thiocarbohydrazide (TCH). In brief, samples were incubated in a 2% aqueous $OsO_4$ solution containing 1.5% potassium ferrocyanide and 2 mM $CaCl_2$ (30 minutes on ice), washed in 1% TCH, rinsed in $ddH_2O$ and incubated again in 2% $OsO_4$. After washing, samples were stained *en bloc* with 1% uranyl acetate (UA), rinsed in $ddH_2O$ and dehydrated in a graded series of ethanol. Samples were embedded into epoxy resin Embed812 (Science Services). After resin curing at 60 °C, the coverslip was removed by dipping into liquid nitrogen. The block was trimmed to a region identified during confocal imaging and serial sections with a thickness of 300 nm cut. Section post-staining was performed with 2% UA in methanol and 1% lead citrate in $ddH_2O$. After application of 15 nm colloidal gold beads, sections were imaged in a 300 kV Thermo Fisher F30 electron microscope. Tilt series were acquired at steps of 1° ranging from −63° to +63°. Tomograms were reconstructed with the IMOD software package[103]. Correlation of confocal images and tomogram slices was done with the EC-CLEM plugin[104] in Icy[105]. Segmentation of the tomograms was performed manually with Microscopy Image Browser[106]. 3D rendering of the segmentation was performed with ORS Dragonfly (www.theobjects.com/dragonfly/index.html).

### Cryo-CLEM

INS-1 832/3 SNAP/FLAG-hGLP-1R cells previously transfected with EGFP-VAPB were plated onto glow discharged (30 seconds, 20 mA) poly-L-lysine-coated Quantifoil R2/2 200 mesh gold EM grids (Electron Microscopy Sciences). Cells were labelled with 500 nM PKmito Orange FX (Spirochrome) and SNAP-Surface Alexa Fluor 647 before stimulation with 100 nM exendin-4 in complete media for 10 minutes. Grids were subsequently plunge-frozen in condensed ethane using a Vitrobot and stored in liquid $N_2$ for further analysis.

The clipped plunge-frozen grids were loaded into the Zeiss correlative cryo-holder which can be transferred into the cryo-LSM and the cryo-FIB-SEM without further removal from the holder. For cryo-confocal data acquisition, the cryo-holder was transferred onto the cryo-Linkam stage (Linkam Scientific) and mounted onto a ZEISS Axio Imager.Z2 fitted with a LSM900 confocal scan head equipped with 3 confocal channels, including the Airyscan 2 detector for super resolution. Low magnification objectives (C Epiplan-Apochromat 5×/0.2 NA DIC, C Epiplan-Apochromat 10×/0.4 NA DIC) were used for overviews, and an LD EC Epiplan-Neofluar 100×/0.75 NA DIC used for high resolution. Images were acquired using the following settings: 488 nm, 561 nm and 640 nm excitation with emission bands of 493–551 nm, 553–612 nm and 613–700 nm. Reflection and brightfield channels were additionally acquired for correlation of light microscopy with SEM images. Confocal images were processed with LSM Plus (linear Wiener deconvolution). For highest resolution, Airyscan detection was used with the following settings: 488 nm, 561 nm and 640 nm excitation with emission bands of 492–549 nm, 564–611 nm and 644–700 nm and images processed using the Airyscan joint deconvolution algorithm. For cryo FIB-SEM data acquisition, the correlative cryo-holder was transferred into the Zeiss Crossbeam 550 equipped with a Quorum cryo stage. The transfer is done through the Quorum Prep Chamber attached to the FIB-SEM chamber. Inside the chamber, the sample surface was sputter-coated with platinum with a current of 5 mA for 45 seconds to make the surface conductive for SEM imaging and FIB milling. After insertion, the FIB-SEM coordinate system was aligned with respect to the LSM dataset using the ZEN software (Zeiss) and ROIs acquired in the LSM relocated and imaged in the FIB-SEM. The sample was then tilted to 54° (perpendicular to the FIB) and a cold platinum deposition applied to minimize the curtaining effect (vertical stripes in the SEM images that result from uneven milling of the sample due to roughness of the sample surface). For this purpose, the gas injection system (GIS) nozzle was inserted at ~2.5 mm from the sample surface and the unheated GIS opened for about 60 seconds resulting in a protection layer with a thickness of ~0.5 µm. Trenches were milled at the ROIs with a FIB current of 3 nA (Gallium ion energy 30 keV) to generate cross-sections. FIB-SEM volume imaging was performed with a milling current of 300 pA (Gallium ion energy 30 keV) using a slice thickness of 30 nm to obtain SEM image stacks representing a volume of 16 × 12 × 9 µm (XYZ). The SEM images were acquired with an electron energy of 2.3 keV and an electron current of 30 pA with a pixel size of 8 nm using the InLens SE detector. The resulting FIB-SEM datasets were imported into ZEN project. After alignment of the image stack, images were processed using "destriping" (ZEN) based on the VSNR algorithm[107]. Processed image stacks were cropped to an appropriate size and a 3D alignment of the SEM and LSM stacks performed. A selected area in the aligned region of the FIB-SEM data was manually segmented, slice by slice, in Fiji, using the freehand selection tool. Outlines were saved as masks, and 3D reconstruction performed in Icy[105].

### Reactive oxygen species (ROS) assay

ROS was determined using the ROS-Glo™ $H_2O_2$ Assay (G8820, Promega) according to the manufacturer's instructions. Briefly, islets were incubated overnight in full media containing 1 µM thapsigargin ± 100 nM exendin-4 at 37 °C and subsequently incubated in 25 µM $H_2O_2$ Substrate diluted in $H_2O_2$ Substrate Dilution Buffer plus 1 µM thapsigargin ± 100 nM exendin-4 in a final volume of 100 µL for 3 hours at 37 °C in a 5% $CO_2$ incubator. Media (50 µL) was combined with 50 µL ROS-Glo™ Detection Solution in a 96-well opaque plate and incubated for 20 minutes at room temperature. Relative luminescence units (RLU) were recorded in a Flexstation 3 plate reader and results normalised to islet cell counts.

### Mitophagy (keima) assay

INS-1 832/3 cells were transfected with the dual-excitation ratiometric biosensor mKeima-Red-Mito-7 and seeded onto 35-mm glass-bottom dishes (MatTek). Cells were imaged 24 hours after seeding in RPMI-1640 buffer without phenol red ± 100 nM exendin-4 by time-lapse microscopy every 12 seconds for 11 minutes in a 37 °C heated chamber using a Leica Stellaris 8 inverted confocal microscope with a 40×/1.2 NA oil immersion objective with excitation wavelengths of 440 and 580 nm and emission wavelength of 620 nm. Ratio fluorescence intensities for the two excitation wavelengths (580/440 nm) was calculated to obtain a response trace as in ref. 108.

### ER and mitochondrial calcium assays

INS-1 832/3 cells were transfected with pCAG G-CEPIA1er (for ER calcium) or pCAG mito-RCaMP1h (for mitochondrial calcium)[109].

Twenty four hours after transfection, cells were washed 1× with PBS and imaged by time-lapse microscopy in RPMI-1640 without phenol red in a 40×/1.2 NA oil immersion objective Nikon Eclipse Ti microscope with an ORCA-Flash 4.0 camera (Hamamatsu) and Metamorph software (Molecular Devices). Mitochondrial calcium was imaged with 561 nm, and ER calcium with 488 nm excitation wavelengths, with images taken every 3 seconds. A baseline reading was recorded for 1 minute followed by 1 μM exendin-4 stimulation for 3 minutes. Raw fluorescent intensities were extracted using Fiji to obtain average intensity traces and AUCs calculated using GraphPad Prism 10.2.1.

## qPCR assays

**SPHKAP knock-down efficiency.** Mouse islets transduced with either Control or SPHKAP shRNA lentiviral particles were lysed using Trizol and RNA extracted using 200 μL chloroform per 1 mL Trizol and then precipitated in 500 μL isopropanol and 5 mg glycogen per mL of Trizol. RNA was washed with 75% EtOH before resuspension in ddH$_2$O. Two hundred and fifty ng of purified RNA were reverse transcribed into cDNA using the high-capacity cDNA reverse transcription kit (Applied Biosystems) and relative mRNA expression determined using FAST SYBR green chemistry qPCR and Applied Biosystems 7500 Real-time PCR system.

**Mitochondrial DNA copy number.** INS1 832/3 cells were seeded at 300,000 cells per well on a 24-well plate. Cells were stimulated ±100 nM exendin-4 for 30 minutes at 37 °C. Total DNA was extracted using 20 mM Tris pH 8.0, 5 mM EDTA, 400 mM NaCl, 1% SDS and 40 μg/mL proteinase K, as previously described[110], and used at 3 ng/mL for qPCR analysis of mitochondrial DNA (mtDNA) relative to genomic DNA (gDNA) as in ref. [111] by quantification of mitochondrial tRNA$^{Trp}$ *versus* nuclear *HK1* $C_T$, with relative mtDNA copy number estimated as $2 \times 2^{\Delta CT}$, and $\Delta C_T = HK1\ C_T$ - tRNA$^{Trp}$ $C_T$. Primers were designed using NCBI Primer Blast and validated in silico and then using a 1:2 standard curve of DNA ranging from 12 to 0.75 ng/mL, resulting in reaction efficiencies of 97– 102%.

qPCR primer sequences are included in Supplementary Table 4.

## Mitochondria-associated membrane (MAM) purification

$2 \times 10^7$ INS-1 832/3 SNAP/FLAG-hGLP-1R cells were seeded in a T75 flask in complete media and cultured overnight. Cells were then stimulated in complete media containing 100 nM exendin-4 for 10 minutes. The crude mitochondrial fraction containing ER-associated membranes was isolated using the mitochondria isolation kit for cultured cells (89874, Thermo Fisher Scientific) according to the manufacturer's protocol. MAMs were subsequently separated from mitochondria by ultracentrifugation at 95,000 × g for 30 minutes in an OptiPrep™ Density Gradient Medium (Sigma–Aldrich). The MAM fraction was diluted 1:10 in mitochondrial resuspension buffer (250 mM mannitol, 5 mM HEPES, 0.5 M EGTA, pH 7.4), pelleted by ultracentrifugation at 100,000 × g for 1 hour and prepared for western blot analysis as described above.

## pcDNA3-GFP11(x7)-VAPB cloning

pcDNA3-GFP11(x7)-VAPB was cloned *in house* from pEGFP-C1-hVAPB and pcDNA3-GFP11(x7)-Actin by PCR amplification of human VAPB with addition of 5′-BglII and 3′-XhoI restriction sites using Fwd 5′-GTACTCAGATCTCGAGCTCAAGC-3′ and Rev 5′- CTCATCTCGAGGAATTGCTACAAG-3′ primers. The second BglII site at position 7107 in pcDNA3-GFP11(x7)-Actin was removed by site directed mutagenesis using Fwd 5′- TGACGTCGACGGATCGGGAAATATCCCGATCCC-3′ and Rev 5′- GGGATCGGGATATTTCCCGATCCGTCGACGTCA-3′ primers with the Quick-Change Lightning kit (Agilent) prior to in frame cloning of human VAPB using T4 DNA ligase (Promega).

## Fluorescent Sensors Targeted to Endogenous Proteins (Fluo-STEP) assays of VAPB-localised cAMP and PKA signalling

The FluoSTEP system was used to determine VAPB-localised cAMP and PKA signalling as in ref. [25]. INS-1 832/3 cells were transfected with pcDNA3-GFP11(x7)-VAPB and pcDNA3.1(+)-FluoSTEP-ICUE (for cAMP) or pcDNA3.1(+)-FluoSTEP-AKAR (for PKA) at a 1:1 ratio. Cells were seeded onto glass-bottom dishes (MatTek) 24 hours before imaging in RPMI-1640 without phenol red + 10% FBS. Cells were imaged by time-lapse microscopy using a Zeiss LSM-780 inverted confocal laser-scanning microscope with a 63×/1.4 NA oil immersion objective and a 488 nm excitation wavelength and emissions at 508 and 583 nm for 2 minutes to record a baseline, followed by agonist stimulation at 100 nM for 5 minutes and 2-minute exposure to 10 μM FSK + 100 μM IBMX to generate a maximal response. FRET emission ratios were calculated and normalised to mean baseline values using GraphPad Prism 10.2.1.

## Quantification of ERMCSs per mitochondrial area using splitFAST

INS-1 832/3 cells were plated onto Matrigel (Gibco) coated glass-bottom imaging chambers (Ibidi) and transfected with 4xMTS-Halo, ER-RspA-NFAST and OMM-short-RspA-CFAST[50]. After 18 hours, cells were labelled with 10 nM JFX646 Halo ligand (kind gift from the Lavis lab, Janelia) in optiMEM (Gibco) for 3 minutes, followed by 3× washes in mKRB buffer (140 mM NaCl, 2.8 mM KCl, 2 mM MgCl$_2$, 1 mM CaCl$_2$, 10 mM HEPES, 10 mM glucose, pH 7.4) at 37 °C. The Lime splitFAST ligand (Twinkle Factory) was prepared at 5 μM in mKRB buffer and passed through a 0.22 μm filter before being added to cells. Exendin-4 was used at 1 μM in Lime splitFAST ligand in mKRB and added to cells for 10 minutes prior to imaging with an LSM980/Airyscan2 point-scanning confocal microscope (Zeiss) in LSMPlus mode. Simultaneous excitation was achieved using solid-state diode lasers at 488 nm/10 mW (0.9% max power) and 639 nm/7.5 mW (1.0% max power), with imaging conditions experimentally selected to minimise crosstalk. The resulting fluorescence was collected using a 63×/1.4 NA oil immersion objective and detected on a 34-channel spectral GaAsP array detector with the gain set at 650 V and split into two ranges of 490–632 nm and 641–694 nm. Samples were imaged at room temperature in a humidified imaging chamber with a Definite Focus module (Zeiss) employed for thermal drift correction and ZEN Blue v3.2 (Zeiss) software used for acquisition and deconvolution (LSMPlus function). The number of contact sites per mitochondrial area was quantified in Fiji. Firstly, the acquired images were Gaussian blurred (sigma radius 1.5) to aid in the generation of reliable downstream masks and puncta detection. An automated Otsu threshold was used to generate mitochondrial masks and quantitate mitochondrial area. These same masks were then applied to the splitFAST channel to ensure that only true contact sites (i.e. puncta on mitochondria) were counted in the analysis. A blinded count of total number of ERMCSs in each cell was taken and normalised to the mitochondrial area.

## Ethics statement

Our research complies with all relevant ethical regulations. Human islets were obtained from authorised islet isolation facilities at CEED and IIDP, with all ethical approvals in place and informed relative consent obtained. These islets have been refused for transplantation, have been quality-controlled, and meet specific criteria for research purposes. Mouse islets were purified from mice bred at the Central Biological Services unit of Imperial College London. All animal procedures were approved by the British Home Office under the UK animals (Scientific Procedures) Act 1986 (Project License number PP7151519 to Dr Aida Martinez-Sanchez, Imperial College London, UK) and from the local ethical committee (Animal Welfare and Ethics

Review Board) at the Central Biological Services unit of Imperial College London.

## Statistical analyses for cellular experiments

All statistical analyses and graph generation was performed using GraphPad Prism 10.2.1 (Dotmatics). Statistical tests used are indicated in the corresponding figure legends. The number of replicates per comparison represents biological replicates, with technical replicates within biological replicates averaged before statistical testing unless indicated. Data are presented as mean ± SEM. The $p$ value threshold for statistical significance was set at 0.05.

## Statistical analyses in human datasets

To compare the effects of *SPHKAP* gene region variants on T2D *versus* BMI, we used GWAS meta-analyses summary statistics from DIA-MANTE consortium (http://www.diagram-consortium.org) for T2D by Mahajan et al.[36], and from GIANT for BMI by Locke et al.[39]. We report SNPs from the region surrounding *SPHKAP* gene (Chr2:228,844,666-229,046,361, NCBI human genome build 37). Data pre-processing involved merging and aligning variant effects to T2D risk-increasing allele. We extracted $\beta$ values for BMI and T2D signals and calculated respective odds ratios (OR) for T2D using the formula OR = exp($\beta$). A scatter plot of SNPs with at least nominally significant ($p < 0.05$) effects on both BMI and T2D was generated using RStudio with the ggplot2 package, and Pearson correlation between BMI and T2D-associated signals calculated in R using the cor() function.

Statistical analysis of association between random blood glucose levels (RG) and coding DNA variants from whole-exome sequencing UKBB European ancestry individuals' data for *SPHKAP* gene variants was performed in PLINK2.0[112] assuming an additive genetic model. The model included covariates age, sex, time since last meal ($t$), accounted for as $t$, $t2$ and $t3$, and six principal components. Individuals were excluded if they had self-reported diabetes or diabetes medication, were pregnant, or had RG ≥20 mmol/L. Chromosome positions are given according to NCBI human genome build 38. The RG phenotype definition is provided in detail in ref. 113.

## Reporting summary

Further information on research design is available in the Nature Portfolio Reporting Summary linked to this article.

# Data availability

All data generated during the current study are included in this manuscript and/or its supplementary information files. Proteomics data is deposited in the ProteomeXchange (PRIDE) repository[114] with accession number PXD056782. The Source Data file included in this paper contains data from individual experiments used to generate the final figures. Source data are provided with this paper.

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

## Acknowledgements

The authors thank Dr Nicholas Vierra, University of Utah, for the SPHKAP antibodies; Prof Toshihiko Oka, Rikkyo University, for the MIC19-3xHA construct; Prof John D. Scott, Howard Hughes Medical Institute, for the SPHKAP-EGFP construct; Dr Emily Eden, UCL, for the EGFP-VAPB construct; the Filadi lab, CNR Institute of Neuroscience, for the ER-RspA-NFAST and OMM-short-RspA-CFAST splitFAST constructs; and the Avezov lab, UK Dementia Research Institute, for the 4xMTS-Halo construct. We thank Dr Aida Martinez-Sanchez, Imperial College London, for providing animal project licence access, the Imperial College Facility for Imaging by Light Microscopy (FILM) for technical support on light microscopy experiments, the Electron Microscopy Centre at the Centre of Structural Biology, Imperial College London, for support during TEM data acquisition, and Dr Ricardo Aramayo at the Electron Microscopy Facility of the MRC Laboratory of Medical Sciences (LMS) for support during cryo-CLEM sample preparation. The room temperature CLEM was supported by the Electron Microscopy and Histology Facility, a Core Facility of the CMCB Technology Platform at TU Dresden. We thank the EM facility of the Max Planck Institute of Molecular Cell Biology and Genetics for their services. We additionally thank the Cambridge Institute for Medical Research (CIMR) Microscopy Facility for their support with acquiring the splitFAST imaging data. The CIMR microscopy LSM980/Airyscan2 was funded by the MRC (MR/Y002172/1). The FILM Facility at Imperial College London is partially supported by funding from the Wellcome Trust (grant 104931/Z/14/Z) and BBSRC (grant BB/L015129/1). The NIDDK-funded Integrated Islet Distribution Program (IIDP) (RRID:SCR_014387) at City of Hope is supported by NIH Grant # 2UC4DK098085. The A.T. lab is funded by grants from Diabetes UK (19/0006094), the MRC (MR/X021467/1), and the Wellcome Trust (301619/Z/23/Z). G.A. is supported by a President's PhD Scholarship from Imperial College London, and A.I.O. by a Commonwealth PhD Scholarship. M.S. and A.Mü. are supported by the German Center for Diabetes Research (DZD e.V.) and the German Ministry for Education and Research (BMBF). The J.N.A. group is funded by a Wellcome Trust Career Development Award (227745/Z/23/Z). This research has been conducted using the UK Biobank Resource under Application Number 35327.

## Author contributions

Conceptualisation, A.T.; Methodology, A.T., E.G., I.P., E.M., J.N.A., A.Mü., A.M.; Investigation, A.T., G.A., A.I.O., L.E.E., M.Z., Y.M., P.P., H.C., S.B., E.M., E.G., A.M., A.Mü.; Formal analysis, A.T., G.A., A.I.O., L.E.E., M.Z., Y.M., E.G., M.A.d.R.B.F.L., Y.P., Z.B., I.P., S.J.M., J.N.A.; Visualisation, A.T., E.G., A.Mü., D.C.A.G., G.A.; Resources, A.T., K.B., M.S., B.J., D.J.W., I.P., J.N.A.; Writing—original draft, A.T.; Supervision and funding acquisition, A.T., I.P., J.N.A. All authors contributed to editing and reviewing the manuscript.

## Competing interests

M.A.d.R.B.F.L. and E.G. are employees of Abberior Instruments, the company manufacturing MINFLUX microscopes. B.J. has received grant funding and acted as a consultant for Metsera Inc. All other authors declare no competing interests.

## Additional information

**Gregory Austin**[1,17], **Affiong I. Oqua**[1,17], **Liliane El Eid**[1], **Mingli Zhu** ®[1], **Yusman Manchanda**[1], **Priyanka Peres**[2], **Helena Coyle**[2], **Yelyzaveta Poliakova** ® [3], **Zhanna Balkhiyarova**[4,5,6], **Karim Bouzakri**[7], **Alex Montoya** ® [8], **Dominic J. Withers** ® [8,9], **Michele Solimena** ® [10,11,12], **Ben Jones** ® [13], **Steven J. Millership**[1], **Steffen Burgold**[14], **David C. A. Gaboriau** ® [15], **Endre Majorovits**[14], **Evelyn Garlick** ® [16], **Maria Augusta do R. B. F. Lima**[16], **Inga Prokopenko** ® [4,5], **Jonathon Nixon-Abell** ® [2], **Andreas Müller** ® [10,11,12] & **Alejandra Tomas** ® [1] ✉

[1]Section of Cell Biology and Functional Genomics, Division of Diabetes, Endocrinology, and Metabolism, Department of Metabolism, Digestion and Reproduction, Imperial College London, London, UK. [2]Cambridge Institute for Medical Research, Department of Clinical Neurosciences, University of Cambridge,

Cambridge, UK. [3]School of Public Health, Imperial College London, London, UK. [4]Department of Clinical and Experimental Medicine, University of Surrey, Guildford, UK. [5]People-Centred Artificial Intelligence Institute, University of Surrey, Guildford, UK. [6]National Heart and Lung Institute, Imperial College London, London, UK. [7]Centre Européen d'Etude du Diabete, Strasbourg, France. [8]MRC Laboratory of Medical Sciences, London, UK. [9]Institute of Clinical Sciences, Faculty of Medicine, Imperial College London, London, UK. [10]Molecular Diabetology, University Hospital and Faculty of Medicine Carl Gustav Carus, TU Dresden, Dresden, Germany. [11]Paul Langerhans Institute Dresden, Helmholtz Center Munich, University Hospital Carl Gustav Carus and Faculty of Medicine, TU Dresden, Dresden, Germany. [12]German Center for Diabetes Research, Neuherberg, Germany. [13]Section of Endocrinology and Investigative Medicine, Division of Diabetes, Endocrinology, and Metabolism, Department of Metabolism, Digestion and Reproduction, Imperial College London, London, UK. [14]Carl Zeiss Microscopy GmbH, ZEISS Gruppe, Oberkochen, Germany. [15]Facility for Imaging by Light Microscopy, National Heart and Lung Institute, Imperial College London, London, UK. [16]Abberior Instruments GmbH, Göttingen, Germany. [17]These authors contributed equally: Gregory Austin, Affiong I. Oqua. ✉e-mail: a.tomas-catala@imperial.ac.uk

