## [Transparent Peer Review file · Nature Communications]

GLP-1R associates with VAPB and SPHKAP at ERMCSs to regulate β -cell mitochondrial remodelling and function

Corresponding Author: Professor Alejandra Tomas

Version 0:

Reviewer comments:

Reviewer #1

(Remarks to the Author)

In this manuscript titled "Inter-organelle contacts between endosomal GLP-1R, VAP-B, and SPHKAP trigger GLP-1R-induced MIC19 phosphorylation and β -cell mitochondrial remodelling", Gregory Austin et al. report that GLP-1R interacts with VAP-B at ER-mitochondria membrane contact sites to engage SPHKAP; in addition, the inter-organelle complex formed by GLP-1R, VAP-B and SPHKAP triggers PKA signalling that phosphorylates MIC19 to remold mitochondrial cristae. The findings shown in this manuscript are potentially interesting. However, the data in the manuscript are preliminary, and the overall novelty of the manuscript is not significant.

Major comments

1. The data related to the imaging and the role of inter-organelle contacts between endosome, ER, and mitochondria are not solid and unclear. The authors demonstrate that three proteins (GLP-1R, VAP-B, and SPHKAP) are highly connected, but these data do not convincingly represent the contacts between the three organelles. To address this, the authors should provide confocal imaging and transmission electron microscopy (TEM) data showing direct contacts between the three organelles under different conditions.
2. In Figure 6, exendin-4-induced mitochondrial remodelling in control and SPHKAP knockdown cells should be tested using transmission electron microscope (TEM). Additionally, it should be investigated whether Mic19 phosphorylation by PKA affects mitochondrial ultrastructure. The TEM data should be provided to support these findings.
3. The quality of some confocal images, such as in Figures 2e, 2f, and 4a, is poor. These images should be repeated using high-resolution confocal imaging to clearly show the colocalization or contacts between the two proteins.
4. In Figure 6g, 6h, 8c, 8d, the mitochondrial morphology should be re-quantified according to the classification in the published paper (Nature. 2023 Aug;620(7976):1101-1108. doi: 10.1038/s41586-023-06441-6.). "The morphological phenotypes were categorized as long tubular morphology, showing elongated and connected phenotype; fragmented phenotype, showing the mitochondrial network being completely broken apart into small network pieces; and short tubular/intermediate phenotype, showing a few elongated pieces and not as fragmented".
5. The authors demonstrate that "SPHKAP localisation to mitochondria depends on its binding to VAP-B via its FFAT motif". The authors should perform VAP-B knockout (KO) or knockdown (KD) experiments to further investigate the mitochondrial localization of SPHKAP.
6. In addition to activating PKA, the functions of the multi-organelle signaling platform formed by GLP-1R, VAP-B, and SPHKAP should be further investigated and discussed.
7. All data are from cell lines, and the physiological function of the multi-organelle signaling platform formed by GLP-1R, VAP-B, and SPHKAP in vivo is lacking and needs to be further investigated.
8. It has been reported that MIC60 and MIC19 are substrates of PKA (Mol Cell. 2016 May 5;62(3):371-384. doi: 10.1016/j.molcel.2016.03.037.) and that GLP-1R are highly associated with PKA (Elife. 2023 Nov 6;12:e80944. doi: 10.7554/eLife.80944.), which significantly diminishes the novelty of this manuscript.

Reviewer #2

(Remarks to the Author)

The therapeutic benefit of glucagon-like peptide-1 receptor agonists (GLP-1RAs) in treating type 2 diabetes are believed to be partially mediated by maintenance and restoration of mitochondrial function in pancreatic β -cells. In this manuscript, Austin et. al. sought to understand the mechanisms responsible for how GLP-1RAs modulate mitochondrial remodeling and turnover. They attempt to elucidate this mechanism by identifying protein binding partners of the GLP-1 receptor (GLP-1R) in vehicle versus 5-minute exendin-4 stimulated rat INS-1 832/3 β -cells constitutively expressing SNAP-FLAG-GLP-1R. Through gene ontology analysis, the identified interactors largely enrich for proteins located to the endomembrane system, cytosol, and endoplasmic reticulum, in addition to factors localized to both mitochondria, and the endoplasmic reticulum-mitochondria contact site (ERMCS).

The authors perform reciprocal pulldown, and biochemical and imaging based colocalization experiments to tie the mitochondrial effect of GLP-1R agonism to its interaction with VAP-B and SPHKAP at ERMCS. The authors demonstrate that the A Kinase Anchor Protein (AKAP) SPHKAP supported GLP-1R localization to ERMCS modulates localized Protein Kinase A (PKA) kinase activity at ERMCS. The mitochondrial contact site and cristae organizing system (MICOS) member MIC19 is next demonstrated to be phosphorylated in response to exendin-4 treatment in a SPHKAP/VAP-B dependent manner, with concurrent changes to mitochondrial morphology. The observations made in this paper, while interesting, are over-interpreted, and at times, very speculative. While the authors do show novelty in the exendin-4 mediated increased mitochondrial PKA activity, the subsequent MIC19 and mitochondrial morphology experiments are not robust and lack data to support the conclusions reached by the authors. The immunoprecipitation experiments used in the study are relied on heavily to support the claims made about the presence of a tripartite signaling hub at ERMCS, yet the results do not contain rigorous enough controls (similar loading amounts across samples, input and unbound fractions), and the authors do not account for the presence of western blot signal for their vehicle control treated samples throughout the paper, nor do they address the potential for protein interaction artefacts through the use of overexpression systems. Furthermore, there are several experiments in the paper lacking sufficient data points to reach the conclusions made by the authors, e.g., the mitochondrial morphology analyses do not have enough samples to make meaningful interpretation. In summary, the work in its current state is not up to the standards of Nature Communications.

Reviewer #3

(Remarks to the Author)

This is an excellent manuscript that elucidates a novel mechanism for endosomal GLP-1R signaling through an interaction with VAP-B at ER-localized ER-mitochondria membrane contact sites to regulate the activity of PKA through the mitochondrial A-kinase anchoring protein SPHKAP. They include genomic data demonstrating a role for SPHKAP in DM2 and BMI and demonstrate how this pathway can regulate mitochondrial function. I am not an expert on mitochondrial function, but the differences seem significant – the majority of my comments will focus on the GPCR-related work. But overall my comments are very minor – I think the approaches are complementary – I like the use of the MS interactome to identify potential targets, the use of confocal to identify the endosome/mitochondria interaction, the CoIPs demonstrating interactions at overexpressed and endogenous levels, the use of biosensors for assessing mitochondrial PKA activity, etc. Overall this is an excellent manuscript.

Major Comments:

- The increase in interaction with Vapb and Vapa and Figure 1 are orders of magnitude higher than any other signaling partners. Presumably this is largely due to the absence of any meaningful interaction at baseline, but it is still very striking. Could the authors comment on that?
- Figure 3 – it would be nice to see a quantification of internalization in Fig 3e if possible. In Fig 3f (quantified in Fig. 3g), it is not clear if any of the differences are statistically significant (many may not be if corrected for multiple comparisons). It is therefore challenging to compare the VAP-B IP with internalization in Fig. 3e.
- For the FFAT domain mutant, the FFAT motif is listed EFFDAxE in Fig 4f but the mutation is DFLTASE to DFLTESE (so not close to the 2 phenylalanines?). Some description of that mutation would be helpful as I found that confusing.
- A quantification of Fig. 5d would be nice as Ex-4 still seems to promote an interaction with SPHKAP-EGFP – just reduced compared to the Ctl RNAi conditions.
- Fig 8: The differences in mitochondrial mean form factor in area seem small – but I have no idea what to even compare it to.

Minor Comments:

Line 295 – “where” should be “were.”

Line 311 – “demonstrating that” would be better phrased as “consistent with.”

Line 350 – “englobing” could be replaced by “encompassing.”

Reviewer #4

(Remarks to the Author)

This study utilized an immunoprecipitation-based mass spectrometry approach to elucidate the interactome of GLP-1R in response to agonist stimulation. The researchers found evidence of a triple organelle contact involving endosomes, ER, and

mitochondria triggered by GLP-1R agonists. This contact subsequently activated Protein Kinase A signaling, leading to the phosphorylation of MIC19 and consequent mitochondria remodeling. The study highlighted the GLP-1R-VAPB-SPHKAP axis as being crucial for the restoration of mitochondrial function induced by GLP-1R agonists. Furthermore, the authors emphasized the potential association between SPHKAP variants and conditions like type 2 diabetes and obesity, underscoring the physiological significance of the GLP-1R-VAPB-SPHKAP axis in metabolic homeostasis maintenance. While the research provided valuable insights into the molecular mechanisms through which GLP-1R agonists influence mitochondrial function. However, the study lacked sufficient evidence to confirm the consistency of the observed phenotypes between in vitro and in vivo settings. All experiments were conducted using rodent cell lines and primary islets with RNAi knockdown, where the transient transfection efficiency in beta cells was noted to be relatively low, and the siRNA knockdown effects were deemed unstable. It is suggested that more rigorous and reasonable control groups be included in the study. Additionally, certain aspects warrant further exploration, such as the quantification of GLP-1RA positive endosomes, the functional dynamics of VAP-B and SPHKAP within the context of their intracellular localization in ER-mitochondria interactions in beta cells, the molecular mechanisms governing GLP-1RA-induced mitochondria remodeling, and whether the effects on insulin secretion are directly influenced by cAMP levels and calcium concentrations in the ER/mitochondria interface. Addressing these areas would strengthen the overall conclusions drawn from the research.

Major concerns

1. In Figure 2, a live-cell time-lapse imaging analysis is required to prove the exendin-4 dependent colocalization or association between the GLP-1R positive endosomes and ER.
2. The insulin secretion of pre-treatment WT control INS-1 832/3 β cell was lack in Fig2h-i and Fig7a, also the insulin secretion fold change under 11 mM glucose and Ex-4 was low.
3. The localization of GLP-1R internalization of certain agonists in primary islets as shown in Fig. 3e appears to resemble alpha cells rather than beta cells. Additionally, in Fig. 8a-b, the PLA signal in primary islets of endogenous GLP-1R with VAP-B and SPHKAP should be detected alongside insulin and glucagon as cell markers to determine localization more accurately. Furthermore, the role of alpha cells should be thoroughly discussed in relation to the findings presented. In addition, it is not clear whether the PLA signals show any changes in Ex-4 pre-treatment or RNAi primary islets.
4. About Figure 4a, the localization pattern of GLP-1R was different from that in Figure 2e. The GLP-1R exhibited a more clustered localization. The authors should check this out. Additionally, a live-cell time-lapse imaging analysis should be done to track the contact among endosome, ER and mitochondria.
5. About Figure 4g, it's interesting that a single point mutation of SPHKAP abolished its mitochondrial localization. The authors should provide stronger evidence to prove SPHKAP's mitochondrial localization depends on its interaction with VAPB through the FFAT domain.
6. Since VAPB is central in organizing mitochondria-ER contact, the authors should check out whether disruption of these protein pairs, GLP-1R-VAPB, VAPB-SPHKAP and GLP-1R-VAPB-SPHKAP, affects the inter-organelle contacts. Electron microscopy analysis is preferable to compare the contact between these organelles.
7. The internalized endosomes are usually destined to lysosomes or recycling endosomes. So how does this GLP-1R-VAPB-SPHKAP axis disassemble to release GLP-1R for recycling?
8. The inter-organelle contacts in mammalian cells are usually organized by several protein pairs, so disrupting one of them often results in nonsignificant affection. In this research, the authors emphasized the importance of SPHKAP in GLP-1R dependent β cell function and survival. It would be beneficial to check out if the overexpression of SPHKAP or GLP-1R-VAPB-SPHKAP axis enhances insulin secretion with or without agonist stimulation.

Minor concerns

1. In Line 159-160, it would be better to give a description of the enriched proteins, such as the total number, and the number of mitochondria localized proteins.
2. About Figure 2f, a statistical analysis of the colocalization efficiency between VAPB and GLP-1R was required.
3. In Figure 6h, the authors should specify how many cells were calculated in the legends. At least, 20 cells may be required to make the calculation solid.
4. About Extended Data Fig. 5d, a statistical analysis of the colocalization efficiency is required. Additionally, the authors may check out whether ER morphology was affected due to the disruption of SPHKAP.
5. About Figure 8, a western blot analysis is required to check out whether SPHKAP protein expression is disrupted or not in the islets by RNAi due to the difference of species.

Reviewer #5

(Remarks to the Author)

Version 1:

Reviewer comments:

Reviewer #1

(Remarks to the Author)

The revised manuscript quality has been improved, and most of my concerns were addressed and resolved.

Reviewer #2

(Remarks to the Author)

I appreciate the authors' attempts to address my previous concerns. I recognize that the authors have made a strong effort to improve their quantitation of changes to mitochondrial morphology using a standardized approach in multiple biological conditions, and these findings are of interest. I also note that there is robust data in this manuscript which supports the modulation of mitochondrial PKA activity by GLP1-R. However, I still believe that the authors have not performed an adequate job in controlling some of their experiments, and they have not sufficiently addressed my concerns regarding their co-IP results. I respect the authors attempt to address concerns through providing additional data with new techniques (CLEM), but I fear these efforts have confused their narrative, and raise more questions than they answer. Overall, I believe that while interesting observations are made, there is a significant number of experiments in this manuscript that raise significant doubts, which I will describe below. I still firmly believe that the observations made in this paper, while interesting, are over-interpreted, and at times, very speculative. Ultimately, the results in totality do not concretely support the model proposed, and thus I cannot support a Nature Communications publication.

Major Comments

Co-immunoprecipitation and blotting experiments

With regards to my previous comment about issues with the vehicle control co-IP samples, I will elaborate further. In their AE-MS experiment in Fig. 1 that sets up the study examining the GLP1-R and VAPB interaction, the differential in enrichment is highly significant, indicating the vehicle control sample immunoprecipitated little to no VAPB. Despite this result, the authors continue to show VAPB interaction in each of their co-IP vehicle control blots, suggesting contamination, and the IP of non-specific interactors. Furthermore, the authors do not include input, unbound, wash, and elution fractions in their blots, which would aid in establishing the efficiency of their enrichments, further complicating the interpretation of their results. Given that a significant amount of their data is based on transiently transfected overexpression systems, the authors do not control for the transfection efficiency, another factor which could influence the immunoprecipitation efficiency of their experiment. Finally, the authors perform their analyses based on measuring the changes between the amount of immunoprecipitated factor to binding partner. Given the fact that the authors have not isolated the variables listed above, this means of comparison is subject to a significant degree of variability. Given they have not isolated these variables above, for example by using the same AE-MS based technique they used in Fig. 1, the conclusions drawn from these experiments are unsubstantiated.

A major experiment that has not been performed but is essential to support the conclusions drawn is a co-immunoprecipitation of endogenous SPHKAP by VAPB in vehicle and exendin-4 treated cells, or a reciprocal pulldown of VAPB by SPHKAP.

I have the following issues with specific co-IP and blotting experiments in the manuscript:

- Supp. Fig. 2C: There is a very apparent unequal efficiency in immunoprecipitation. This experiment underlies a major conclusion about correlative GLP1R and VAPB interaction and should be repeated.
- Fig. 3A: The levels of TOM20 are uneven between samples, of which there are only 2. This therefore does not support the conclusion. Mitochondrial associated membrane isolation should be performed to confirm this.
- Supp. Fig. 6A: There are unequal amounts of input across samples and no apparent differences in interactor levels.
- Supp. Fig. 6D: Unequal IP levels of GLP1R.
- Supp. Fig. 6F,H: Tubulin amounts are different across samples.
- Supp. Fig. 7I: There is very unequal MIC19-3XHA immunoprecipitated. The comparison of phospho level to the IP MIC19 is inaccurate. The conclusion drawn that it is specifically related to SPHKAP and VAPB is therefore unsubstantiated.
-

Colocalization

Another significant issue I have with the results is the almost complete absence of quantitative colocalization analysis. The authors provide a few instances of colocalization analysis using Manders' coefficient, yet this same quantitation is not performed elsewhere. This analysis is required to support the claims made by the authors. Below are some instances where this would be required to support the conclusions made:

- Fig. 1F: This experiment should show vehicle and exendin-4 treated cells. The authors should use their SPLIC ERMCS tag to quantify the colocalization of three channels to support the claim that it is occurring at ERMCS in exendin-4 treated cells. A minimum of 20 regions of interest (ROI) is required for a quantitative change such as this.
- Fig. 1H: Similarly, this colocalization needs to be quantified. The magnification is too low to draw the conclusion of an interaction here. Vehicle controls should be quantified for statistical analysis.
- Supp. Fig. 1F: colocalization analysis is required along with vehicle control.
- Fig. 3B: colocalization analysis is required along with vehicle control.
- Fig. 3C: colocalization analysis is required along with vehicle control. Higher magnification required.
- Fig. 3D: colocalization analysis is required along with vehicle control. What is the expected distance between the ER and Mito for an interaction to support signaling as is suggested? They appear too distant to draw this conclusion.
- Supp. Fig. 4C: This shows a clear gap between endosome and mitochondrion, and does not support the conclusion that they are interacting.
- Fig. 4C,D: Why is the SPLIC ERMCS construct not used to determine the localization to ERMCS? Also, the signal for

SPHKAP is much lower than other examples in the paper. This result is therefore not reliable. Additionally, the authors claim that SPHKAP is part of a network associated with, but not inside mitochondria. I am unsure how they draw this conclusion based on this low resolution image. The authors would have to use their SPLIC system, or perform enrichment of mitochondrial associated membranes using density gradient ultracentrifugation. Colocalization quantification of 4D would also be required to support its association with the ER.

- Supp. Fig. 7B: It is unclear how this image is demonstrating colocalization. Colocalization analysis is required along with vehicle control.
- Supp Fig. 8A: The authors do not elaborate on how this is demonstrating mitophagy, as the image does not show engulfment of mitochondria. Given the absence of any known non-receptor mediated (PARKIN) or receptor mediated (BNIP3/NIX/FUNDC1) mitophagy markers on light microscopy experiments, the conclusions reached are incredibly speculative.
- Supp. Fig. 8B: How is the DRP1 mCherry expression controlled for? DRP1 should be cytosolic, whereas this image shows almost a complete absence of signal in the red channel, suggesting low transfection levels, or insufficient laser power. This is not a fair comparison between control and treated samples, and does not support the conclusion of recruitment being specific to the experimental condition.

Minor Comments

- The MIC19 phosphorylation result is overinterpreted. While it is an interesting observation, the blot appears preliminary, and the blot in the supplement has uneven loading. The conclusions drawn from this are speculative at best. I highly suggest that the authors tone down the language used in interpreting this observation.
- Fig. 1B: The overlapping names of each point in this plot make it almost completely indiscernible.

Reviewer #3

(Remarks to the Author)

The authors have fully addressed my concerns.

Reviewer #4

(Remarks to the Author)

The authors have effectively addressed the previous concerns and implemented substantial improvements, which have significantly elevated the quality and credibility of the study. However, there are a few minor issues that could further strengthen the manuscript.

1. Colocalization analysis in Figures 4c and 4d: A colocalization analysis should be conducted and presented for Figures 4c and 4d. This would provide more in-depth insights into the relationships between SPHKAP localization and the defects of mitochondria.
2. It's pretty interesting that SPHKAP exhibited ERMCS-like localization pattern. The mitochondria-associated ER membranes could be purified and detected by western blot to further verify its localization.
3. Effect of SPHKAP overexpression on ERMCS: The knockdown of SPHKAP was shown to eliminate the disruptive effect of Ex-4 on ERMCS. It would be valuable for the authors to investigate whether the overexpression of SPHKAP has any impact on ERMCS. This additional experiment could further clarify the role of SPHKAP in the regulation of ERMCS and enhance the comprehensiveness of the study.

Reviewer #5

(Remarks to the Author)

Version 2:

Reviewer comments:

Reviewer #2

(Remarks to the Author)

The revised manuscript quality has been improved and most of my concerns were addressed and resolved.

Reviewer #4

(Remarks to the Author)

The revised manuscript quality has been improved, and most of my concerns were addressed and resolved.

Reviewer #5

(Remarks to the Author)

Response to Reviewers - Austin *et al.*, GLP-1R associates with VAPB and SPHKAP at ERMCSs to regulate β -cell mitochondrial remodelling and function.

We would like to thank all the reviewers for their thorough evaluation of our manuscript. We have addressed their points in full, including with several new experiments, as explained below.

Reviewer #1 (Remarks to the Author):

In this manuscript titled "Inter-organelle contacts between endosomal GLP-1R, VAP-B, and SPHKAP trigger GLP-1R-induced MIC19 phosphorylation and β -cell mitochondrial remodelling", Gregory Austin *et al.* report that GLP-1R interacts with VAP-B at ER-mitochondria membrane contact sites to engage SPHKAP; in addition, the inter-organelle complex formed by GLP-1R, VAP-B and SPHKAP triggers PKA signalling that phosphorylates MIC19 to remold mitochondrial cristae. The findings shown in this manuscript are potentially interesting. However, the data in the manuscript are preliminary, and the overall novelty of the manuscript is not significant.

Major comments

1. The data related to the imaging and the role of inter-organelle contacts between endosome, ER, and mitochondria are not solid and unclear. The authors demonstrate that three proteins (GLP-1R, VAP-B, and SPHKAP) are highly connected, but these data do not convincingly represent the contacts between the three organelles. To address this, the authors should provide confocal imaging and transmission electron microscopy (TEM) data showing direct contacts between the three organelles under different conditions.

This point has been addressed with the following experiments:

Fig. 3b,c; Supplementary Movie 2: Confocal microscopy colocalisation, including by time-lapse, between SNAP/FLAG-hGLP-1R (labelled at the cell surface with membrane-impermeable SNAP-Surface 647 prior to exendin-4 stimulation) with the ER-mitochondria contact site (ERMCS) fluorescent marker SPLICS Mt-ER Long P2A in INS-1 832/3 cells.

Fig. 3d-f: Correlative light and electron microscopy (CLEM) analysis of a SNAP/FLAG-hGLP-1R-positive endosome (labelled as in Fig. 3b) docked to EGFP-VAPB-positive ER at an ERMCS.

Fig. 3g: TEM imaging of a SNAP/FLAG-hGLP-1R-positive endosome (labelled at the cell surface with membrane-impermeable SNAP-biotin + streptavidin conjugated to 5-nm gold followed by exendin-4 stimulation) docked to the ER and mitochondria in ERMCSs in primary mouse islets transduced with a SNAP/FLAG-hGLP-1R-expressing adenovirus.

Supplementary Fig. 4: Cryo-CLEM (high-resolution confocal microscopy + cryo-FIB-SEM) analysis of a SNAP/FLAG-hGLP-1R-positive endosome (labelled as in Fig. 3b) docked to EGFP-VAPB-positive ER in close proximity to a mitochondrion in INS-1 832/3 cells.

2. In Figure 6, exendin-4-induced mitochondrial remodelling in control and SPHKAP knockdown cells should be tested using transmission electron microscope (TEM). Additionally, it should be investigated whether Mic19 phosphorylation by PKA affects mitochondrial ultrastructure. The TEM data should be provided to support these findings.

The TEM experiment requested by the reviewer is now shown in Fig. 7a,b,c.

Regarding MIC19 effects, while there is previously published data indicating that PKA-dependent MIC19 phosphorylation is involved in regulation of mitophagy (PMID: 27153535), and MIC19 deficiency has been shown to result in abnormal mitochondrial morphology (PMID: 27317679, PMID: 37473754, PMID: 31097788, PMID: 28526561), we would like to highlight that our study does not imply that all the effects of ERMCS-localised GLP-1R-dependent PKA signalling in mitochondria will be due solely to MIC19 phosphorylation. In fact, our interactome has identified a range of other targets for the GLP-1R in both ERMCS and mitochondria, suggesting that a number of concerted

effects are taking place downstream of ERMCS-localised GLP-1R signalling, with MIC19 phosphorylation being one identified downstream target. We have now included a note in the discussion to clarify this point as follows:

“Overall, all these MS-identified interactions suggest that the effect of GLP-1R signalling at ERMCS is likely to extend well beyond MIC19 phosphorylation into a more comprehensive programme of regulation of ER, mitochondria and lipid metabolic pathways.”

3. The quality of some confocal images, such as in Figures 2e, 2f, and 4a, is poor. These images should be repeated using high-resolution confocal imaging to clearly show the colocalization or contacts between the two proteins.

Further confocal microscopy experiments, including time-lapse imaging, to image GLP-1R-VAPB and GLP-1R-ERMCS co-localisation have now been performed and included in Fig. 1h, Fig. 3c, Supplementary Fig. 1d, Supplementary Movie 1 and Supplementary Movie 2.

4. In Figure 6g, 6h, 8c, 8d, the mitochondrial morphology should be re-quantified according to the classification in the published paper (Nature. 2023 Aug;620(7976):1101-1108. doi: 10.1038/s41586-023-06441-6.). “The morphological phenotypes were categorized as long tubular morphology, showing elongated and connected phenotype; fragmented phenotype, showing the mitochondrial network being completely broken apart into small network pieces; and short tubular/intermediate phenotype, showing a few elongated pieces and not as fragmented”.

We have taken on board the suggestion of the reviewer and quantified all the previous and new mitochondrial morphology data as in von der Malsburg *et al.*, Nature 2023. These quantifications can be found in Fig. 6a,b,c, Fig. 9a,b,c,d, and Fig. 10d.

5. The authors demonstrate that “SPHKAP localisation to mitochondria depends on its binding to VAP-B via its FFAT motif”. The authors should perform VAP-B knockout (KO) or knockdown (KD) experiments to further investigate the mitochondrial localization of SPHKAP.

This experiment has now been performed and is shown in Fig. 4c. Further experiments to clarify the localisation of SPHKAP, which we now show to be at ERMCSs rather than inside mitochondria, are included in Fig. 4d and Supplementary Fig. 6c.

6. In addition to activating PKA, the functions of the multi-organelle signaling platform formed by GLP-1R, VAP-B, and SPHKAP should be further investigated and discussed.

We have performed a range of experiments to investigate the functional effects of SPHKAP-dependent ERMCS-localised GLP-1R-induced PKA activity, including:

Fig. 5a, effect on mitochondrial membrane potential in INS-1 832/3 cells

Fig. 5b, Supplementary Fig. 7d, effect on ATP generation in INS-1 832/3 cells

Fig. 5c, Supplementary Fig. 7e,f, effect on insulin secretion in INS-1 832/3 cells

Fig. 5d, effect on cell survival to ER stress in INS-1 832/3 cells

Fig. 5f, effect on mitophagy in INS-1 832/3 cells

Fig. 5h, effect on Drp1 S616 phosphorylation in INS-1 832/3 cells

Fig. 6c,d,e, Fig. 7a,b,c, Supplementary Fig. 8d,e, effect on mitochondrial remodelling in INS-1 832/3 cells

Fig. 8b, effect on ATP generation in mouse primary islets

Fig. 8c, effect on ROS generation in mouse primary islets

Fig. 8d, effect on insulin secretion in mouse primary islets

Fig. 9c,d, Supplementary Fig. 9e, effect on mitochondrial remodelling in mouse primary islets

Supplementary Fig. 7g,h, effect on ER and mitochondrial calcium in INS-1 832/3 cells

Supplementary Fig. 7j, effect on mitochondrial DNA copy number in INS-1 832/3 cells

7. All data are from cell lines, and the physiological function of the multi-organelle signaling platform formed by GLP-1R, VAP-B, and SPHKAP in vivo is lacking and needs to be further investigated.

Data in primary mouse islets is shown in Fig. 3g; Fig. 8a-d; Fig. 9a-d; Supplementary Fig. 9a-e; Supplementary Fig. 5b.

Data in primary human islets is shown in Fig. 10a-d.

Data from human genetic studies is shown in Supplementary Fig. 5c-f.

Experiments in living mice are outside of the scope of the current study, where we have identified for the first time the existence of an ERMCS-localised GLP-1R signalling complex and determined its functional importance in β -cell lines and primary islets. Follow up studies on a β -cell-specific conditional SPHKAP KO mouse model are warranted and will be pursued by our lab in subsequent studies.

8. It has been reported that MIC60 and MIC19 are substrates of PKA (Mol Cell. 2016 May 5;62(3):371-384. doi: 10.1016/j.molcel.2016.03.037.) and that GLP-1R are highly associated with PKA (Elife. 2023 Nov 6;12:e80944. doi: 10.7554/eLife.80944.), which significantly diminishes the novelty of this manuscript.

The GLP-1R, as a $G\alpha_s$ -coupled GPCR, is well known to signal via cAMP generation and PKA activation. There is however very little information on the spatiotemporal organisation of this signalling. Efforts are being pursued by many leading labs to define the nature and regulation of subcellular signalling hubs, or nanodomains, downstream of GPCR-induced cAMP generation and PKA activation (PMID: 33334857, PMID: 38310024, PMID: 37075112), including for the GLP-1R (35294858). The existence of multiprotein complexes that define the architecture of these nanodomains, with signalling restricted to complex-associated downstream effectors is inferred, but their nature is largely unknown. Much less is currently known about the assembly of biomolecular condensates at inter-organelle membrane contact sites, so that their signalling would enable the transfer of information and signal coordination across organelles and even sub-organelle regions. The study of the nature and components of membrane contact sites is in itself a hot topic for research, with many unknown areas regarding their composition and the mechanisms involved in regulating their action. Moving to the specialised pancreatic β -cells, these areas are even more understudied, including defining the importance of GLP-1R-induced signalling hubs in the regulation of β -cell survival and insulin secretion, their influence in the pathophysiology of type 2 diabetes and related metabolic disturbances, and their role in enabling metabolic adaptation in the face of changing nutritional and glucolipotoxic stress conditions. We believe that ours is a seminal study that identifies and characterises the function of a hitherto unknown signalling complex at ERMCS that contributes to define the spatiotemporal organisation of signalling of the GLP-1R in pancreatic β -cells and is likely to be applicable to other $G\alpha_s$ -coupled GPCRs engaging SPHKAP or similarly FFAT-motif containing AKAPs at VAPB-defined ERMCS to control mitochondrial function, with potential further roles in lipid metabolism which we plan to investigate in future studies.

Reviewer #2 (Remarks to the Author):

The therapeutic benefit of glucagon-like peptide-1 receptor agonists (GLP-1RAs) in treating type 2 diabetes are believed to be partially mediated by maintenance and restoration of mitochondrial function in pancreatic β -cells. In this manuscript, Austin et. al. sought to understand the mechanisms responsible for how GLP-1RAs modulate mitochondrial remodeling and turnover. They attempt to elucidate this mechanism by identifying protein binding partners of the GLP-1 receptor (GLP-1R) in vehicle versus 5-minute exendin-4 stimulated rat INS-1 832/3 β -cells constitutively expressing SNAP-FLAG-GLP-1R. Through gene ontology analysis, the identified interactors largely enrich for proteins located to the endomembrane system, cytosol, and endoplasmic reticulum, in addition to factors localized to both mitochondria, and the endoplasmic reticulum-mitochondria contact site (ERMCS).

The authors perform reciprocal pulldown, and biochemical and imaging based colocalization experiments to tie the mitochondrial effect of GLP-1R agonism to its interaction with VAP-B and SPHKAP at ERMCS. The authors demonstrate that the A Kinase Anchor Protein (AKAP) SPHKAP supported GLP-1R localization to ERMCS modulates localized Protein Kinase A (PKA) kinase activity at ERMCS. The mitochondrial contact site and cristae organizing system (MICOS) member MIC19 is next demonstrated to be phosphorylated in response to exendin-4 treatment in a SPHKAP/VAP-B dependent manner, with concurrent changes to mitochondrial morphology. The observations made in this paper, while interesting, are over-interpreted, and at times, very speculative. While the authors do show novelty in the exendin-4 mediated increased mitochondrial PKA activity, the subsequent MIC19 and mitochondrial morphology experiments are not robust and lack data to support the conclusions reached by the authors. The immunoprecipitation experiments used in the study are relied on heavily to support the claims made about the presence of a tripartite signaling hub at ERMCS, yet the results do not contain rigorous enough controls (similar loading amounts across samples, input and unbound fractions), and the authors do not account for the presence of western blot signal for their vehicle control treated samples throughout the paper, nor do they address the potential for protein interaction artefacts through the use of overexpression systems.

Regarding the co-immunoprecipitation experiments and associated controls:

- a) Besides always loading the same volume per lane, loading amounts are inherently controlled in our experiments by blotting for SNAP after stripping the same membranes previously blotted for the co-IP factor analysed in each experiment. We always show this SNAP blot in all our co-immunoprecipitations and our quantifications are always normalised to the level of SNAP/FLAG-hGLP-1R signal per lane. This ensures that any potential differences in sample loading or in the amount of SNAP/FLAG-hGLP-1R immunoprecipitated per sample are considered when quantifying the amount of factor co-immunoprecipitated with SNAP/FLAG-hGLP-1R under the different conditions analysed in our experiments.
- b) As requested by the reviewer, input and output fractions, demonstrating the effectiveness of our IPs to pull down SNAP/FLAG-hGLP-1R, are now shown in Supplementary Fig. 1c and Supplementary Fig. 6a.
- c) Regarding vehicle controls, these are always included in all our co-IP experiments. Our conclusions are based on measurements of the change between the amount of factor co-immunoprecipitated in exendin-4 stimulated over vehicle conditions (see quantifications in Fig. 1d, Fig. 1e, and Fig. 4b, as well as all co-IP blots throughout the paper, which all include a vehicle control so that levels in vehicle *versus* exendin-4-stimulated conditions can be compared). We therefore respectfully disagree with the reviewer that “we do not account for the presence of western blot signal for vehicle control treated samples throughout the paper”.

Beyond co-IP experiments, VAPB-localised GLP-1R-induced PKA activity is now verified, and shown to be dependent on SPHKAP, by our FluoSTEP assays in Fig. 4h.

We additionally demonstrate the docking of SNAP/FLAG-hGLP-1R-positive endosomes to EGFP-VAPB-positive ER at ERMCSs by CLEM (Fig. 3d,e,f), and cryo-CLEM (Supplementary Fig. 4a-c), as well as the association of SNAP/FLAG-hGLP-1R with ERMCSs by confocal and time-lapse microscopy (Fig. 3b,c; Supplementary Movie 2).

Increased interaction of exendin-4-stimulated SNAP/FLAG-hGLP-1R with endogenous VABP, including quantification, is shown in Fig. 1e, and for endogenous SPHKAP is shown in Supplementary Fig. 6b.

Interactions between endogenous GLP-1R, SPHKAP and VAPB in primary mouse and human islets are verified using proximity ligation assays (PLAs) shown in Fig. 8a, Supplementary Fig. 9a,b, Fig. 10a,b.

Furthermore, there are several experiments in the paper lacking sufficient data points to reach the conclusions made by the authors, e.g., the mitochondrial morphology analyses do not have enough samples to make meaningful interpretation. In summary, the work in its current state is not up to the standards of Nature Communications.

Mitochondrial morphology effects have now been analysed in depth in Fig. 6a-e; Fig. 7a,b; Fig. 9a-d; Fig. 10c,d; Supplementary Fig. 8d,e; and Supplementary Fig. 9d,e. Additionally, we now include data on mitophagy (Fig. 5f; Supplementary Fig. 8a,b) and recruitment of Drp1 to mitochondria (Fig. 5g; Supplementary Fig. 8c; Supplementary Movies 3 and 4).

Please note that the n number included in the quantifications of changes in mitochondrial morphology refer to the number of independent biological repeats performed rather than the number of cells quantified. For INS-1 832/3 cells and dispersed islet cell experiments, we have quantified a total of 30 to 130 cells per experiment and condition, imaged from $n=4-5$ independent repeats, with 3 different fields of view imaged per repeat; For intact islet experiments, we have quantified all visible cells in 15 to 30 islets per experiment and condition, taken from $n=5-6$ independent repeats (meaning $n=5-6$ batches of islets from separate mice). We have added this information to the paper methods to clarify our sample sizes.

We hope that this thoroughly revised version of our manuscript, including many new experiments beyond the examination of mitochondrial morphology effects downstream of GLP-1R signalling in β -cells, has contributed to increase the enthusiasm of the reviewer towards our data.

Reviewer #3 (Remarks to the Author):

This is an excellent manuscript that elucidates a novel mechanism for endosomal GLP-1R signaling through an interaction with VAP-B at ER-localized ER-mitochondria membrane contact sites to regulate the activity of PKA through the mitochondrial A-kinase anchoring protein SPHKAP. They include genomic data demonstrating a role for SPHKAP in DM2 and BMI and demonstrate how this pathway can regulate mitochondrial function. I am not an expert on mitochondrial function, but the differences seem significant – the majority of my comments will focus on the GPCR-related work. But overall my comments are very minor – I think the approaches are complementary – I like the use of the MS interactome to identify potential targets, the use of confocal to identify the endosome/mitochondria interaction, the CoIPs demonstrating interactions at overexpressed and endogenous levels, the use of biosensors for assessing mitochondrial PKA activity, etc. Overall this is an excellent manuscript.

Major Comments:

- The increase in interaction with Vapb and Vapa and Figure 1 are orders of magnitude higher than any other signaling partners. Presumably this is largely due to the absence of any meaningful interaction at baseline, but it is still very striking. Could the authors comment on that?

We agree with the reviewer that the large difference in VAPA/B interaction with SNAP/FLAG-hGLP-1R in vehicle *versus* exendin-4-stimulated conditions probably reflects very little interaction between VAPA/B and SNAP/FLAG-hGLP-1R under vehicle conditions, or alternatively a very prominent increase of this interaction following exendin-4 stimulation. This result is very reassuring as it indicates that the interaction is indeed happening between VAPA/B and active receptor, and not with the nascent receptor polypeptide being made in the ER during biosynthesis. We also assume that the MS analysis is a lot more sensitive than our co-IP + Western blot assays, which are known for their semiquantitative capacity associated with saturation of signal, which makes the difference between vehicle and stimulated conditions, despite being readily quantifiable and significant, appear much less pronounced than in our MS interactome data.

We have included a comment in the discussion to highlight that we have likely identified the main site of action for endosomal GLP-1R:

“...finding not only that the receptor engages with ER-endosome MCSs to organise its signalling, but that the main constituents of these inter-organelle contacts, namely the VAPs, are by a considerable margin the most prominent intracellular binding partners of active GLP-1R. This suggests that engagement of VAPs, and subsequent generation of ER-localised GLP-1R signalling, is not just one of many pathways engaged by the receptor, but the main site of action of GLP-1R following its internalisation.”

- Figure 3 – it would be nice to see a quantification of internalization in Fig 3e if possible. In Fig 3f (quantified in Fig. 3g), it is not clear if any of the differences are statistically significant (many may not be if corrected for multiple comparisons). It is therefore challenging to compare the VAP-B IP with internalization in Fig. 3e.

As requested by the reviewer, quantification of leftover surface receptor levels (as a proxy for receptor internalisation) is now included in Fig. 2c for $n=3$ islet batches from separate mice. Regarding the comparisons between receptor internalisation and VAPB co-IP results with the panel of GLP-1R agonists (now shown in Fig. 2d), we agree with the reviewer that there is some variability in the co-IP data; however, when taken as a whole, we can clearly see an inverse correlation between leftover surface receptor levels and co-immunoprecipitated VAPB. We have included a sentence in the results to explain this correlation, rather than claiming individual effects per agonist for the co-IP results:

“VAPB co-immunoprecipitation using the panel of GLP-1RAs described above unveiled a correlation between the level of GLP-1R–VAPB binding and the degree of receptor internalisation elicited by each of the tested GLP-1RAs.”

- For the FFAT domain mutant, the FFAT motif is listed EFFDAXE in Fig 4f but the mutation is DFLTASE to DFLTESE (so not close to the 2 phenylalanines?). Some description of that mutation would be helpful as I found that confusing.

An explanation about the rationale behind the chosen mutation to disturb SPHKAP binding to VAPB via its pFFAT motif is now included in the manuscript as follows: “We next generated a mutant version of SPHKAP-EGFP harbouring a point mutation in its pFFAT motif (SPHKAP-EGFP A215E, DFLTASE to DFLTESE). This strategy was based on previous structural data showing that position 5 of the FFAT core motif is positioned within a hydrophobic pocket in the MSP domain of VAP (PMID: 16004875), with previous studies indicating that introducing a negative charge in this position causes a steric hindrance which effectively disrupts FFAT motif-VAP interaction (PMID: 35611050).”

- A quantification of Fig. 5d would be nice as Ex-4 still seems to promote an interaction with SPHKAP-EGFP – just reduced compared to the Ctl RNAi conditions.

This quantification has now been performed and is shown in Fig. 4b.

- Fig 8: The differences in mitochondrial mean form factor in area seem small – but I have no idea what to even compare it to.

As explained above for reviewer 1, we have re-analysed existing data, and added substantial new data, using the classification method from von der Malsburg *et al.*, Nature 2023. New quantifications can be found in Fig. 6a-c, Fig. 9a-d, and Fig. 10d, in addition to assessments of mitochondrial mean area and form factor shown in Supplementary Fig. 8d,e, and Supplementary Fig. 9e, and quantification of TEM data shown in Fig. 7b.

Minor Comments:

Line 295 – “where” should be “were.” This text has now been modified.

Line 311 – “demonstrating that” would be better phrased as “consistent with.” This text has now been modified.

Line 350 – “englobing” could be replaced by “encompassing.” This text has now been modified.

Reviewer #4 (Remarks to the Author):

This study utilized an immunoprecipitation-based mass spectrometry approach to elucidate the interactome of GLP-1R in response to agonist stimulation. The researchers found evidence of a triple organelle contact involving endosomes, ER, and mitochondria triggered by GLP-1R agonists. This contact subsequently activated Protein Kinase A signaling, leading to the phosphorylation of MIC19 and consequent mitochondria remodeling. The study highlighted the GLP-1R-VAPB-SPHKAP axis as being crucial for the restoration of mitochondrial function induced by GLP-1R agonists. Furthermore, the authors emphasized the potential association between SPHKAP variants and conditions like type 2 diabetes and obesity, underscoring the physiological significance of the GLP-1R-VAPB-SPHKAP axis in metabolic homeostasis maintenance. While the research provided valuable insights into the molecular mechanisms through which GLP-1R agonists influence mitochondrial function. However, the study lacked sufficient evidence to confirm the consistency of the observed phenotypes between in vitro and in vivo settings. All experiments were conducted using rodent cell lines and primary islets with RNAi knockdown, where the transient transfection efficiency in beta cells was noted to be relatively low, and the siRNA knockdown effects were deemed unstable. It is suggested that more rigorous and reasonable control groups be included in the study.

We have chosen an RNAi-based knockdown strategy to avoid functional adaptation during selection of VAPB or SPHKAP CRISPR/Cas9 KO subclones. This is particularly relevant in our study as we are investigating mechanisms of mitochondrial rewiring that are prone to functional adaptation so that the phenotype might become masked by the period of cell selection required. Knockdown efficiency in INS-1 832/3 cells is ~69% for VAPB (as quantified in Supplementary Fig. 1g) and ~65% for SPHKAP (as quantified in Supplementary Fig. 6g). We are not sure what the reviewer is referring to when stating: “siRNA knockdown effects were deemed unstable”: in this revised manuscript we show significant effects of VAPB knockdown on GLP-1R-induced insulin secretion (Fig. 1i, Supplementary Fig. 1h), and SPHKAP co-IP with SNAP/FLAG-hGLP-1R (Fig. 4a,b); as well as significant effects of SPHKAP knockdown on GLP-1R-induced mitochondrial PKA activity (Fig. 4g), mitochondrial membrane potential (Fig. 5a) , ATP generation [in INS-1 832/3 cells (Fig. 5b, Supplementary Fig. 7d) and islets (Fig. 8b)], insulin secretion (Fig. 5c, Supplementary Fig. 7e), cell survival to ER stress (Fig. 5d), mitophagy (Fig. 5f), mitochondrial remodelling [in INS-1 832/3 cells (Fig. 6c-e, Supplementary Fig. 8e) and islets (Fig.9c,d, Supplementary Fig. 9e)], and ROS levels in thapsigargin-exposed islets (Fig. 8c). Finally, we also show a significant effect of SPHKAP knockdown on GLP-1R-induced PKA activation at VAPB loci with our adapted FluoSTEP biosensor assay (Fig. 4h).

Additionally, certain aspects warrant further exploration, such as the quantification of GLP-1RA positive endosomes, the functional dynamics of VAP-B and SPHKAP within the context of their intracellular localization in ER-mitochondria interactions in beta cells, the molecular mechanisms governing GLP-1RA-induced mitochondria remodeling, and whether the effects on insulin secretion are directly influenced by cAMP levels and calcium concentrations in the ER/mitochondria interface. Addressing these areas would strengthen the overall conclusions drawn from the research.

See answers below.

Major concerns

1. In Figure 2, a live-cell time-lapse imaging analysis is required to prove the exendin-4 dependent colocalization or association between the GLP-1R positive endosomes and ER.

As suggested by the reviewer, a time-lapse confocal microscopy experiment of SNAP/FLAG-hGLP-1R localisation with EGFP-VAPB is now included in Supplementary Fig. 1d and Supplementary Movie 1.

2. The insulin secretion of pre-treatment WT control INS-1 832/3 β cell was lack in Fig2h-i and Fig7a, also the insulin secretion fold change under 11 mM glucose and Ex-4 was low.

As requested, these data have now been added in Supplementary Fig. 1h, Supplementary Fig. 1i, and Supplementary Fig. 7e. Secretion responses to GLP-1R agonists in INS-1 832/3 cells measured at 11 mM glucose typically range from 1.5 to 4-fold, as seen in previous studies from our lab (PMID: 36774542).

3. The localization of GLP-1R internalization of certain agonists in primary islets as shown in Fig. 3e appears to resemble alpha cells rather than beta cells.

The reviewer is probably referring to the apparent peripheral localisation of SNAP/FLAG-hGLP-1R signal in islets transduced with SNAP/FLAG-hGLP-1R adenoviruses. Please note that this apparent localisation relates to the adenoviral transduction being restricted to the outermost cell layers of intact islets, as well as our inability to image across the centre of the islet with conventional confocal microscopy due to the islet thickness, rather than any localisation of endogenous GLP-1R. We have repeated this experiment imaging exclusively at the outer layer of transduced islet cells rather than across the islet centre, and results are shown and quantified in Fig. 2c.

Additionally, in Fig. 8a-b, the PLA signal in primary islets of endogenous GLP-1R with VAP-B and SPHKAP should be detected alongside insulin and glucagon as cell markers to determine localization more accurately. Furthermore, the role of alpha cells should be thoroughly discussed in relation to the findings presented. In addition, it is not clear whether the PLA signals show any changes in Ex-4 pre-treatment or RNAi primary islets.

As requested, we have performed the suggested PLA experiment under both vehicle and exendin-4-stimulated conditions with insulin co-staining, which is now shown in Supplementary Fig. 9b. Please note that we were unable to label glucagon as well as insulin due to the incompatibility of antibody species with the ones used for the PLA. Please also note that, as explained above, the apparent peripheral PLA signal in our original experiment is due to our inability to image across the centre of the islet with regular confocal microscopy as well as the reduced accessibility of antibodies to the islet core, despite permeabilisation, and is not indicative of any restriction on the location of the interactions. We have strived to image only the outermost layer of the islet in the new PLA experiment shown in Supplementary Fig. 9b.

4. About Figure 4a, the localization pattern of GLP-1R was different from that in Figure 2e. The GLP-1R exhibited a more clustered localization. The authors should check this out. Additionally, a live-cell time-lapse imaging analysis should be done to track the contact among endosome, ER and mitochondria.

The GLP-1R is well known to undergo fast internalisation and endosomal trafficking in response to exendin-4 stimulation (PMID: 32436216). The receptor subsequently concentrates in a perinuclear location, which, depending on the individual cell, can take different times to occur, but typically occurs within 15-30 minutes post-exendin-4 stimulation. As suggested by the reviewer, we have now performed a time-lapse confocal microscopy experiment to track the localisation of the SNAP/FLAG-hGLP-1R to ERMCSs over time, and the results are shown in Fig. 3c and Supplementary Movie 2.

5. About Figure 4g, it's interesting that a single point mutation of SPHKAP abolished its mitochondrial localization. The authors should provide stronger evidence to prove SPHKAP's mitochondrial localization depends on its interaction with VAPB through the FFAT domain.

We have performed additional experiments that demonstrate that SPHKAP is in fact localised to ER loci likely to represent ERMCSs rather than inside mitochondria, with this localisation depending on SPHKAP binding to VAPB via its pFFAT motif. Please see data included in Fig. 4c,d, and Supplementary Fig. 6c and e. Please see also response to Reviewer 3 for the rationale behind the choice of SPHKAP mutation to disturb pFFAT-VAPB binding.

6. Since VAPB is central in organizing mitochondria-ER contact, the authors should check out whether disruption of these protein pairs, GLP-1R-VAPB, VAPB-SPHKAP and GLP-1R-VAPB-SPHKAP, affects the inter-organelle contacts. Electron microscopy analysis is preferable to compare the contact between these organelles.

We have now performed a similar TEM experiment to the one suggested by the reviewer in Fig. 7. Here, we note that exendin-4 stimulation of INS-1 832/3 cells results not only in mitochondrial remodelling (leading to reduced mitochondrial area and number of cristae per mitochondria as shown in Fig. 7a,b), but also in a reduction on ERMCS length (Fig. 7c), as well as in the number of ERMCSs per mitochondrial area [measured using a splitFAST approach (Fig. 7d,e)]. We speculate that this might represent a negative feedback loop to downregulate GLP-1R signalling from ERMCSs. Additionally, SPHKAP knockdown leads to reduced ERMCS length under vehicle conditions and no further effect of exendin-4 (Fig. 7c).

7. The internalized endosomes are usually destined to lysosomes or recycling endosomes. So how does this GLP-1R-VAPB-SPHKAP axis disassemble to release GLP-1R for recycling?

See previous answer for a possible hypothesis about ERMCS-localised GLP-1R signalling hub downregulation. Further experiments beyond the scope of the present paper will be required to establish the precise mechanism of nanodomain disassembly at this location. As SPHKAP pFFAT motif can be regulated by phosphorylation/dephosphorylation (PMID: 33124732), it is possible that a kinase-dependent mechanism is involved in regulating the activity of the GLP-1R-induced ERMCS-localised cAMP/PKA signalling complex.

8. The inter-organelle contacts in mammalian cells are usually organized by several protein pairs, so disrupting one of them often results in nonsignificant affection. In this research, the authors emphasized the importance of SPHKAP in GLP-1R dependent β cell function and survival. It would be beneficial to check out if the overexpression of SPHKAP or GLP-1R-VAPB-SPHKAP axis enhances insulin secretion with or without agonist stimulation.

We have now performed this experiment in Supplementary Fig. 7f and we have not detected any effect of SPHKAP over-expression on exendin-4-induced potentiation of insulin secretion in INS-1 832/3 cells.

Minor concerns

1. In Line 159-160, it would be better to give a description of the enriched proteins, such as the total number, and the number of mitochondria localized proteins.

We have expanded the description of relevant GLP-1R interactors in the revised manuscript discussion.

2. About Figure 2f, a statistical analysis of the colocalization efficiency between VAPB and GLP-1R was required.

Please see colocalisation results for this experiment shown here below (Fig. R1). Please note that SNAP/FLAG-hGLP-1R and EGFP-VAPB do not fully colocalise as the receptor remains in endosomes docked at VAPB-positive ER loci.

Fig. R1. Co-localisation (Manders' coefficient, tM1) between SNAP/FLAG-hGLP-1R and EGFP-VAPB WT *versus* P56S in exendin-4-stimulated INS-1 832/3 SNAP/FLAG-hGLP-1R cells, calculated with the Coloc2 plugin in Fiji; $n=3$; * $p<0.05$ by unpaired t-test.

3. In Figure 6h, the authors should specify how many cells were calculated in the legends. At least, 20 cells may be required to make the calculation solid.

Please see response to reviewer 2 above: "Please note that the n number included in the quantifications of changes in mitochondrial morphology refer to the number of independent biological repeats performed rather than the number of cells quantified. For INS-1 832/3 cells and dispersed islet cell experiments, we have quantified a total of 30 to 130 cells per experiment and condition, imaged from $n=4-5$ independent repeats, with 3 different fields of view imaged per repeat; For intact islet experiments, we have quantified all visible cells in 15 to 30 islets per experiment and condition, taken from $n=5-6$ independent repeats (meaning $n=5-6$ batches of islets from separate mice). We have added this information to the paper methods to clarify our sample sizes."

4. About Extended Data Fig. 5d, a statistical analysis of the colocalization efficiency is required. Additionally, the authors may check out whether ER morphology was affected due to the disruption of SPHKAP.

Please see colocalization results for this experiment shown here below (Fig. R2). Please note that SNAP/FLAG-hGLP-1R and EGFP-VAPB do not fully colocalise as the receptor remains in endosomes docked at VAPB-positive ER loci.

Fig. R2. Co-localisation (Manders' coefficient, tM1) between SNAP/FLAG-hGLP-1R and EGFP-VAPB in exendin-4-stimulated Control *versus* SPHKAP RNAi-treated INS-1 832/3 SNAP/FLAG-hGLP-1R cells, calculated with the Coloc2 plugin in Fiji; $n=7$; ns, non-significant by unpaired t-test.

5. About Figure 8, a western blot analysis is required to check out whether SPHKAP protein expression is disrupted or not in the islets by RNAi due to the difference of species.

See quantification of SPHKAP knockdown in islets transduced with Control *versus* SPHKAP shRNA in Supplementary Fig. 9c, as well as a relevant explanation for the level of transduction and knockdown expected from the whole islet included in the manuscript:

"Lentiviral transduction of cells within intact islets is restricted to the outermost cell layers so that only a small fraction of cells (which we estimate at ~25% of the total) is effectively transduced. However, those represent a higher functional cell pool due to necrosis of the islet core in non-vascularised isolated islets (PMID: 28832685), so that it is still possible to assess potential functional differences from a relatively small percentage of transduced cells. Accordingly, we could only detect a tendency for reduced SPHKAP mRNA levels in whole islets transduced with SPHKAP versus Control shRNA lentiviruses (Supplementary Fig. 9c), which nevertheless suggests a good level of downregulation from the pool of transduced cells. We next analysed some key signalling outcomes from these transduced islets."

Reviewer #5 (Remarks to the Author):

We thank Reviewer 2 for their thorough assessment, and the rest of the reviewers for their clear support for publication of our manuscript.

Please find below a point-by-point response to each of the reviewers' queries, as well as a description of the novel data added to the manuscript during this latest round of review.

Reviewers' comments:

Reviewer #1 (Remarks to the Author):

The revised manuscript quality has been improved, and most of my concerns were addressed and resolved.

We thank the Reviewer for their kind comment and their support for publication of our manuscript.

Reviewer #2 (Remarks to the Author):

I appreciate the authors' attempts to address my previous concerns. I recognize that the authors have made a strong effort to improve their quantitation of changes to mitochondrial morphology using a standardized approach in multiple biological conditions, and these findings are of interest. I also note that there is robust data in this manuscript which supports the modulation of mitochondrial PKA activity by GLP1-R.

We thank the Reviewer for their kind comments and support for data included in the manuscript.

However, I still believe that the authors have not performed an adequate job in controlling some of their experiments, and they have not sufficiently addressed my concerns regarding their co-IP results. I respect the authors attempt to address concerns through providing additional data with new techniques (CLEM), but I fear these efforts have confused their narrative, and raise more questions than they answer. Overall, I believe that while interesting observations are made, there is a significant number of experiments in this manuscript that raise significant doubts, which I will describe below. I still firmly believe that the observations made in this paper, while interesting, are over-interpreted, and at times, very speculative. Ultimately, the results in totality do not concretely support the model proposed, and thus I cannot support a Nature Communications publication.

Please find a point-by-point response to these criticisms here below, which we believe we have addressed in full during this re-review.

Major Comments

Co-immunoprecipitation and blotting experiments

With regards to my previous comment about issues with the vehicle control co-IP samples, I will elaborate further. In their AE-MS experiment in Fig. 1 that sets up the study examining the GLP1-R and VAPB interaction, the differential in enrichment is highly significant, indicating the vehicle control sample immunoprecipitated little to no VAPB. Despite this result, the authors continue to show VAPB

interaction in each of their co-IP vehicle control blots, suggesting contamination, and the IP of non-specific interactors.

The Reviewer queries about differences in enrichment for VAPB binding between the co-IP MS and the Western blot validation approaches – in this regard, we would like to point out that the MS is much more sensitive compared to the semiquantitative Western blot approach, which is prone to saturation and has a well-described low dynamic range. We nevertheless detect clear and statistically significant enrichment in VAPB - SNAP/FLAG-hGLP-1R binding under agonist-stimulated *versus* vehicle conditions (Fig. 1d,e), an observation that is additionally corroborated using several complementary techniques across the manuscript.

Furthermore, the authors do not include input, unbound, wash, and elution fractions in their blots, which would aid in establishing the efficiency of their enrichments, further complicating the interpretation of their results.

With regards to the input and output (unbound) fractions requested by the Reviewer, these are provided for 2 separate co-IP experiments (Supplementary Fig. 1c and Supplementary Fig. 6a), validating the efficiency of our immunoprecipitations (which in any case uses well-validated commercially available magnetic beads) – we would like to emphasise that, while we could provide further blots with input/unbound fractions for every co-IP included in the manuscript, we do not consider this necessary because we do not evaluate absolute interaction levels between the receptor (or any other protein bait) and the interacting factors, but rather the relative level of interaction per molecule of immunoprecipitated receptor under each condition tested, so that the exact amount of receptor pulled-down out of the total cellular receptor pool is irrelevant for our calculations or for the conclusions of the study. We always normalise our data to the exact amount of receptor pulled down per condition to quantify differences between conditions. This also means that any non-specific background interaction is equally present per condition.

Given that a significant amount of their data is based on transiently transfected overexpression systems, the authors do not control for the transfection efficiency, another factor which could influence the immunoprecipitation efficiency of their experiment.

Regarding the doubt expressed by the Reviewer about potential variability of transfection efficiencies, it is important to clarify that for all our experiments with transient transfections, cells are transfected in bulk prior to splitting them into different wells for the different conditions tested, so that there is the same transfection rate for all conditions per experimental repeat. As our statistical analyses are paired, this ensures that transfection efficiency is not a confounder in our quantifications or in the interpretation of our results.

Finally, the authors perform their analyses based on measuring the changes between the amount of immunoprecipitated factor to binding partner. Given the fact that the authors have not isolated the variables listed above, this means of comparison is subject to a significant degree of variability. Given they have not isolated these variables above, for example by using the same AE-MS based technique they used in Fig. 1, the conclusions drawn from these experiments are unsubstantiated.

See above for our comments on how we control for the variables referred to by the Reviewer (pull-down efficiency and transfection efficiency).

A major experiment that has not been performed but is essential to support the conclusions drawn is a co-immunoprecipitation of endogenous SPHKAP by VAPB in vehicle and exendin-4 treated cells, or a reciprocal pulldown of VAPB by SPHKAP.

Regarding the requested co-immunoprecipitation of SPHKAP and VAPB, this experiment has now been performed using EGFP-VAPB as bait with GFP Trap beads, followed by Western blot analysis of co-IP endogenous SPHKAP (Supplementary Fig 6h). We would also like to highlight that the association between SPHKAP and VAPB via the FFAT motif of SPHKAP has been validated in this manuscript using alternative techniques, and has previously been shown in Vierra *et al.*, Nat Commun. 2023, and more recently in Lee *et al.*, Res Sq [Preprint]. 2025.

I have the following issues with specific co-IP and blotting experiments in the manuscript:

- Supp. Fig. 2C: There is a very apparent unequal efficiency in immunoprecipitation. This experiment underlies a major conclusion about correlative GLP1R and VAPB interaction and should be repeated.
- Supp. Fig. 6D: Unequal IP levels of GLP1R.
- Supp. Fig. 6F,H: Tubulin amounts are different across samples.
- Supp. Fig. 7I: There is very unequal MIC19-3XHA immunoprecipitated. The comparison of phospho level to the IP MIC19 is inaccurate. The conclusion drawn that it is specifically related to SPHKAP and VAPB is therefore unsubstantiated.

Regarding the issues that the Reviewer raises about variability in receptor amounts pulled down in different conditions of certain co-IP experiments: We would like to reiterate that our results are normalised to the amount of receptor immunoprecipitated per condition (evaluated by re-blotting for SNAP in the same Western blot membranes after stripping), so that any variations in the overall amount of pulled-down receptor or bait protein are taken into account in our quantifications and therefore do not affect any of the conclusions of the study. Regardless, we have re-run blots for some co-IPs included in the study where the amounts of receptor or bait protein pulled down per condition were particularly variable, and these new blots are now included in Supplementary Fig. 2c and Supplementary Fig. 7k. The same consideration applies to Western blots showing knockdown efficiencies, as results are normalised to the amount of Tubulin per condition by stripping and re-blotting the same membranes with the corresponding antibody. We have nevertheless re-run these blots with more even amounts per lane to facilitate their visual interpretation, with these now included in Supplementary Fig. 6i and Supplementary Fig. 6j.

- Fig. 3A: The levels of TOM20 are uneven between samples, of which there are only 2. This therefore does not support the conclusion. Mitochondrial associated membrane isolation should be performed to confirm this.

This experiment has been removed from the manuscript and substituted by MAM purification as suggested by the Reviewer.

- Supp. Fig. 6A: There are unequal amounts of input across samples and no apparent differences in interactor levels.

Supplementary Fig. 6a corresponds to input and output (unbound) fractions from the co-IP results shown in Fig. 4a. The key factor here is the change in receptor level in our lysates between input and output fractions (before and after receptor pull-down), not the *absolute* amount of receptor in the input lysates. This change is consistent across all conditions and results in similar amounts of receptor being pulled down – the limiting factor is the number of beads used per condition for the IP, which is the same for all conditions, not the amount of receptor in the lysate, which is in excess as these cells over-express the SNAP/FLAG-hGLP-1R. Also please see above for an explanation on how we normalise the IP results to the amount of receptor pulled down per condition.

Colocalization

Another significant issue I have with the results is the almost complete absence of quantitative colocalization analysis. The authors provide a few instances of colocalization analysis using Manders' coefficient, yet this same quantitation is not performed elsewhere. This analysis is required to support the claims made by the authors. Below are some instances where this would be required to support the conclusions made:

- Fig. 1F: This experiment should show vehicle and exendin-4 treated cells.
- Fig. 1H: Similarly, this colocalization needs to be quantified. The magnification is too low to draw the conclusion of an interaction here. Vehicle controls should be quantified for statistical analysis.
- Supp. Fig. 1F: colocalization analysis is required along with vehicle control.
- Fig. 3B: colocalization analysis is required along with vehicle control.
- Fig. 3C: colocalization analysis is required along with vehicle control. Higher magnification required.

With regards to the Reviewer's concerns about the localisation of SNAP/FLAG-hGLP-1R under vehicle conditions in different experiments: we would like to clarify that in this study the receptor is labelled at the start of the experiment with a membrane impermeable fluorescent SNAP-Surface probe so that only the pool of receptors at the plasma membrane (and therefore able to bind to agonists) is visualised. SNAP/FLAG-hGLP-1R localisation to the plasma membrane under vehicle conditions has been observed by us in numerous occasions (Fig. 1f in Jones *et al.*, DOM 2022; Fig. 4e in Jones *et al.*, Nat Commun 2018; etc.). Co-localisation with intracellular structures will be, by definition, not observed under these conditions. Regardless, we have now provided images of SNAP/FLAG-hGLP-1R localisation under vehicle conditions in Supplementary Fig. 1d, Supplementary Fig. 1h, Fig. 3a, and Supplementary Fig 7b to meet the request of the Reviewer, which adds to previously included images of SNAP/FLAG-hGLP-1R localisation under vehicle conditions from Fig. 2c and the initial frame of Supplementary Movie 1.

Similarly, as requested by the Reviewer, we now provide quantifications of the degree of co-localisation for our imaging experiments, shown in Supplementary Fig. 1e, Supplementary Fig. 1i, Fig. 3b, Fig. 4e, and Supplementary Fig. 7c. Additionally, we would like to clarify that all our images are taken in high resolution confocal or super-resolution mode and are only presented in a small size to fit with the allocated figure space: the actual image size is a lot bigger than currently shown and we can easily provide magnified insets of existing images to clarify any doubts in localisation that the Reviewer might have. Alternatively, the Reviewer can simply zoom into the provided images to observe any details – even a 1000x zoom will not affect the resolution of our images.

The authors should use their SPLIC ERMCS tag to quantify the colocalization of three channels to support the claim that it is occurring at ERMCS in exendin-4 treated cells. A minimum of 20 regions of interest (ROI) is required for a quantitative change such as this.

The quantification of co-localisation between the receptor and the SPLICS biosensor suggested by the Reviewer in vehicle *versus* exendin-4-stimulated conditions has now been performed and is included in Fig. 3b. The Reviewer's suggestion to use the SPLICS construct in conjunction with EGFP-VAPB or SPHKAP-EGFP and SNAP/FLAG-hGLP-1R imaging is technically impossible due to the spectral overlap of the green fluorescent proteins present in these constructs. The Reviewer suggests using "3 channels" with the SPLICS construct to prove that the receptor is localised to ERMCSs: however, by definition, the use of SPLICS allows the visualisation of contacts between the ER and mitochondria due to protein complementation: labelling both organelles separately on top of the SPLICS marker is therefore unnecessary and would reduce the clarity of our observations. A similar experiment to the one suggested by the Reviewer (including mitochondria, ER/VAPB, and the receptor) has already been performed in the CLEM analysis of Fig. 3e,f, the cryo-CLEM images of Supplementary Fig 4, and the TEM analysis of gold-labelled SNAP/FLAG-hGLP-1R localisation in primary islets of Fig. 3g. We also now include a new additional confocal image of SNAP/FLAG-hGLP-1R, mitochondria and EGFP-VAPB in Fig. 3d.

- Fig. 3D: colocalization analysis is required along with vehicle control. What is the expected distance between the ER and Mito for an interaction to support signaling as is suggested? They appear too distant to draw this conclusion.

Regarding the CLEM data shown in Fig. 3e,f: the reviewer queries about the distance between the ER and mitochondria in the volume EM data. We estimate this to be around 15 nm in the example shown, which falls within the 10-30 nm typically predicted distance of a membrane contact site. We now additionally include a Z-stack video of the EM data used for the 3D reconstruction shown in Fig. 3f to facilitate its visualisation (Supplementary Movie 3).

- Supp. Fig. 4C: This shows a clear gap between endosome and mitochondrion, and does not support the conclusion that they are interacting.

With regards to Supplementary Fig. 4c, we detect a SNAP/FLAG-hGLP-1R-positive endosome docked to the ER which itself is interacting with a mitochondrion. We made the very best use of our limited access to cryo-CLEM technology to obtain this dataset and were unable to obtain further examples of 3D volume EM data. We still thought of interest to include this dataset in the Supplementary files as it highlighted the intimate association of GLP-1R-positive endosomes with the VAPB-positive ER, itself connected to a mitochondrion, using a cutting-edge technique. We have now clarified in the manuscript text that direct contacts appear to occur between GLP-1R-positive endosomes and VAPB-positive ER, with the VAPB-positive ER further engaging mitochondria, as also seen in both our room temperature CLEM data and in the immunoEM data from Fig. 3e-g, as well as by SPLICS co-localisation (both static and over time) in Fig. 3a-c and Supplementary Movie 2.

- Fig. 4C,D: Why is the SPLIC ERMCS construct not used to determine the localization to ERMCS? Also, the signal for SPHKAP is much lower than other examples in the paper. This result is therefore not reliable. Additionally, the authors claim that SPHKAP is part of a network associated with, but not inside mitochondria. I am unsure how they draw this conclusion based on this low resolution image. The authors would have to use their SPLIC system, or perform enrichment of mitochondrial associated membranes using density gradient ultracentrifugation. Colocalization quantification of 4D would also be required to support its association with the ER.

The Reviewer requests more evidence that SPHKAP is localised to ERMCSs rather than inside mitochondria. We already show SPHKAP-EGFP localisation data in a series of confocal z-stack slices which allows the visualisation of the spatial localisation of SPHKAP-EGFP *versus* the mitochondrial marker in 3D (Supplementary Fig. 6d). We have now additionally purified MAMs from mitochondria extracted from exendin-4-stimulated INS-1 832/3 SNAP/FLAG-hGLP-1R cells and analysed by WB for the presence of SPHKAP, VAPB and SNAP/FLAG-hGLP-1R as further proof (Supplementary Fig. 6g). Please note that our conclusion is in agreements with Vierra *et al.*, Nat Commun. 2023, and Lee *et al.*, Res Sq [Preprint]. 2025, which also detect SPHKAP bound to VAPB in the ER.

- Supp. Fig. 7B: It is unclear how this image is demonstrating colocalization. Colocalization analysis is required along with vehicle control.

This is now included as requested (both the vehicle control in Supplementary Fig. 7b and the quantification of co-localisation in Supplementary Fig. 7c).

- Supp Fig. 8A: The authors do not elaborate on how this is demonstrating mitophagy, as the image does not show engulfment of mitochondria. Given the absence of any known non-receptor mediated (PARKIN) or receptor mediated (BNIP3/NIX/FUNDC1) mitophagy markers on light microscopy experiments, the conclusions reached are incredibly speculative.

The reviewer queries about the example of mitophagy shown in Supplementary Fig 8a. Please note that this data should be considered in conjunction with our quantification of mitophagy by Keima assay shown in Fig. 5f as well as the results of the mCherry-Drp1 recruitment to mitochondria shown in Fig. 5g, Supplementary Fig. 8c, and Supplementary Movies 4 and 5.

- Supp. Fig. 8B: How is the DRP1 mCherry expression controlled for? DRP1 should be cytosolic, whereas this image shows almost a complete absence of signal in the red channel, suggesting low transfection levels, or insufficient laser power. This is not a fair comparison between control and treated samples, and does not support the conclusion of recruitment being specific to the experimental condition.

With regards to the levels of mCherry-Drp1 in these experiments, please note that, as explained above, our cells have been transfected in bulk prior to splitting them in different wells to perform the experiments under vehicle or exendin-4-stimulated conditions, so that the level of expression of mCherry-Drp1 is the same in both cases. We now include images with increased brightness and contrast to facilitate the cytoplasmic visualisation of mCherry-Drp1 under vehicle conditions as an answer to this query from the Reviewer.

Minor Comments

- The MIC19 phosphorylation result is overinterpreted. While it is an interesting observation, the blot appears preliminary, and the blot in the supplement has uneven loading. The conclusions drawn from this are speculative at best. I highly suggest that the authors tone down the language used in interpreting this observation.

PKA-dependent exendin-4-induced MIC19 phosphorylation has now been observed in 4 independent biological repeats and quantified in Fig. 5e. With regards to the effect of VAPB and SPHKAP RNAi, we have now repeated this experiment by performing analysis of changes in PKA phosphorylation state of MICOS complex components co-IPed with MIC19 in vehicle *versus* exendin-4-stimulated

conditions in Control *versus* VAPB+SPHKAP RNAi-treated cells, where we observe a complex pattern of VAPB/SPHKAP modulation of PKA-dependent phosphorylation not only restricted to MIC19 (Supplementary Fig. 7k). The manuscript text has now been modified to reflect these results, and we have toned down our interpretation of the results in the manuscript text as suggested by the reviewer.

- Fig. 1B: The overlapping names of each point in this plot make it almost completely indiscernible.

We have tried to fix the minor issue of name sizes in Fig. 1b by increasing the display font in the graph.

Reviewer #3 (Remarks to the Author):

The authors have fully addressed my concerns.

We thank the Reviewer for their kind comment and their support for publication of our manuscript.

Reviewer #4 (Remarks to the Author):

The authors have effectively addressed the previous concerns and implemented substantial improvements, which have significantly elevated the quality and credibility of the study.

We thank the Reviewer for their kind comment and their support for publication of our manuscript.

However, there are a few minor issues that could further strengthen the manuscript.

1. Colocalization analysis in Figures 4c and 4d: A colocalization analysis should be conducted and presented for Figures 4c and 4d. This would provide more in-depth insights into the relationships between SPHKAP localization and the defects of mitochondria.

See response to Reviewer 2 above, this quantification has now been performed and included in the manuscript.

2. It's pretty interesting that SPHKAP exhibited ERMCS-like localization pattern. The mitochondria-associated ER membranes could be purified and detected by western blot to further verify its localization.

See response to Reviewer 2 above, this quantification has now been performed and included in the manuscript.

3. Effect of SPHKAP overexpression on ERMCS: The knockdown of SPHKAP was shown to eliminate the disruptive effect of Ex-4 on ERMCS. It would be valuable for the authors to investigate whether the overexpression of SPHKAP has any impact on ERMCS. This additional experiment could further clarify the role of SPHKAP in the regulation of ERMCS and enhance the comprehensiveness of the study.

The suggested experiment of measuring ERMCS length following SPHKAP-EGFP overexpression has now been performed by TEM analysis of SPHKAP-EGFP transfected cells, and results have been added to Fig. 7c.

Reviewer #5 (Remarks to the Author):

Additionally, we have performed the following new experiments:

a) Two-colour MINFLUX super-resolution imaging and estimation of nearest neighbour distances between SNAP/FLAG-hGLP-1R and VAPB in exendin-4-stimulated INS-1 832/3 SNAP/FLAG-hGLP-1R cells, showing a distance of ~28 nm between GLP-1R-positive endosomes and VAPB-positive ER (Fig. 1i,j).

b) Co-IP of SNAP/FLAG-hGIPR with SPHKAP under vehicle *versus* GIP-stimulated conditions, demonstrating that the closely related incretin receptor GIPR also forms similar interactions with SPHKAP in pancreatic β -cells (Supplementary Fig. 6c).

c) Appearance of PKA-R1 α -EGFP-positive puncta in exendin-4-stimulated INS-1 832/3 cells, demonstrating the formation of PKA-R1 α -positive biomolecular condensates downstream of GLP-1R stimulation (Supplementary Fig 7e).